# The MondoA-dependent TXNIP/GDF15 axis predicts oxaliplatin response in colorectal adenocarcinomas

Jinhai Deng[1,2,24], Teng Pan [1,3,24], Dan Wang[1], Yourae Hong[4], Zaoqu Liu [5], Xingang Zhou[6], Zhengwen An[1], Lifeng Li[7], Giovanna Alfano[1], Gang Li[8], Luigi Dolcetti[1], Rachel Evans[1,9], Jose M Vicencio[1], Petra Vlckova [10], Yue Chen[11], James Monypenny[1], Camila Araujo De Carvalho Gomes [12], Gregory Weitsman[1], Kenrick Ng[13], Caitlin McCarthy[1], Xiaoping Yang[14], Zedong Hu[4], Joanna C Porter [15], Christopher J Tape[10], Mingzhu Yin [2], Fengxiang Wei[3], Manuel Rodriguez-Justo[16], Jin Zhang[17,18,19,20,21], Sabine Tejpar[3], Richard Beatson [1,15,22]✉ & Tony Ng [1,12,23]✉

## Abstract

Chemotherapy, the standard of care treatment for cancer patients with advanced disease, has been increasingly recognized to activate host immune responses to produce durable outcomes. Here, in colorectal adenocarcinoma (CRC) we identify oxaliplatin-induced Thioredoxin-Interacting Protein (*TXNIP*), a MondoA-dependent tumor suppressor gene, as a negative regulator of Growth/Differentiation Factor 15 (GDF15). GDF15 is a negative prognostic factor in CRC and promotes the differentiation of regulatory T cells (Tregs), which inhibit CD8 T-cell activation. Intriguingly, multiple models including patient-derived tumor organoids demonstrate that the loss of TXNIP and GDF15 responsiveness to oxaliplatin is associated with advanced disease or chemotherapeutic resistance, with transcriptomic or proteomic GDF15/TXNIP ratios showing potential as a prognostic biomarker. These findings illustrate a potentially common pathway where chemotherapy-induced epithelial oxidative stress drives local immune remodeling for patient benefit, with disruption of this pathway seen in refractory or advanced cases.

**Keywords** Colorectal Cancer; Oxaliplatin; TXNIP; GDF15; Functional Biomarker
**Subject Categories** Biomarkers; Cancer

## Introduction

Colorectal adenocarcinoma (CRC) has the fourth highest mortality among cancers, and is characterized by its aggression and heterogeneity (Siegel et al, 2017; Sung et al, 2021). Randomized controlled clinical trials have established that chemotherapy results in improved clinical outcomes (Schilsky, 2018). 5-FU (fluorouracil), oxaliplatin, and irinotecan are the foundation of first-line (FOLFOX) and second-line (FOLFIRI) (Hess et al, 2010) treatment, respectively. Despite mechanistic differences, all chemotherapy regimens induce apoptosis of replicating cells, leading to a reduction in tumor volume. Chemotherapeutic regimens have historically been regarded as immunologically silent or toxic, however, this view is being increasingly challenged with reports showing that these treatments can modulate immune cells within the tumor microenvironment (TME) (Zitvogel et al, 2008; Opzoomer et al, 2019).

Harnessing the immune system is crucial in achieving long-term durability of response (Kroemer et al, 2013), and chemotherapy reportedly activates antitumor immune responses through several mechanisms (Gebremeskel et al, 2017; Galluzzi et al, 2018;

[1]Richard Dimbleby Laboratory of Cancer Research, School of Cancer & Pharmaceutical Sciences, King's College London, London, UK. [2]Clinical Research Centre (CRC), Medical Pathology Centre (MPC), Cancer Early Detection and Treatment Centre (CEDTC), Translational Medicine Research Centre (TMRC), Chongqing University Three Gorges Hospital, Chongqing University, Wanzhou, Chongqing, China. [3]Longgang District Maternity & Child Healthcare Hospital of Shenzhen City (Longgang Maternity and Child Institute of Shantou University Medical College), 518172 Shenzhen, China. [4]Digestive Oncology Unit and Centre for Human Genetics, Universitair Ziekenhuis (UZ) Leuven, Leuven, Belgium. [5]Department of Interventional Radiology, The First Affiliated Hospital of Zhengzhou University, Zhengzhou, China. [6]Department of Pathology, Beijing Ditan Hospital, Capital Medical University, Beijing, China. [7]Internet Medical and System Applications of National Engineering Laboratory, Zhengzhou, China. [8]Department of General Surgery, Peking University Third Hospital, Beijing, China. [9]Translational Medicine, Oncology R&D, AstraZeneca, Cambridge, UK. [10]Cell Communication Lab, UCL Cancer Institute, 72 Huntley Street, London WC1E 6DD, UK. [11]Centre for Cancer Genomics and Computational Biology, Barts Cancer Institute, Queen Mary University of London, London, UK. [12]UCL Cancer Institute, University College London, London, UK. [13]Department of Medical Oncology, University College London Hospitals NHS Foundation Trust, London, UK. [14]Centre of Excellence for Mass Spectrometry, Proteomics Facility, The James Black Centre, King's College London, London, UK. [15]Centre for Inflammation and Tissue Repair, UCL Respiratory, Division of Medicine, University College London (UCL), Rayne Building, London, UK. [16]Department of Histopathology, University College London Hospital, London, UK. [17]3rd Department of Breast Cancer Prevention, Treatment and Research Centre, Tianjin, PR China. [18]Key Laboratory of Breast Cancer Prevention and Therapy (Ministry of Education), Tianjin, PR China. [19]Tianjin's Clinical Research Centre for Cancer, Tianjin, PR China. [20]Key Laboratory of Cancer Prevention and Therapy, Tianjin, PR China. [21]National Clinical Research Centre for Cancer, Tianjin Medical University Cancer Institute and Hospital, Tianjin, PR China. [22]Centre for the Tumour Microenvironment, Barts Cancer Institute, Queen Mary University of London, London, UK. [23]Cancer Research UK City of London Centre, London, UK. [24]These authors contributed equally: Jinhai Deng, Teng Pan. ✉E-mail: r.beatson@ucl.ac.uk; tony.ng@kcl.ac.uk

Krysko et al, 2012; Parra et al, 2018; Hodge et al, 2013). For example, chemotherapy-induced immunogenic cell death (ICD) leads to cells exposing or releasing damage-associated molecular patterns (DAMPs), such as HSP70, calreticulin, ATP, high-mobility group box 1 (HMGB1), type I IFN, and cancer cell-derived nucleic acids (Galluzzi et al, 2018; Krysko et al, 2012). These mediators drive antitumor immune responses via innate immune cells (dendritic cells (DC), macrophages, NK cells, γδT cells) and adaptive immune cells (T and B cells). In addition, chemotherapy has been shown to upregulate HLA expression and alter the peptides presented on MHC class I molecules, enabling an antitumor T-cell response through the expression of, and reaction to, neoantigens (Gebremeskel et al, 2017). Other chemotherapy-induced antitumor immunological mechanisms include the downregulation of immune checkpoint molecules (e.g., PD-L1) (Parra et al, 2018; Hodge et al, 2013), however, knowledge of these mechanisms has not yet been translated into a targeted chemo-immunotherapeutic treatment regimes. These antitumor immunological benefits of chemotherapy are, of course, balanced by pro-tumor impacts; chemotherapy-induced apoptosis itself, whether epithelial or immune, has been shown to be associated with immunosuppression in multiple cancers (Zhu et al, 2017; Gadiyar et al, 2020).

Thioredoxin-interacting protein (TXNIP), an alpha-arrestin protein, is commonly considered a master regulator of cellular oxidation, regulating the expression of Thioredoxin (Trx) via direct binding (Nishiyama et al, 1999; Junn et al, 2000). It has been seen to be silenced by genetic or epigenetic events in a wide range of human tumors, whilst *Txnip*-deficient mice have a higher incidence of spontaneous hepatocellular carcinoma (Sheth et al, 2006; Morrison et al, 2014; Jiao et al, 2019; Nishizawa et al, 2011). Consequently, *TXNIP* is considered a tumor suppressor gene (TSG) (Deng et al, 2023). In cell biology, TXNIP has been reported to regulate the cell cycle, oxidative stress responses, angiogenesis, glycolysis and the NLRP3 inflammasome (Klein Geltink et al, 2017; Y. Lu et al, 2021; Muri et al, 2021; Y. Yang et al, 2020; Wu et al, 2013; Waldhart et al, 2017; Kuljaca et al, 2009; Jeon et al, 2005; Yi et al, 2018). Previous studies have shown chemotherapy drives an increase of TXNIP expression leading to cell cycle arrest and death in epithelial cells (Di et al, 2021; Woolston et al, 2013), however, there are currently no studies that assess the effect of chemotherapy-induced TXNIP expression on the cells that survive chemotherapy, and an understanding of their impact on the TME.

Growth/differentiation factor 15 (GDF15), is a distant member of the TGF-β superfamily (Bootcov et al, 1997). At rest, GDF15 is produced at low levels by most epithelial tissues, however in cancers it is frequently overexpressed, particularly in hepatocellular carcinoma, prostate cancer and colorectal cancer (Boyle et al, 2009; Bauskin et al, 2006). Initially, GDF15 was identified as anti-tumorigenic protein with pro-apoptotic capability (Baek et al, 2001). However, its tumor-promoting effects are now well-documented to the extent that it is being promoted as a serological biomarker, with increased concentrations being associated with progression, recurrence, and death (Brown et al, 2006; Welsh et al, 2003), whilst overexpressing or knockout murine models have demonstrated its promotional role in tumorigenesis (Tsai et al, 2018). Immunologically, GDF15 is considered an anti-inflammatory factor, supported by the evidence that ubiquitous overexpression decreased systemic inflammatory responses (Kim et al, 2013) alongside its negative functional impact on macrophages, dendritic cells and NK cells, coupled with its ability to induce Tregs (Roth et al, 2010; Y. Gao et al, 2021; Wang et al, 2021). As a soluble protein, GDF15 exerts its effects by binding to its cognate receptor, GDNF-family receptor a-like (GFRAL) (Hsu et al, 2017; Emmerson et al, 2017; Mullican et al, 2017; L. Yang et al, 2017) or interaction partner, CD48 receptor (SLAMF2) (Wang et al, 2021), with the latter still requiring additional verification.

In this work, using a variety of in vitro models, including patient-derived tumor organoids (PDTOs) we demonstrate that oxaliplatin-based chemotherapy reshapes the TME via an increase in ROS-mediated MondoA-dependent TXNIP expression, resulting in decreased expression and secretion of GDF15, leading to a decrease in regulatory T-cell (Treg) differentiation and increase in CD8 T-cell activity. To support the concept of a TXNIP/GDF15/Treg regulatory axis in situ, an anti-correlation of TXNIP and GDF15 was observed in matched fresh patient tissue (pre and post chemotherapy), fixed tissue, whole tumor transcripts, and single-cell seq data, while GDF15 was further seen to correlate with Foxp3/FOXP3 in fixed tissue samples and the TCGA transcriptomic dataset. With regards clinical impact, both low TXNIP and high GDF15 were shown to be indicators of poor prognosis when assessing protein or transcript expression, allowing us to postulate that the inversion of this phenotype through chemotherapeutic treatment may be associated with positive outcome. Furthermore, this oxaliplatin-dependent inversion was seen to be absent in CRC cell lines derived from secondary sites, in a similar manner to chemotherapy-resistant CRC cell lines, with aggressive primary tumors also showing a similar trend. These data suggest that the loss of a responsive TXNIP/GDF15 axis allows for tumor survival, with this concept being supported by transcriptomic analysis of primary and metastatic disease and responsive and non-responsive cases. Beyond the biology, this study illustrates the potential of a pre-treatment GDF15/TXNIP ratio as a tool to predict chemotherapeutic response in patients, potentially allowing for appropriate immunotherapy (e.g., GDF15 antagonists) to be administered to non-responders at an early timepoint in a precise and informed manner.

## Results

### TXNIP is upregulated after chemotherapy and associated with favorable prognosis

TXNIP is relatively well-studied in cancer and has been reported to have tumor-suppressive effects as discussed (Masutani, 2022). In CRC, TXNIP expression has been observed to be decreased in tumor cases compared to normal tissues (Takahashi et al, 2007). In support of this, analysis of the TCGA COAD (CRC) database showed decreased *TXNIP* mRNA in tumor samples compared to normal controls (Appendix Fig. S1A). To validate this, we collected 42 CRC patient samples and observed that tumors presented lower expression of TXNIP as compared to adjacent normal tissue (ANT) (Fig. 1A,B). We then used single-cell transcriptomics to confirm the same observation in epithelial cells in CRC (Fig. 1C).

TXNIP has previously been shown to be increased during chemotherapy-induced cell death (Di et al, 2021; Woolston et al, 2013). As TXNIP is considered vital in the regulation of intracellular reactive oxygen species (ROS), which are generated by chemotherapeutic treatment, we questioned whether TXNIP

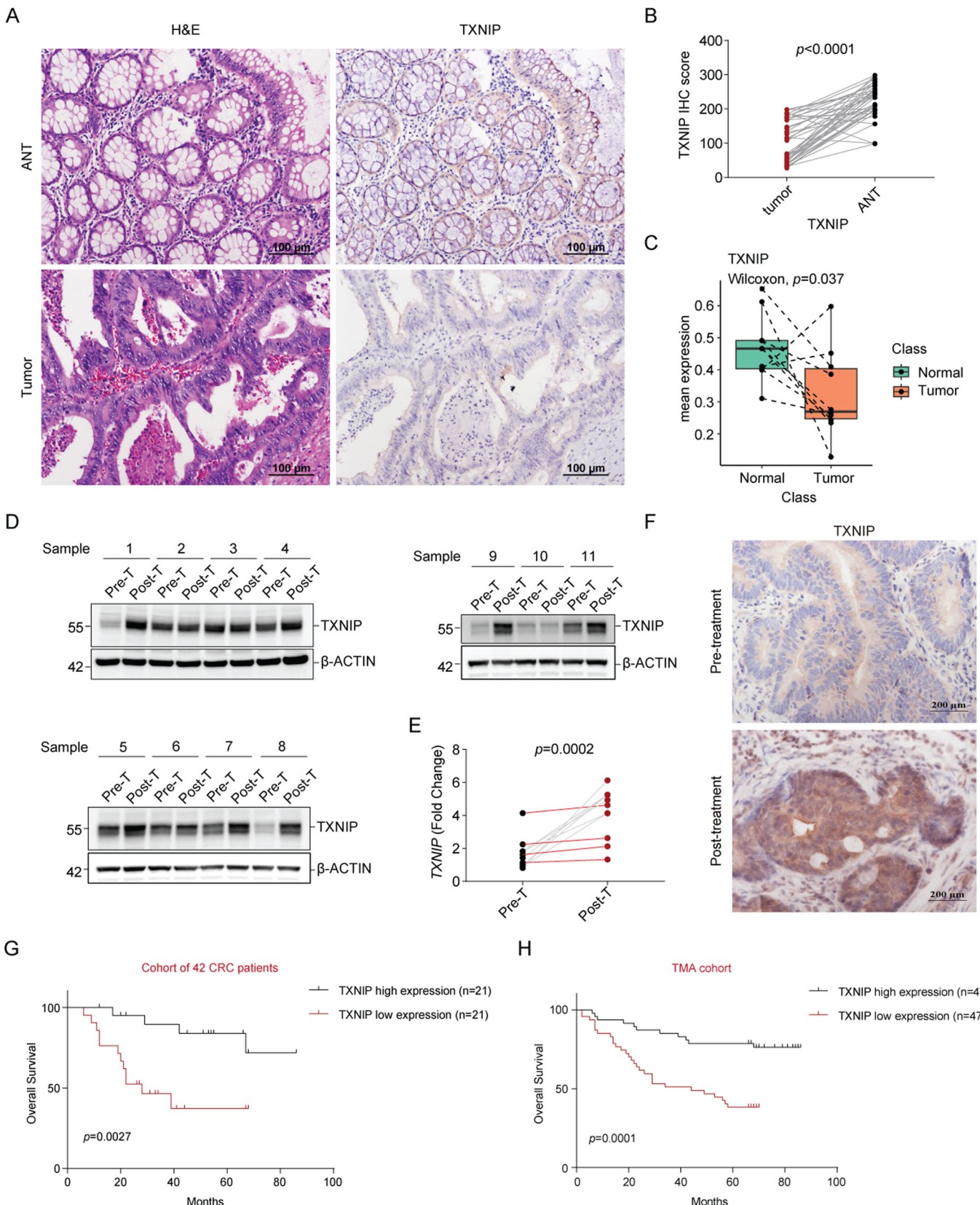

◄ **Figure 1. Lower TXNIP expression is observed in CRC tumor samples, however it is increased post chemotherapeutic treatment.**

Low levels of TXNIP are associated with poor prognosis. (A) Example of H&E and TXNIP staining in primary CRC tumors and matched adjacent normal tissue (ANT) samples. Magnification ×200. (B) Pooled TXNIP scoring from primary CRC tumors and matched ANT samples ($n = 42$) (Appendix Table S1). (C) *TXNIP* transcript expression in single epithelial cells derived from matched primary CRC tumors and adjacent normal colon ($n = 10$ pairs). The box plots describe the median and interquartile range (IQR), and the whiskers depict the 1.5× IQR. Data from GSE132465. (D) TXNIP expression in 11 paired treatment-naive (Pre-T) tumor samples and oxaliplatin-based neo-adjuvant chemotherapy-treated tumor samples (Post-T). Appendix Table S2 (E) *TXNIP* mRNA levels in samples from (D) (aggressive [≥T4M1] cases highlighted in red). $N = 11$ (Appendix Table S2). (F) Example of TXNIP staining in matched Pre-T and Post-T samples. Magnification ×400. (G, H) Kaplan–Meier analysis of overall survival in CRC patients with different TXNIP staining scores from a cohort of 42 CRC patients (G), Appendix Table S1, and CRC tumor tissue microarray ($n = 94$), Appendix Table S3 (H). Data in (G, H) were analyzed using two-tailed log-rank test; data in (B, E) were analyzed using two-tailed, two-sample unpaired Student's *t* test. Data in (C) were analyzed using Wilcoxon paired test. Values were expressed as mean ± SEM. ***$P < 0.001$, ****$P < 0.0001$, vs. Control. Source data are available online for this figure.

could additionally act as a survival factor. To test this, we took biopsies from CRC patients before and after oxaliplatin-based chemotherapy and measured TXNIP, finding an increase in expression after chemotherapy in 8/11 patients (Fig. 1D–F). Somewhat presciently, in light of subsequent findings, 3/11 patients (patients 2, 3 and 10), with advanced disease, showed no increase after treatment (Fig. 1D–F). To assess for any association between TXNIP expression and disease progression, and to test whether the oxaliplatin inspired increase we had observed would be of benefit to the patient, we used two historic tissue cohorts and two publicly available transcriptomic datasets. High levels of both the protein (significantly) and the transcript (not significantly) were seen to be associated with favorable prognosis (Fig. 1G,H; Appendix Fig. S1B,C). Moreover, in historic patient cohorts, TXNIP expression was observed to be significantly negatively correlated with clinical stage and lymph node metastasis, with no correlation with respect to age, sex, or tumor size (Appendix Tables S4 and S5).

## MondoA regulates chemotherapy-induced TXNIP expression

To assess the relative expression change of *TXNIP* after chemotherapy, compared to other transcripts, we used primary colorectal cancer cell lines (DLD1 and HCT15) and treated them with a clinically relevant concentration (10 μM) (Schmidt et al, 2022) of oxaliplatin or vehicle. The dead cells were discarded and the live cells were sent for RNA sequencing analysis. The results showed that *TXNIP* was upregulated as one of the top differentials in both cell lines (Fig. 2A,B; Datasets EV1 and EV2); validated by RT-PCR and western blot (Figs. 2C–F and EV1A–D). Further to this, oxaliplatin upregulated TXNIP in a time-dependent (Figs. 2C,D and EV1A,B) and dose-dependent fashion (Figs. 2E,F and EV1C,D). Three-dimensional (3D) cell models are reported to be more accurate in mimicking in vivo features such as drug responses (Jensen and Teng, 2020), therefore, we assessed whether this response was observed in cell line-derived spheroids and two patient-derived organoids. In both models, we observed the upregulation of *TXNIP* mRNA (Fig. EV1E–H) and TXNIP protein (Fig. EV1I–L) after oxaliplatin treatment, with spheroids showing greater responsiveness. This difference is most likely due to culturing conditions or differences in the number and location of cycling cells.

The thioredoxin (Trx) antioxidant system includes NAPDH, thioredoxin reductase (TrxR), and Trx. TXNIP is essential for redox homeostasis due to its ability to bind to Trx and inhibit Trx function and expression (J. Lu and Holmgren, 2014). As discussed, oxaliplatin treatment induces ROS (Rottenberg et al, 2020),

whilst oxidative stress is associated with TXNIP expression (Ogata et al, 2013). As such, we went back to first principles and assessed the impact of different concentrations of glucose on TXNIP induction +/− oxaliplatin treatment, finding a concentration-dependent effect (Fig. EV2A). Intriguingly, high glucose alone was able to induce increased TXNIP expression. We then assessed if oxaliplatin treatment drove an increase in glucose uptake, with this seen at concentrations >10 mM (Fig. EV2B). Next, to investigate the impact of glucose metabolism on TXNIP induction, we treated cells with Antimycin A, an inhibitor of oxidative phosphorylation, finding a complete block in oxaliplatin-induced TXNIP (Fig. EV2C). However, we cannot state whether this finding is driven by a decrease in ROS or by the inhibitor's effect on ATP-mediated activation of MondoA (Wilde et al, 2019). To attempt to address whether the increase in TXNIP expression after oxaliplatin treatment was mediated by ROS we first observed that oxaliplatin was able to increase ROS production (Fig. EV3A,B), in line with previous studies (Santoro et al, 2016). We then administered N-acetyl-L-cysteine (NAC), a reactive oxygen species inhibitor, and observed no increase in oxaliplatin-dependent TXNIP expression post treatment (Figs. 2G, H and EV3C).

We next investigated which transcription factor may mediate ROS-induced *TXNIP* expression. The RNA-seq data revealed 23 differentially expressed genes (DEGs) shared between both cell lines, including *TXNIP* (Fig. EV3D,E; Dataset EV2). One of these DEGs, was arrestin domain-containing protein 4 (*ARRDC4*). *ARRDC4* was increased after oxaliplatin treatment (Fig. EV3D,E; validated by RT-qPCR (Fig. EV3F,G)), and, like TXNIP, this increase was shown to be dependent on ROS (Fig. EV3F,G). TXNIP and ARRDC4 are paralogs showing 63% similarity and are both regulated by the transcription factor MondoA (Stoltzman et al, 2008; Peterson et al, 2010), indeed TXNIP and ARRDC4 have been reported to be highly MondoA-dependent (Richards et al, 2018). We therefore assessed MondoA expression before and after oxaliplatin treatment, finding no change (Fig. EV3H). With MondoA having previously been shown to shuttle into the nucleus to carry out its functions (Richards et al, 2018), we assessed for MondoA in different cellular fractions. The result showed MondoA was indeed translocated into the nucleus after oxaliplatin treatment (Fig. EV3I).

To assess the role of MondoA/*MLXIP* in oxaliplatin-induced TXNIP upregulation, we established *MLXIP*-knockout cells and *MLXIP*-knockdown cells using CRISPR-Cas9 and siRNA, respectively. Using these models we saw that the removal or decrease of MondoA/*MLXIP* resulted in the loss of increased expression of both TXNIP and ARRDC4 after oxaliplatin treatment (Fig. 2I–L). To further strengthen our conclusions, we used ChIP-PCR to verify

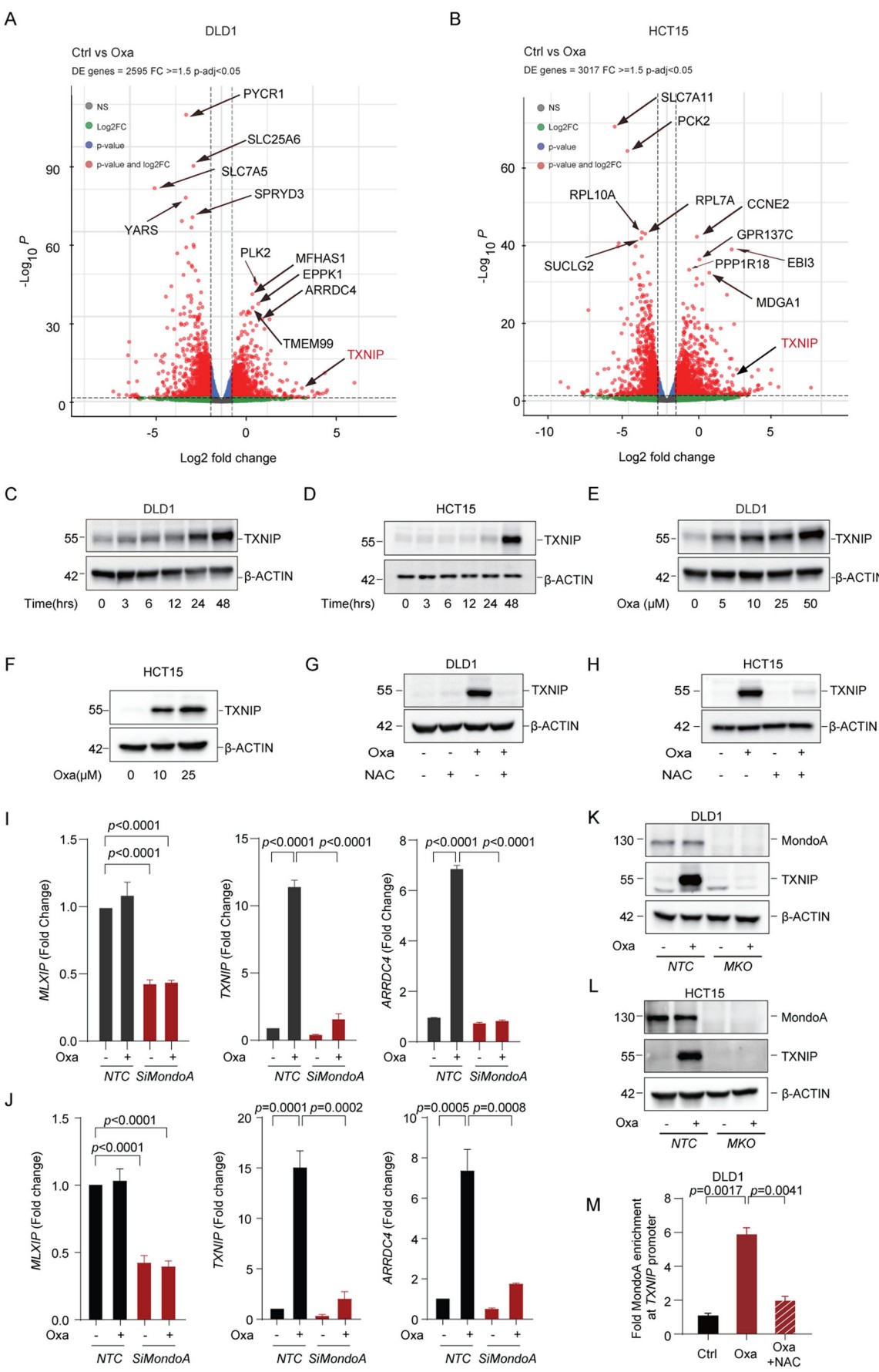

**Figure 2.   ROS mediates chemotherapy-induced TXNIP expression by modulating MondoA.**

(A, B) DLD1 cells (A) or HCT15 cells (B) were treated with 10 μM oxaliplatin for 48 h and surviving cells were analyzed by RNA sequencing. A volcano plot (log2 FC versus negative log of $P$ value) was used to visualize statistically significant gene expression changes (fold ≥1.5 and adjusted $P$ value < 0.05). *TXNIP* and other highly differentially expressed (DE) transcripts are labeled. The number of DE genes is indicated in the upper left. Three biological replicates per group (Dataset EV1). (C, D) Western blotting analysis of TXNIP expression in DLD1 cells (C) or HCT15 cells (D) treated with oxaliplatin at different time points. β-ACTIN was used as an internal reference. (E, F) Western blotting analysis of TXNIP expression in DLD1 cells (E) or HCT15 cells (F) treated with oxaliplatin at different doses for 48 h. (G, H) Immunoblot analysis of TXNIP in DLD1 cells (G) or HCT15 cells (H) treated with N-acetyl-L-cysteine (1.25 mM) or oxaliplatin (10 μm) or the combinational treatment for 48 h. (I, J) Quantification of *MLXIP* (MondoA), *TXNIP* and *ARRDC4* mRNA in DLD1 cells (I) or HCT15 cells (J) upon knockdown of *MLXIP* by siRNA after treatment with 10 μm oxaliplatin treatment for 48 h. (K, L) Immunoblot analysis of TXNIP expression in *MLXIP*-knockout DLD1 cells (K) or HCT15 cells (L) after 10 μm oxaliplatin treatment for 48 h. (M) MondoA occupancy on the promoters of *TXNIP* in DLD1 cells treated with 10 μm oxaliplatin or the combinational treatment with NAC (1.25 mM) for 48 h. Results shown, excluding (A, B), are representative of three independent experiments. All values were expressed as mean ± SEM. Two-tailed Student's $t$ test; **$P$ < 0.01, ***$P$ < 0.001, ****$P$ < 0.0001, vs. Control. Source data are available online for this figure.

the dependence of these processes on MondoA/*MLXIP*. Relative to the control, the amount of MondoA on the *TXNIP* promoter was significantly increased after oxaliplatin treatment, which was compromised after combined treatment with NAC (Fig. 2M). Taken together, these results demonstrated that ROS production was responsible for oxaliplatin-induced TXNIP overexpression by activating MondoA transcriptional activity.

## TXNIP regulates the expression and secretion of GDF15

TXNIP has been reported to regulate both the innate and adaptive arms of the immune system (Guo et al, 2022). In support of this, we found TXNIP expression to be positively associated with the expression of T-cell markers, antigen presentation, and cytokine transcripts when using the TCGA COAD dataset (Appendix Fig. S2A). The enrichment of TXNIP in the cytoplasm indicated that TXNIP may mediate antitumor effects by regulating immunologically relevant cytoplasmic processes (Appendix Fig. S2B,C) (R. Zhou et al, 2010), for example, the NLRP3 inflammasome (R. Zhou et al, 2010). The formation and activation of the NLRP3 inflammasome leads to self-cleavage and activation of caspase 1, which in turn promotes the release of the pro-inflammatory cytokine IL-1β. However, the correlation between *TXNIP* and *IL1B* or *IL18* was not significant (Appendix Fig. S2A). Similarly, knockout of *TXNIP* led to no alteration in caspase 1 activation and IL-1β production (Appendix Fig. S2D,E), with no detectable IL-1β protein in the supernatants, suggesting TXNIP failed to activate the NLRP3 inflammasome upon chemotherapeutic treatment.

We therefore considered whether TXNIP may be capable of regulating the expression and/or secretion of other immunologically relevant soluble factor(s) from the epithelial cell. To this end, we performed mass-spectrometric analysis of supernatants collected from non-targeting control (NTC) and *TXNIP*-KO (TKO) DLD1 cells and identified 157 differentially expressed soluble proteins ($P$ < 0.05). Complete proteomics data can be found in Dataset EV3. Growth/Differentiation Factor 15 (GDF15) was the most highly differentiated secreted protein associated with *TXNIP* loss (Fig. 3A). This result was confirmed using a cytokine array, where GDF15 was additionally seen to be secreted at lower levels in response to oxaliplatin; in line with the upregulation of TXNIP (Fig. 3B). These results showed that oxaliplatin decreases GDF15 secretion in a TXNIP-dependent manner, and that the knockout of *TXNIP* alone could drive the secretion of GDF15. Intriguingly, other factors were seen to be altered in a similar manner to GDF15, for example plasminogen activating inhibitor

(PAI-1; SERPINE1. Fig. 3B Row I, columns 1 and 2) suggestive of a *TXNIP*-dependent signature, however with the proteomics showing GDF15 as the dominant factor, we focused on this pathway.

Having established a link between GDF15 and TXNIP, we next assessed the effects of oxaliplatin treatment on GDF15. The downregulation of GDF15 was more pronounced at later time points and higher drug dosages; the opposite trend to TXNIP (Appendix Fig. S3A,B). Using western blotting, we showed that *TXNIP*-KO rescued the inhibitory effects of oxaliplatin on GDF15 expression in DLD1 cells (Fig. 3C,E), with a similar pattern being observed in *TXNIP*-KO HCT15 cells (Fig. 3D,F). It is important to note, however, that we saw clear evidence that *TXNIP* was not solely responsible for the downregulation of GDF15 post oxaliplatin treatment (Fig. 3C–G; Appendix Fig. S3E). In contrast, *TXNIP*-overexpressing DLD1 cells showed lower GDF15 expression compared to control cells (Appendix Fig. S3C,D). We quantified soluble GDF15 concentrations by ELISA finding >5 ng/ml in the supernatant of *TXNIP*-KO cells (Fig. 3G), while a higher expression of GDF15 was also detected in *TXNIP*-KO PDTOs (Fig. 3H). Next, using confocal imaging, we observed GDF15 was enriched in the cytoplasm in untreated cells, suggestive of it being stored in secretory granules, with no staining seen after oxaliplatin treatment. In line with immunoblot analysis, confocal imaging showed *TXNIP*-KO cells expressing more GDF15, which, unlike the control, was retained after oxaliplatin treatment (Fig. 3I; Appendix Fig. S3E).

As ROS mediated the activation of the MondoA-TXNIP axis, we aimed to assess the effect of these factors on GDF15 expression. In line with our previous findings, knocking down *MLXIP* decreased the expression of TXNIP, but increased GDF15 expression (Fig. 3J; Appendix Fig. S3F), suggesting the involvement of MondoA/*MLXIP* in the regulation of GDF15 expression. Furthermore, pre-incubation of the target cells with NAC abolished the suppression of GDF15 by oxaliplatin, which was partially rescued by overexpressing *TXNIP* (Appendix Fig. S3F), suggestive of the important role of ROS in GDF15 regulation. Collectively, these data demonstrated the activation of MondoA by ROS modulates both TXNIP and GDF15.

Finally, given our previous data (Fig. EV3) we looked to assess the role of ARRDC4 on GDF15 expression. In the absence of oxaliplatin, knocking down *ARRDC4* in DLD1 and HCT15 cells drove an increase in GDF15. When challenged with oxaliplatin, both ARRDC4 and TXNIP expression increased and GDF15 decreased. When the *ARRDC4* knockdown was challenged TXNIP increased further and GDF15 decreased further

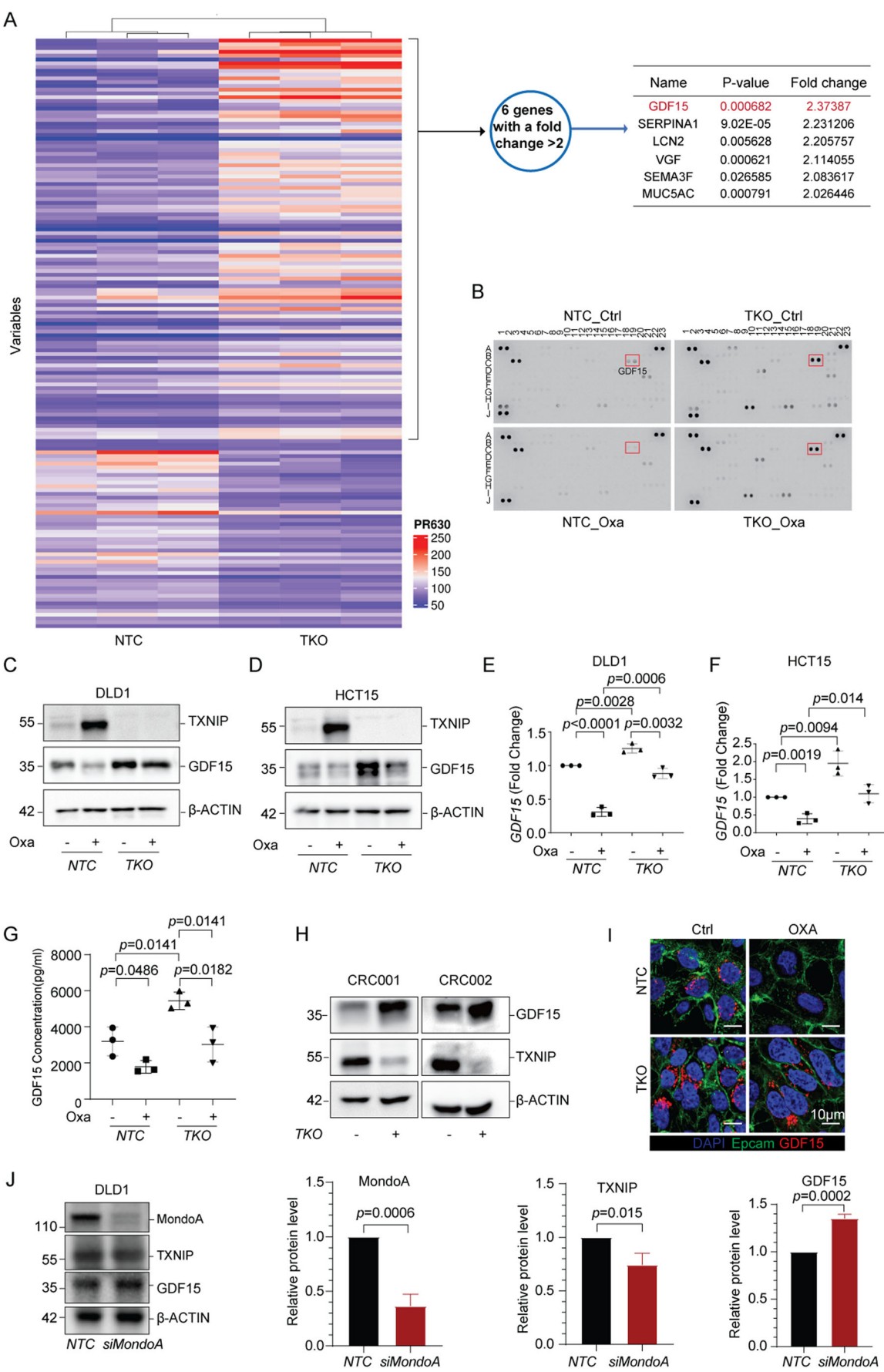

**Figure 3.   TXNIP regulates GDF15 expression.**

(**A**) Proteomic analysis of the conditional media from *TXNIP*-KO (TKO) and control (NTC) DLD1 cells as assessed by mass spectrometry. Heatmap illustrating differentially expressed proteins (left panel) and table showing the top six upregulated proteins in conditional media from TKO cells (right table) (Dataset EV3). (**B**) In total, 105 plex cytokine arrays incubated with conditional media from TKO and NTC cells with or without 10 μM oxaliplatin treatment for 48 h. The respective GDF15 spot is highlighted (red box). (**C, D**) Immunoblotting of TXNIP and GDF15 in NTC and TKO DLD1 cells (**C**) and NTC and TKO HCT15 cells (**D**) with or without drug treatment (10 μm oxaliplatin for 48 h). (**E, F**) Pooled densitometry data from three repeats of (**C, D**). Standard error bars shown. (**G**) GDF15 concentration in conditional media for (**E**) were determined by ELISA. Standard error bars are shown. (**H**) Immunoblot of TXNIP and GDF15 in NTC (TKO-) and TKO (TKO + ) PDTOs: CRC001 (left panel), CRC002 (right panel). (**I**) Immunofluorescent detection of GDF15 in NTC and TKO DLD1 cells with or without 10 μm oxaliplatin treatment for 48 h as assessed by confocal microscopy. DAPI (blue), Epcam (green), GDF15 (red). (**J**) Left panel: Immunoblotting of MondoA, TXNIP and GDF15 in *MLXIP*-knockdown (siMondaA) and control (NTC) DLD1 cells. Middle and right panels: MondoA, TXNIP and GDF15 protein expression levels in siMondoA (si*MLXIP*) and NTC DLD1 cells. The results shown are representative of three independent experiments, excluding (**A, B**). All values were expressed as mean ± SEM. Two-tailed Student's *t* test; *$P < 0.05$, **$P < 0.01$, ***$P < 0.001$, ****$P < 0.0001$, vs. Control. Source data are available online for this figure.

(Appendix Fig. S3G–J). Given the common regulatory pathways and homology between TXNIP and ARRDC4, and their similar functional roles, we suggest these data are evidence of redundancy within this system.

## GDF15 expression is upregulated in CRC and associated with poor prognosis

Consistent with previous reports (Bauskin et al, 2006; Wallin et al, 2011a; Brown et al, 2003) *GDF15* was observed to be upregulated in CRC tumor samples in comparison with normal tissue or epithelial cells by both TCGA COAD and scRNA epithelial transcriptomic analyzes respectively (Appendix Fig. S4A,B). This observation was validated by IHC staining (Fig. 4A,B). To assess if the inverse relationship of TXNIP and GDF15 we had observed in vitro could be observed in situ, we assessed relative transcriptomic and protein expression using the TCGA COAD dataset and historic patient samples, respectively, finding the same inter-relationship (Fig. 4C,D). Using the same pre-T and post-T fresh patient samples described in Fig. 1, we probed for GDF15, finding decreased GDF15 expression after treatment, except for the same three aggressive cases which had previously been observed to show no increased TXNIP expression (Fig. 4E–G).

We then sought to understand the clinical relevance of GDF15 in CRC. When assessing for the impact of increased expression of GDF15 on survival, we found associations between low GDF15 and improved outcome at the protein level in tissue (Fig. 4H,I), and in two independent public transcriptomic datasets (Appendix Fig. S4C,D), suggesting that GDF15 contributes to tumor progression in CRC, in accordance with previous studies (Wallin et al, 2011a; Brown et al, 2003). In an opposite manner to TXNIP, GDF15 showed a significantly positive correlation with clinical stage and lymph node metastasis in CRC specimens (Appendix Tables S6 and S7).

## The role of TXNIP–GDF15 axis in immune regulation

GDF15 has been reported to have multiple immunological impacts; however, some reports have been queried owing to the discovery of contaminating TGF-β1 in recombinant GDF15 preparations (Olsen et al, 2017). As such, to explore the immune impacts of GDF15, we opted to predominantly use cellular systems and resultant conditioned supernatant (Appendix Fig. S5A,B). In light of other secreted factors being seen to be regulated by TXNIP (Fig. 3A,B), we included double knockouts (*TXNIP* and *GDF15* knockout;

GTKO) as well as an overexpression system (*GDF15*a) to test for GDF15-specific effects. However, we do not know the impact of knocking out or overexpressing *GDF15* on the broader secretome.

When stimulating PBMCs with anti-CD3 and anti-CD28 in the presence of GDF15-enriched conditioned media from the TXNIP-KO cell line, we observed a small but significant decrease in cell number that was reversed using supernatant from a *TXNIP/GDF15* double KO cell line (Fig. 5A,B). Further analysis showed that both CD8 and CD4 T-cell proliferation was inhibited by GDF15-enriched supernatant (Fig. 5C–F), with IFNγ concentrations in the supernatant also being seen to lower (Fig. 5G,H).

A recent paper has shown that GDF15 is able to drive the differentiation of regulatory T cells (Tregs) from naive CD4s via an interaction with CD48 (Wang et al, 2021). Working on the hypothesis that it was Tregs that were inhibiting the T-cell proliferation and IFNγ release within the mixed PBMC population, we observed a GDF15-dependent increase of Foxp3 within the CD4 pool (Fig. 5I), however to a much lesser extent than when using recombinant active TGFβ1 (Fig. 5J). To support these data, we assessed for associations between GDF15 and *FOXP3*/Foxp3 in TCGA COAD dataset and our historic 42 patient cohort, respectively, finding a significantly positive correlation between *GDF15* and *FOXP3* and enrichment of Foxp3 in the GDF15 high cases (Fig. 5K–M). Furthermore, when stimulating naive CD4 T cells in the presence of GDF15-enriched supernatant we were able to both differentiate these cells into functional Tregs and also block the generation of this functionality using an anti-CD48 antibody (Fig. 5N,O). However, it must be stressed that the binding and functional impacts of GDF15's interaction with CD48 still require further verification.

To assess if the *GDF15*-dependent presence of Tregs may be associated with a decrease in activated cytotoxic CD8 T cells, we interrogated the TCGA COAD dataset. We found that low *GDF15* tumors carried significantly higher levels of *CD8, CD69, IL2RA, CD28, PRF1, GZMA, GZMK, TBX21, EOMES*, and *IRF4* (Appendix Fig. S6A,C,D,E,G,H,I,K,L,M); transcripts indicative of activated cytotoxic CD8 T cells. High *GDF15* tumors were enrichment for *FOXP3* and, interestingly, *RORC* (Appendix Fig. S6J,N). These data support the hypothesis that GDF15 induces Foxp3$^{+ve}$ Tregs which inhibit CD8 T-cell proliferation and activation in the TME.

Loss of TXNIP/GDF15 axis functionality in advanced disease and the use of pre-treatment GDF15/TXNIP ratio as a biomarker of clinical response.

With high GDF15, Treg infiltration, and CD8 T-cell dysfunction all being shown to be associated with poor prognosis in CRC

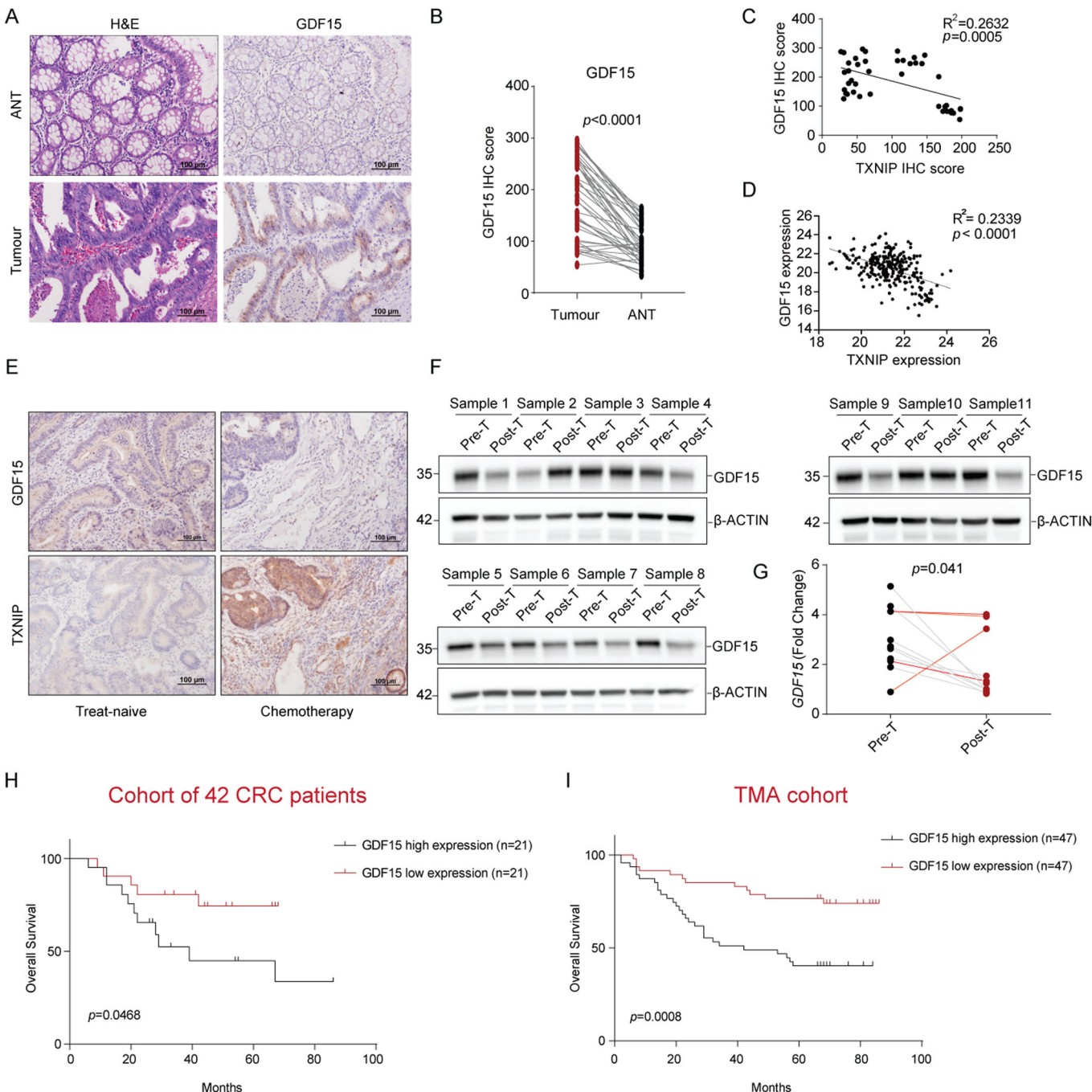

**Figure 4. Higher GDF15 expression is observed in CRC tumor samples, however it is decreased post chemotherapeutic treatment.**

High levels of GDF15 are associated with poor prognosis. (A) Detection of GDF15 in both tumor and adjacent normal tissue (ANT) samples from patients with primary colorectal cancer. Magnification ×200. H&E staining the same as in Fig. 1A. (B) Statistical analysis of GDF15 IHC score between ANT and tumor tissue ($n = 42$) (Appendix Table S1). (C, D) Correlations of TXNIP and GDF15 protein (cohort of 42 CRC patients), Appendix Table S1 (C) and *TXNIP* and *GDF15* transcripts (TCGA COAD) (D). Pearson correlation coefficients ($R^2$) are indicated. (E) Sequential sections from colorectal tumor samples collected pre- and post neo-adjuvant chemotherapy. Detection of TXNIP and GDF15 by IHC. TXNIP image the same as used in Fig. 1F. (F) GDF15 expression in 11 paired treatment-naive (Pre-T) tumor samples and oxaliplatin-based neo-adjuvant chemotherapy-treated tumor samples (Post T) (Appendix Table S2) (G) *GDF15* mRNA levels in samples from (F) (aggressive (≥ T4M1) cases highlighted in red). $N = 11$ (Appendix Table S2). (H, I) Kaplan–Meier analysis of overall survival in CRC patients with different GDF15 staining scores from a cohort of 42 CRC patients, Appendix Table S1 (H) and CRC tumor tissue microarray ($n = 94$), Appendix Table S3 (I). Data in (H) were analyzed using two-tailed log-rank test; data in (B, G) were analyzed using two-tailed, two-sample unpaired Student's *t* test. All values were expressed as mean ± SEM. *$P < 0.05$, ****$P < 0.0001$, vs. Control. Source data are available online for this figure.

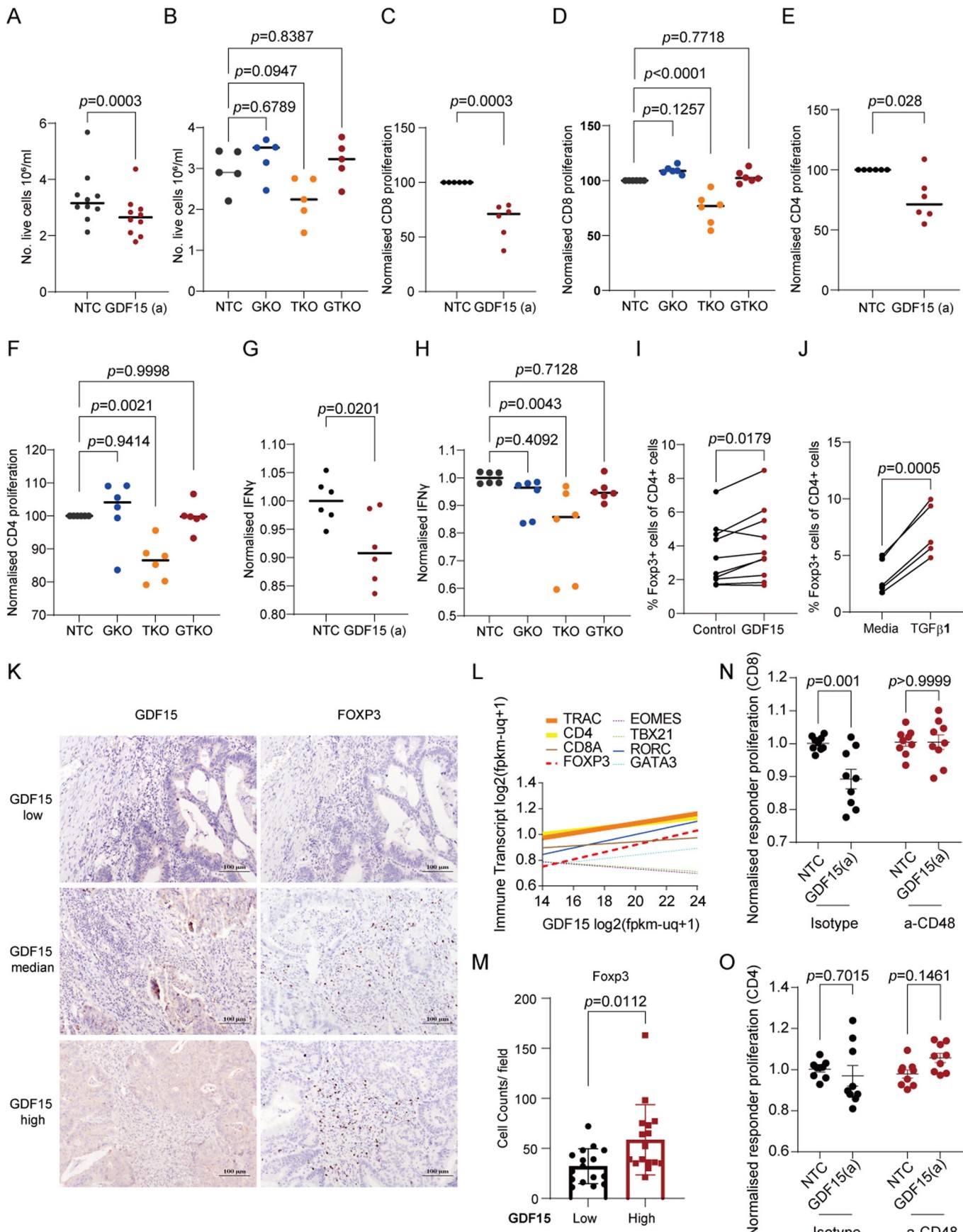

**Figure 5. GDF15 induces Tregs in a CD48-dependent manner.**

(A, B) PBMCs were stimulated with anti-CD3 and anti-CD28 for 4 days in the presence of fresh supernatant from indicated cell lines (NTC,GKO,TKO,GTKO; GDF15a). Live cells were counted using trypan blue and a haemocytometer. $n = 10$ biological replicates (A) and $n = 5$ biological replicates (B). Paired $t$ test used for (A); two-way ANOVA used for (B). (C–F) Labeled PBMCs were stimulated with anti-CD3 and anti-CD28 for 4 days in the presence of fresh supernatant from indicated cell lines, before being stained with anti-CD3 and anti-CD8 (C, D) or anti-CD4 (E, F) antibodies and measured by flow cytometry. Normalized proliferation (normalized to MFI from control: i.e., cells treated with supernatant from NTC cells) on gated $CD3^+CD8^+$ or $CD3^+CD4^+$ cells is shown. $n = 6$ biological replicates. Paired $t$ test used for (C, E); two-way ANOVA used for (D, F). (G, H) Normalized IFNγ concentrations (normalized to MFI from control: i.e., cells treated with supernatant from NTC cells) in the supernatant of cells from (C–F). Paired $t$ test used for (G); two-way ANOVA used for (H). $n = 6$ biological replicates. (I, J) PBMCs were stimulated with anti-CD3 and anti-CD28 for 4 days in the presence of fresh supernatant from NTC or *GDF15*a cell lines (I) or media alone or 5 ng/ml recombinant human TGFβ1 (J). Cells were stained with anti-CD3, anti-CD4 antibodies extracellularly before intranuclear staining of Foxp3 was performed. % of $CD4^+Foxp3^+$ cells are shown. Paired $t$ test used. $n = 10$ biological replicates (I) and $n = 5$ biological replicates (J). (K) Immunohistochemistry using anti-GDF15 and anti-Foxp3 antibodies on serial sections from colorectal cancer cases (Appendix Table S1). $n = 42$. (L) Correlations of indicated immune transcripts (normalized for *PTPRC* [CD45] expression) and *GDF15* transcripts from TCGA COAD dataset. Thick line indicates $R^2$ value > 0.1 and dashed line indicates transcription factor. (M) Pooled data from (K) showing $Foxp3^+$ cell counts in $GDF15^{low}$ and $GDF15^{high}$ populations; median split. $n = 42$ cases. Unpaired $t$ test. (N, O) Isolated naive CD4 cells were stimulated with anti-CD3 and anti-CD28 for 4 days in the presence of indicated cell line supernatant and either isotype control (10 μg/ml) or anti-CD48 (10 μg/ml) as indicated. These cells were then co-cultured with anti-CD3 stimulated proliferation dye-labeled responder PBMCs for 4 days, before cells were stained for CD3, CD8, and CD4. Normalized proliferation dye (MFI) of the indicated responder population is shown. Paired $t$ test. $n = 9$ biological triplicate, technical triplicate. All values were expressed as mean ± SEM. $*P < 0.05$, $**P < 0.01$, $***P < 0.001$, $****P < 0.0001$, vs. Control. Source data are available online for this figure.

(Wallin et al, 2011b; Jobin et al, 2017; Betts et al, 2012) (GDF15 being additionally associated with recurrence (Wallin et al, 2011a)), and with the majority of CRC patients being treated with oxaliplatin, we next considered whether the TXNIP/GDF15 axis, an axis which should regulate these processes to the benefit of the patient, remained functional in metastatic disease. In the course of this project we had observed a clear distinction in the TXNIP/GDF15 response to oxaliplatin when looking at cell lines derived from primary and secondary sites (Fig. 6A,B). This difference can be seen most clearly when assessing the ratio of GDF15 to TXNIP (GDF15/TXNIP) pre-treatment (Fig. 6C). We next assessed if there was a difference in the correlation between TXNIP and GDF15 in metastatic and primary disease, finding the significant inverse relationship in primaries discussed earlier was lost in metastatic samples (Fig. 6D,E). As resistance to chemotherapy is commonly observed in patients with metastatic disease, we developed two oxaliplatin-resistant lines, finding that they also lost oxaliplatin-induced TXNIP/GDF15 responsiveness (Fig. 6F,G; Appendix Fig. S7A–C), with GDF15/TXNIP ratios strongly resembling those of the cell lines derived from different sites (Fig. 6H).

We next considered whether this oxaliplatin resistance-associated loss of TXNIP/GDF15 responsiveness could be observed in progressive primary tumors. We first assessed TXNIP–GDF15 correlations in primary samples where chemotherapeutic response was known (nonresponder vs responder), finding the inverse 'functional' relationship was only present in responders (Fig. 6I,J). We then assessed our pre-treatment and posttreatment fresh tumor samples finding similar ratios to those observed in the cell line models when splitting the cohort into aggressive ( ≥ T4M1) and non-aggressive ( < T4M1) disease (Fig. 6K). These data collectively suggest that the loss of the responsive TXNIP/GDF15 axis (oxaliplatin-inducing ROS, driving TXNIP upregulation via MondoA, leading to a decrease in GDF15 secretion) is associated with both disease progression and chemotherapeutic resistance.

We then questioned whether or not the pre-treatment ratio of GDF15/TXNIP could be used as a potential biomarker of oxaliplatin treatment responsiveness. To test this hypothesis we first assessed whether or not the ratio could be used to differentiate cell lines from primary or secondary sites (Appendix Fig. S7E) or oxaliplatin-resistant lines from non-resistant (Appendix Fig. S7F)

or aggressive from non-aggressive tumors (Appendix Fig. S7G) as controls. We then tested this ratio using a publicly available dataset finding that pre-oxaliplatin treatment GDF15/TXNIP ratio could be used to determine treatment response (Fig. 6L). To check if the pre-treatment GDF15/TXNIP ratio could be used for patients treated with FOLFIRI we performed the same analyzes finding no significance (Fig. EV4A–D). This oxaliplatin specificity was then confirmed by western blot analysis in DLD1 and HCT15 cells treated with 5-FU or SN38 (Fig. EV4E,F).

Finally, as the data clearly showed a differential in ratio change between pre and posttreated "aggressive" and "non-aggressive" groups (definitions in the appropriate legend), we tested a new metric, posttreatment GDF15/TXNIP ratio divided by pre-treatment GDF15/TXNIP ratio (Appendix Fig. S7H–J), to see if this would improve the overall differential. We found that by adopting the new metric not only did the combined differential increase (Mean of 6.10 vs 1.66 [fold change of 3.7] for single pre-treatment GDF15/TXNIP ratio against 0.18 vs 1.30 [fold change of 7.2] for the combined) but so did the significance (0.0017–0.0011) (Fig. 6M,N). Given that there are no publicly available datasets pre and post oxaliplatin treatment, we used organoids derived from primary tumors to test this new metric by measuring GDF15 and TXNIP pre and post oxaliplatin treatment (Appendix Fig. S7D). Splitting the organoid groups into those with extra-mural invasion (considered more aggressive) and those without (less aggressive), we could see a significant difference between the groups (Fig. 6O).

## Discussion

Colorectal cancer is the third most common cancer worldwide, with 1.9 million cases reported in 2020. Five-year survival ranges greatly, from 13 to 88%, depending on stage at presentation, age, and sex (Siegel et al, 2020). Chemotherapy, predominantly oxaliplatin-based, is the most common first-line therapy and has been increasingly shown to be capable of turning a "cold tumor" with low active immune infiltrate into a "hot tumor" with improved infiltration. This conversion lays the foundation for current combinational chemo-immunotherapies, however, beyond innate stimulation through disease associated molecular patterns

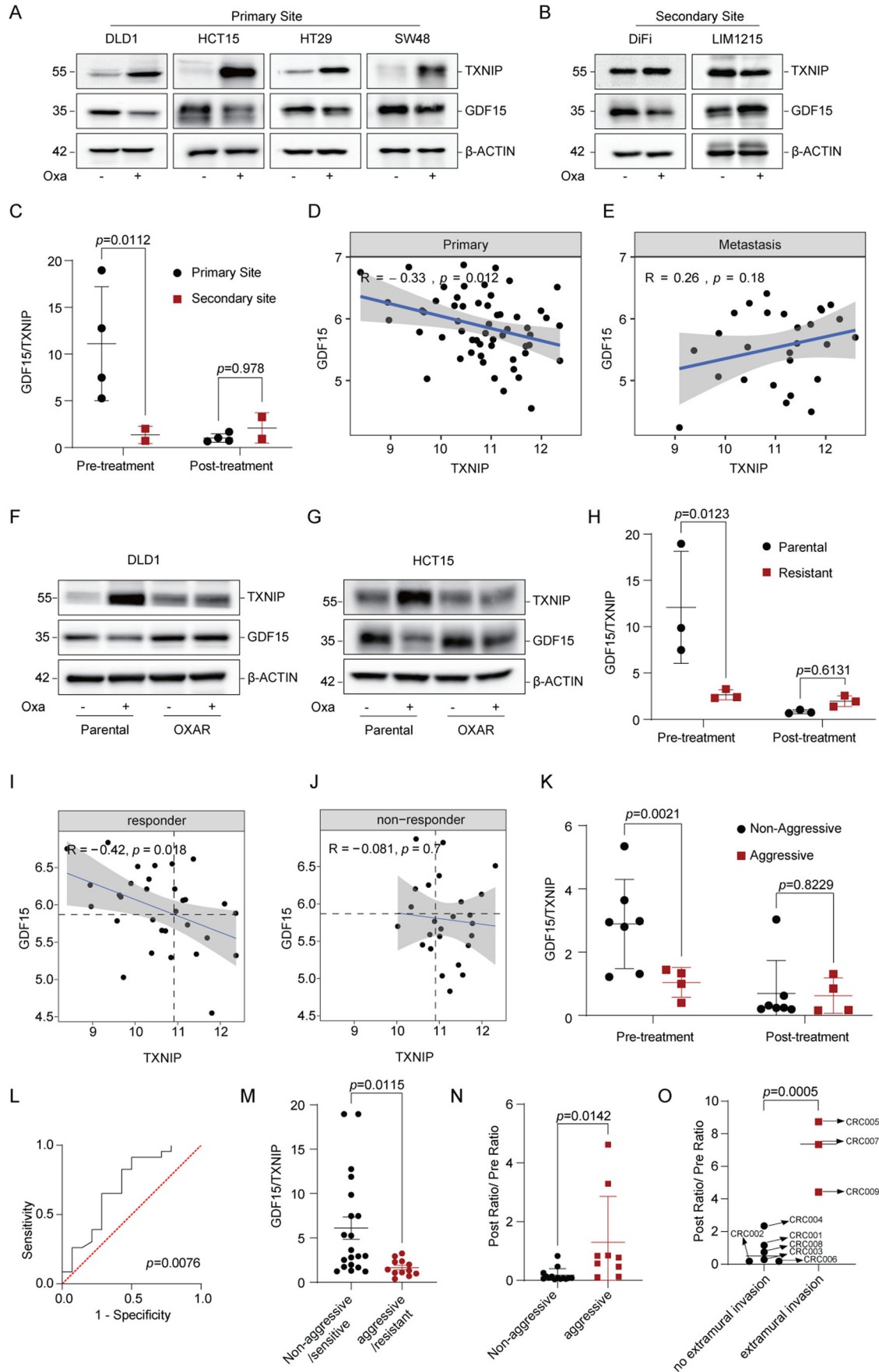

◀ **Figure 6. Loss of a oxaliplatin responsive TXNIP/GDF15 axis is associated with advanced disease and oxaliplatin resistance, with GDF15/TXNIP ratios showing potential as biomarkers of treatment response.**

(A, B) Immunoblot analysis of TXNIP and GDF15 expression after 48 h of 10 μm oxaliplatin treatment in colorectal cancer cell lines, including DLD1, HCT15, HT29, SW48 (A, derived from primary site), and DiFi, LIM1215 (B, derived from secondary site). (C) The ratio of GDF15/TXNIP for cell lines in A–B treated as indicated and measured using densitometry. $n = 4$ for primary, $n = 2$ for secondary. Standard deviation shown. (D, E) Microarray data showing the correlation between *GDF15* and *TXNIP* mRNA expression in primary (D) or metastatic (E) CRC tumors. R and P values shown (Pearson's). (F, G) Immunoblot analysis of TXNIP and GDF15 expression after 48 h of 10 μm oxaliplatin treatment in oxaliplatin-resistant (OXAR) cells: DLD1-OXAR (F) and HCT15-OXAR (G). (H) Ratio of GDF15/TXNIP for cell lines in (F, G) treated as indicated and measured using densitometry. $n = 3$ (2 biological replicates [2 lines] and 1 technical replicate). Standard deviation shown. (I, J) Microarray data showing the correlation between *GDF15* and *TXNIP* mRNA expression in primary tumors that respond (responder; I) or do not respond (nonresponder; J) to FOLFOX chemotherapy. R and P values shown (Pearson's). (K) Ratio of GDF15/TXNIP for primary tumors in Figs. 1D and 4F treated as indicated as measured using densitometry. Aggressive ($\geq$ T4M1) $n = 4$, non-aggressive ($<$ T4M1) $n = 7$ (Appendix Table S2). (L) Receiver operating characteristic (ROC) curve showing area under the curve and P values for the use of pre-treatment *GDF15/TXNIP* ratio in predicting responsiveness to oxaliplatin (O; responder [$n = 23$] and nonresponder [$n = 14$]) using publicly available data. Wilson/Brown test. (M) Pooled pre-treatment data (ratio of GDF15/TXNIP) from (C, H, K) with "aggressive" classed as secondary site, resistant to oxaliplatin and aggressive ($\geq$ T4M1) and 'non-aggressive' primary site, sensitive to oxaliplatin and non-aggressive ($<$ T4M1). Aggressive $n = 9$, non-aggressive $n = 14$. Standard deviation shown. (N) Posttreatment GDF15/TXNIP ratio divided by pre-treatment GDF15/TXNIP ratio for (C, H, K). "Aggressive" and "Non-aggressive" defined as in (M). Aggressive $n = 9$, non-aggressive $n = 14$. Standard deviation shown. (O) Posttreatment GDF15/TXNIP ratio divided by pre-treatment GDF15/TXNIP ratio for patient-derived organoids grouped into primary tumors with and without extra-mural invasion. With extra-mural invasion $n = 3$, without extra-mural invasion $n = 6$ (Table EV1). *$P < 0.05$ using Sidak's multiple comparisons test (C, H, K) **$P < 0.01$, ***$P < 0.001$ using Mann–Whitney (M, N) or unpaired t test (O). Western results shown (A, B, F, G) are representative of three independent experiments. Source data are available online for this figure.

(DAMPs) and the presentation of neoantigens, our understanding into exactly how the immune system, especially the adaptive arm, is "reawakened" is limited.

Although tumor suppressor genes (TSGs) are well known to function by targeting oncoproteins for degradation or inducing cell death per se., we have sought to understand the role of one particular TSG, TXNIP, in mediating chemotherapy-induced immunogenicity. Our interest in TXNIP stemmed from its reported role in regulating epithelial oxidative stress and our observation of its increased expression in fresh tumor samples after oxaliplatin treatment. By taking this observation, interrogating it in vitro, and investigating TXNIP's role in regulating the TME, these data have revealed a previously unreported epithelial-immune axis, namely ROS-MondoA-TXNIP–GDF15-Treg. (Fig. 7, Schematic diagram).

The balance of reductive and oxidative processes is crucial for cellular life. Dysregulation can promote oxidative stress which contributes to diverse pathologies, including neurodegenerative disorders, autoimmune diseases and cancers. Intracellular ROS in tumor cells has been observed to increase upon chemo- and radiotherapy, leading to apoptosis (Perillo et al, 2020). In addition, ROS levels in innate or adaptive immune cells are broadly associated with activation and antitumor effects (Y. Yang et al, 2020); (Kalafati et al, 2020; Scharping et al, 2021). A recent study by Gao et al identified that the ROS induced by chemotherapy increased the secretion of HMGB1 to facilitate the infiltration of T cells (Q. Gao et al, 2019), highlighting the importance of ROS in mediating cancer-immune cross talk. In this study, we found oxaliplatin-induced ROS generation could activate MondoA which, in turn, induced TXNIP expression. Furthermore, combining mass spectrometry, proteomic array and genetically modified models (CRISPR-KO and CRISPR-activation), before verifying in situ, we revealed that the ROS/MondoA/TXNIP axis negatively regulated GDF15 expression and secretion. It must be stressed that these data do not place TXNIP as the sole regulator of GDF15, for example ARRDC4 can also be seen to regulate GDF15. We envisage TXNIP as one of a number of ROS-dependent GDF15 regulators, with this redundancy potential evidence of the importance of this regulatory framework.

Further support for both TXNIP and ARRDC4's role in regulating GDF15 after the induction of ROS comes from a pan cancer meta-analysis assessing the impact of metformin (which has been reported to inhibit ROS) on gene expression. Here the top two downregulated genes were *TXNIP* and *ARRDC4* and the top four upregulated genes were *DDIT4, CHD2, ERN1*, and *GDF15* (Schulten and Bakhashab, 2019).

It is important to remember at this point that one of TXNIP's key roles is to prevent ROS accumulation by preventing uptake of glucose by inhibiting GLUT receptors directly. It does this in response to MondoA's activation to glycolysis. This 'cutting off at source' approach should not be overlooked as manipulating the glucose environment, and therefore these processes, as seen in EV2 could potentially affect the potency of oxaliplatin in CRC.

The main shortcoming of this paper is the lack of mechanistic understanding linking TXNIP to GDF15. There are 650 transcription factors that have been shown, or are predicted, to bind to GDF15 promoter and/or enhancer regions. By assessing our list of differentially expressed genes (Dataset EV1) for the presence of these factors we identified 6 GDF15 binding TFs that show significantly decreased expression after oxaliplatin treatment in both cell lines (*ATF4, MYC, SREBF1, PHB2, HBP1, KLF9*). There was only one, *MYC*, that was downregulated by oxaliplatin treatment (validated; Fig. EV5A), and with this downregulation partially being rescued in a matched *TXNIP* knockout line (Fig. EV5B). We then observed that c-myc has been shown or is predicted to bind to promoter/enhancer regions of the top five transcriptomic and proteomic differentials in *TXNIP* knockout lines, including TXNIP itself (apart from C16orf90). Even with c-myc's promiscuity (binds to 10–20% of all promoters/enhancers) this may be suggestive of a specific relationship. Finally, when looking at the correlations between these 6 TFs and *TXNIP* and *GDF15* in the TCGA COAD dataset, *MYC* has the greatest and most significant negative correlation to *TXNIP* ($r = -0.4631$, $P = 1.42e-28$) and the greatest and most significant positive correlation to *GDF15* ($r = 0.4653$, $P = 7.32e-29$). *ATF4* and *PHB2* are the other TFs in the list, that show the same significant trends (Fig. EV5C), and therefore may play a role in the TXNIP-independent oxaliplatin-dependent regulation of GDF15. Further

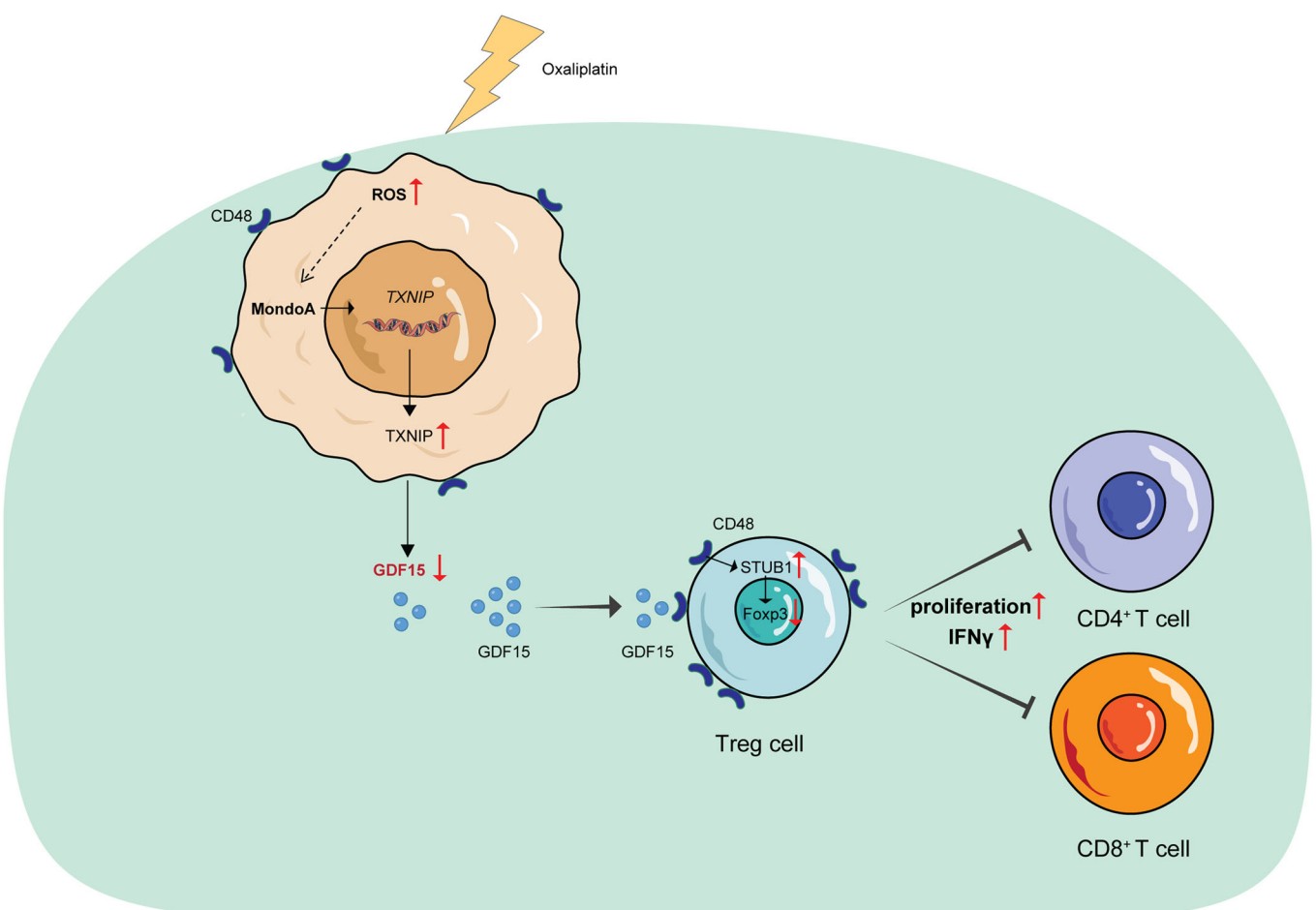

**Figure 7.** Schematic diagram of the underlying mechanism of oxaliplatin-induced immunogenicity by regulating MondoA/TXNIP/GDF15 signaling pathway in CRC.

exploration of these additional TFs is outside the scope of the current manuscript.

MYC's role in bridging from TXNIP to GDF15 is further supported by a recent paper which shows that TXNIP is "a broad repressor of MYC genomic binding" and that "*TXNIP* loss mimics MYC overexpression" (Lim et al, 2023). Furthermore, the interdependent regulatory relationship between MondoA, TXNIP, and MYC has been seen in a variety of models (BR and DE, 2015), whilst the impact of NAC on MYC-dependent pathways has been seen in lymphoma (Yao et al, 2023). These studies lend credence to the idea that MYC is the most likely TXNIP-regulated TF that regulates GDF15 in our systems.

GDF15 has previously been shown to promote "M2" macrophage differentiation, inhibit NK cell function and dendritic cell maturation (Kleinertz et al, 2019; Z. Zhou et al, 2013), however as described, the purified recombinant tools used in these studies have been shown to be contaminated with active TGF-β, raising justifiable concerns (as all these effects can be ascribed to this pleotropic cytokine) (Olsen et al, 2017; Rochette et al, 2020). In this study, to avoid this issue, we prioritized the use of cellular systems for our immunological assays. A recent study, which used mass spectrometry to confirm the material they used was not contaminated with TGF-β, found that recombinant GDF15 was

able to induce and maintain Tregs via interaction with CD48 on naive T cells (Wang et al, 2021). Our findings support this concept, further adding tissue validation (the association of high GDF15 and FOXP3/Foxp3) and the potential of preventing this process using CD48 blockade. Given these data and the well-reported negative prognostic impacts of Tregs in tumors, including in CRC, and the positive impact of chemotherapy, we put forward the following model. (1) Chemotherapy either promotes cell death or induces oxidative stress and ROS formation in the cells that survive. (2) The cells that survive do so by increasing TXNIP expression to help alleviate the impact of chemotherapy-induced ROS (or naturally carry a high level of TXNIP and are selected for). (3) This high level of TXNIP inhibits GDF15 expression which consequently inhibits the local generation of Tregs from naive CD4 cells. (4) This decrease in Tregs allows other T cells, especially CD8s, to function and help to eradicate the remaining tumor, facilitating a durable response.

One of the most intriguing aspects of this work is the impact of the post-chemotherapeutic change (TXNIP^lowGDF15^high to TXNIP^highGDF15^low), and the lack of change, on outcome. *TXNIP* is a known TSG and, as such, we, along with many others, show increased expression is associated with better prognosis, whilst the inverse is true for GDF15 (leading to the ongoing development of

targeting drugs) (Wallin et al, 2011b). These data suggest that the post-chemotherapeutic change, something validated in primary CRC cell lines, spheroids, PDTOs and, critically, patients themselves, is associated with positive outcome. The lack of responsiveness seen in cell lines derived from secondary sites, resistant models and fresh tumors taken from patients with more advanced disease, suggests that this axis is "broken" in these contexts. These data are supported by publicly available transcriptomic data showing that the negative correlation, indicative of response, is not seen in either primaries that do not respond to chemotherapy or in metastases. As such, these collective data suggest that there is a subgroup of patients who intrinsically carry, or develop, a lack of responsiveness, raising the possibility of using biopsies as a stratification tool. Indeed we were able to demonstrate that the pre-treatment GDF15/TXNIP ratio was able to predict tumors that were responsive to oxaliplatin from those that were not.

Aware of the fact that the change in GDF15/TXNIP ratio pre and post treatment would likely give a better differential between aggressive and non-aggressive groups, and aware of the fact that pre and posttreatment biopsies are often difficult to control and justify clinically, we combined these ratios and tested this new metric using organoids. Using this technique and this new parameter/metric (change in GDF15/TXNIP ratio pre and post treatment) we were able to demonstrate that organoids have potential as sentinels of oxaliplatin responsiveness and disease progression.

With this knowledge it may well then be possible to predict oxaliplatin non-responders, using a single GDF15/TXNIP pre-treatment ratio (biopsy; transcript or protein), or a potentially more sensitive combined posttreatment/pre-treatment ratio (organoids; protein), and tailor treatment plans accordingly. Indeed this methodology is especially pertinent to the use of anti-GDF15 therapeutics, allowing their potential use early in disease. As such these data champion targeted, effective therapy through biological understanding and functional assessment.

# Methods

This study was conducted and reported with reference to the International Committee of Medical Journal Editors (ICMJE) recommendations and REporting recommendations for tumor MARker prognostic studies (REMARK) guidelines.

## Public dataset analysis

The cancer genome atlas (TCGA) was used to compare the differential expression of *TXNIP/GDF15* between adjacent normal samples and cancer patient samples. Gene expression data from TCGA was downloaded from either the GDC data portal (https://www.genome.gov/Funded-Programs-Projects/Cancer-Genome-Atlas) or UCSC Xena functional genomics explorer (https://www.xenabrowser.net). Both colon adenocarcinoma (COAD) and rectal adenocarcinoma (READ) cohorts were included as colorectal cancer cases. Four public datasets were used in this study for prognostic analyzes, including GSE29621, GSE38832, GSE6988, and GSE52735. These datasets were downloaded from the Gene Expression Omnibus (GEO, http://www.ncbi.nlm.nih.gov/geo). For the survival analysis, the continuous variables were dichotomized via the survminer R package, and

the Kaplan–Meier curves were performed using the survival R package. To measure *TXNIP* and *GDF15* expression in normal and tumor epithelial cells from paired samples at single-cell level, we used normalized scRNA-seq data from ten paired samples from colorectal cancer patients deposited in GSE132465. Microarray data from responder and nonresponder to FOLFOX and FOLFIRI therapy for primary and metastatic lesions was downloaded from GSE28702 and normalized using RMA and converted to the gene level using an appropriate average. ROC analysis for publicly available data was performed using rocplot.org (Fekete and Győrffy, 2023).

## Human samples

This study was performed in accordance with the principles of the Declaration of Helsinki and the experiments conformed to the principles set out in the Department of Health and Human Services Belmont Report, including informed consent. The collection of pre and posttreatment samples from CRC patients was approved by Peking University Third Hospital Medical Science Research Ethics Committee (Reference number IRB00006761-M2022237). Information on the human cohorts is provided in Appendix Tables S1 and S2 in the Appendix. Two cohorts, including 42 CRC tissues with tumor tissue and corresponding adjacent normal tissues (Appendix Table S1) and 11 CRC tissues with pre- and matched post-oxaliplatin-based chemotherapy (Appendix Table S2 were retrospectively collected from May 2014 to March 2021. A human colorectal cancer tissue microarray (TMA) purchased from Shanghai Outdo Biotech Company Ltd (Shanghai, China). All tissue samples were collected before chemotherapy treatment. The TMA contained 97 colorectal cancer samples and paired adjacent normal tissues collected from patients between 2009 and 2018 and were accompanied by patient clinical data. Patient information for the TMA is provided in Appendix Table S3. Organoids were grown from colorectal adenocarcinoma patient-derived material under REC 20/YH/0088, HTA license 12055. Leukocyte cones were ordered from the National Health Service Blood and Transplant Service (NHSBTS). The NHSBTS obtains informed consent from donors to provide materials to partners under the terms of their UK HTA (Human Tissue Authority) license. Partners must apply for access to the materials within the terms of this license.

## Immunohistochemical (IHC) staining

The tumor tissues excised during the operation were immediately placed in 10% formalin for fixation (X. Zhou et al, 2021). To begin with, FFPE slides were dewaxed and rehydrated. After antigen retrieval in 0.01 M sodium citrate buffer (pH 6.0) in a microwave for 20 min, slides were treated with peroxidase block for 5 min and protein block solution for another 5 min at RT. Then Slides were incubated with primary antibody against TXNIP (Abcam, ab188865; 1:250), GDF15 (Proteintech, 27455-1-AP; 1:500), and FOXP3 (Abcam, ab215206; 1:1000) overnight at 4 °C. Post primary antibody incubation, tissues were incubated with secondary antibodies (EnVision Chem Detection Kit, DaKo Cytomation) at room temperature for 30 min. Followed by incubation with horseradish enzyme-labeled streptavidin solution for 10 min and then stained with DAB and haematoxylin. The stained tissues were interpreted by two pathologists blinded to the clinical parameters. Staining percentage scores were defined as: expression

intensity × expression area. Expression intensity was scored from 0 to 3 ($10 \times 20$ magnification, five different random fields of view were selected), representing negative, weakly staining (light yellow), moderately staining (pale brown with light background), and strongly staining (dark brown without background), respectively. Expression area was scored from 0 to 4: 0 (1–5%), 2 (26–50%), 3 (51–75%) and 4 (> 75%). representing <5, 6–25, 26–50, 51–75, and, respectively. The degree of positive staining: 1–3 was classified as weakly positive (+); 4–6 as moderately positive (++); and 7–12 as strongly positive (+++). The intraclass correlation coefficient (ICC) analysis was used for assessing the level of agreement between independent reviewers. The ICC scores were 0.893, 0.912, and 0.905 for samples stained with anti-TXNIP, anti-GDF15 and anti-FOXP3 antibodies, respectively.

### scRNA-seq analysis

For comparing *GDF15* and *TXNIP* expression in colorectal cancer tumor samples, we used log transformed-normalized single-cell RNA sequencing data derived from 63 colorectal cancer patients (Joanito et al, 2022) deposited at the Synapse (syn26844071) and extracted only tumor cells.

### Western blot

Cells were seeded into six-well plates ($4 \times 10^5$ cells per well). The following day cells were replaced with fresh media for 1 h and then treated as indicated in the Figures. Cell fractionation was performed with NE-PER™ Nuclear and Cytoplasmic Extraction Reagent (Thermo Fisher Scientific, 78833), buffers were added with protease and phosphatase inhibitors. Following two washes with PBS, cells were lysed in 150–200 µl 1 × sample lysis buffer (5 × sample lysis buffer: 2.5 ml 1 M Tris pH 6.8, 1 g SDS, 5 ml glycerol). Cell lysates were measured using the BCA assay (Pierce™ BCA Protein Assay Kit, 23227) and run on SDS–PAGE with 30 µg protein loaded. After blocking in 5% milk or 5% bovine serum albumin (BSA) in tris-buffered saline and Tween-20 (TBS-T) for 2 h at room temperature. Antibodies against MondoA (Abcam [ab77294], 1:1000), IL-1β (1:1000, Cell Signaling Technology [D3U3E, 12703]), Caspase 1 (1:1000, Cell Signaling Technology [D7F10, 3866]), TXNIP (1:1000, Cell Signaling Technology [D5F3E, 14715]), Cas9 (1:1000, Santa Cruz [sc-517386]), GDF15 (1:1000, Abcam [ab206414]), Trx (1:1000, Cell Signaling Technology [C63C6, 2429]), C-myc (1:1000, Cell Signaling Technology [D84C12, 5605]), β-Actin (1:5000, Proteintech [6609-1-Ig]), GAPDH (1:5000, Proteintech [60004-1-Ig]) and Lamin A (1:1000, Cell Signaling Technology [133A2, 86846]) were used for incubation overnight at 4 °C.

### Cell lines and reagents

Human colon adenocarcinoma cell lines DLD1 (CCL_221) and SW48 (CCL_231) were purchased from ATCC. LIM1215 (CVCL_2574) was a generous gift from Dr. Sabine Teipar (University Leuven, Belgium). HT29 (CVCL_A8EZ), DiFi (CVCL_6895) and HCT15 (CVCL_0292) were generous gifts from Dr. Juan Jose Garcia Gomez (University College London). DLD1, HCT15, HT29, and LIM1215 were maintained at 37 °C with 5% $CO_2$ in RPMI supplemented with 10% fetal bovine serum (FBS), 1% penicillin/streptomycin (P/S), and L-glutamine (2 mM). DIFI and SW48 were grown at 37 °C with 5% $CO_2$ in DMEM supplemented

with 10% FBS, 1% P/S and L-glutamine (2 mM). All the CRC cell lines tested negative for mycoplasma throughout the study and had been authenticated (STR profiling) within the past 3 years.

### RNA sequencing

The RNA-Seq experiments were performed by Novogene (Cambridge, UK) Company Limited (Pan et al, 2020). Briefly, total RNA from CRC cells was isolated using TRIzol reagent. Messenger RNA was purified from total RNA using poly-T oligo-attached magnetic beads. After fragmentation, the first strand cDNA was synthesized using random hexamer primers, followed by the second strand cDNA synthesis. The library was ready after end repair, A-tailing, adapter ligation, size selection, amplification, and purification. For the data analysis, base calls were performed using CASAVA. Reads were aligned to the genome using the split read aligner TopHat (v2.0.7) and Bowtie2, using default parameters. HTSeq was used to estimate abundance.

### Transfection

For transient transfection, siRNA was transfected into different cell lines using Lipofectamine™ RNAiMAX Transfection Reagent (Thermo Fisher Scientific, 13778075). In total, $3 \times 10^5$ cells were seeded in six-well plates in an antibiotic-free complete medium. After 24 h, 5 µl of Lipofectamine™ RNAiMAX Transfection Reagent and 25 pM siRNA (All from Dharmacon. NTC D-001810-10-05, MLXIP L-008976-00-0005, ARRDC4 L-019366-02-0005) were mixed thoroughly and incubated for 20 min before added to the cells at room temperature. Knockdown efficiency was assessed by western blot and PCR analysis after 48 h.

### RNA isolation and reverse transcription quantitative real-time PCR (RT-qPCR)

Cells were lysed in 0.7 ml of TRIzol Lysis Reagent (Invitrogen, 15596026), vortexed, and incubated for 10 min at room temperature. RNA was extracted using the RNeasy Mini Kit (Qiagen, 74104) in the presence of RNase-free DNase (Qiagen, 79254). cDNA was synthesized by reverse transcription using a SuperScript™ II Reverse Transcriptase kit (Thermo Fisher Scientific, 18064022). RT-qPCR was performed with Power SYBR green PCR master mix (Applied Biosystems, 4309155). Primers used: TXNIP forward: GACCTGCCCCTGGTAATTGG, reverse: GGGAGGAGCTTCTGGGGTAT. GAPDH forward: CTCCTGTTCGACAGTCAGCC, reverse: CCCAATACGACCAAATCCGTTG. ARRDC4 forward: GCCAGCCAGTTCAGTATGGA, reverse: GCATAATTTGGTGGTGCTTCAGG. MLXIP forward: ACGGCTCTGTGGACGTAGA, reverse: GGCTCTTCCAGTACTTCCCTTC. Data analysis was conducted with the QuantStudio 6 Flex Real-Time PCR System. Relative mRNA levels were calculated with normalization to the housekeeping gene *GAPDH*. (NB. *GAPDH* did not change after chemotherapy treatment as assessed in the RNA-seq analysis).

### Chromatin immunoprecipitation-quantitative polymerase chain reaction (ChIP-PCR)

DLD1 cells were seeded in 75-cm² flasks (~ 40% confluency). Overnight, cells were replaced with fresh media for 1 h and then

either treated or not treated with oxaliplatin/ NAC as indicated in the Figures. After 48 h, cells were cross-linked with 1% formaldehyde and quenched by glycine. Chromatin extraction was performed using the Chromatin Extraction Kit (ab117152) followed by sonication. Equal amount of chromatin was incubated overnight at 4 °C with 2 µg of anti- MondoA (Proteintech, 13614-1-AP) or IgG (Cell Signaling Technology, 2729). ChIP pull-down assays were performed using the ChIP Kit Magnetic One-Step (ab 156907) according to the manufacturer's instructions. Recovered DNA was quantified by RT-qPCR using primers specific for *TXNIP* promoter region (forward: CACAGCGATCTCACTGATTG; reverse: GTTAGTTTCAAGCAGGAGGC) under the following conditions: 40 cycles of denaturation at 95 °C for 15 s and annealing at 56 °C for 20 s, followed by extension at 72 °C for 40 s. The specificity of the PCR product was assessed by Sanger sequencing.

## Spheroid formation assay

The spheroid culture was performed using suspensions of cells with at least 90% viability. The spheroid formation was performed with 1000 vital cells in 100 µl per well in a low-attached 96-well plate (Corning, 3474) under standard culture conditions. DLD1 spheroids were formed after 24 h of seeding. HCT15 spheroids were formed after 72 h of seeding. CellTiter-Glo® 3D cell viability reagents (Promega) were used to analyze spheroid viability as per the manufacturer's instructions. Three-dimensional cultures were treated with oxaliplatin and incubated for 48 h.

## Patient-derived tumor organoids (PDTOs)

University College Hospital London (UCLH) provided us with colonic tissues from colorectal cancer patients in accordance with the guidelines of the European Network of Research Ethics Committee (EUREC) following European, national, and local law. HTA licence: 12055, REC reference: 15/YH/0311 as overarching biobank ethical approval. Informed consent forms were signed by all the participants in the study. Patient consent can be withdrawn at any time, resulting in the prompt disposal of the tissue and any derived material.

CRC cells were isolated as described by Sato et al (Sato et al, 2011). Briefly, specimens were washed with 10 ml PBS and then cut into small pieces (1–2 mm) with 10 ml of digestion buffer (Table EV2). Tissue and digestion buffer were transferred to a gentleMACS C Tube (run protocol 37C_h_TDK_1) (Miltenyi Biotec, 130-096-334) and incubated at 37 °C for 1 h. The supernatant was aspirated after samples were filtered through 100 µm strainers (732-2759) into a 50-ml tube, and centrifuged at $800 \times g$ for 2 min. After incubating with ACK lysis buffer (A1049201) at room temperature for 5 min, samples were washed with PBS twice. Cell pellet was resuspended in appropriate volume of Matrigel and 40 µL organoid: Matrigel droplets were plated into a six-well plate.

After incubation at 37 °C for 10–20 min, 2 ml of complete medium (Table EV2) supplemented with the ROCK Inhibitor Y-27632 (10 µM, 72302) were added in each well. Medium was changed twice a week until ready for passage. For RT-qPCR and western blot analyzes, organoids were seeded in six-well plate and collected after drug treatments indicated.

## ROS measurement

ROS level in cells was detected using DHE (Dihydroethidium) Assay Kit–Reactive Oxygen Species (Abcam, ab236206). Around $1 \times 10^5$ cells were added to V-bottom plate. In total, 130 µL ROS staining buffer and then 100 µL Cell-Based Assay Buffer were used according to the manufacturers' guides. The fluorescence was measured using an excitation wavelength between 480 and 520 nm and an emission wavelength between 570 and 600 nm.

## 2-NBDG staining and flow cytometry for glucose uptake

In all, $6 \times 10^5$ cells were seeded into 6-cm dishes and then treated with 0, 5, 10, 25, 25, 50 µM oxaliplatin for 48 h. Medium was removed and fresh medium containing 100 µM 2-NBDG (Thermo Fisher Scientific, N13195), a fluorescently-labeled deoxyglucose analog, was added. Cells were incubated at 37 °C for 2 h, before immediately being analyzed by flow cytometry (FITC channel).

## CRISPR-CAS9 genome engineering

*MLXIP* (MondoA), *TXNIP* and *GDF15* knockouts in cells and organoids were carried using the CRISPR/Cas9 system and the Edit-R CRISPR/Cas9 gene engineering protocol (Horizon). Guide RNAs for *TXNIP* (Edit-R CRISPR (knockout) Human *TXNIP* crRNA, Catalog ID:CM-010814-01-0002), *GDF15* (Edit-R CRISPR (knockout) Human *GDF15* crRNA, Catalog ID:CM-019875-01-0002), and *MLXIP* (MondoA) (Edit-R CRISPR (knockout) Human MLXIP crRNA, Catalog ID:CM-008976-01-0002) were purchased from Horizon.

Cells were transfected in a six-well plate with crRNA: tracrRNA transfection complex and Cas9 mRNA, using DharmaFECT Duo Transfection Reagent (Horizon, T-2010-02) (Table EV3). After 48 h, a BD Aria Fusion cell sorter was used to sort GFP-positive single cells into 96-well plates. To measure TXNIP and GDF15 levels, each clone was expanded for 3–6 weeks. The following knockout clones were chosen: three *TXNIP* knockout clones, three MLXIP-knockout clones, and four *GDF15* knockout clones. A heterogenous knockout cell line was generated by combining knockout clones of each gene and their functional evaluation was performed. Stabilities of the knockouts were checked every five passages using RT-qPCR and western blot analysis.

The neon® Transfection System (Thermo Fisher Scientific, MPK5000) was used for CRISPR Editing of organoids. In total, $1 \times 10^5$ organoids were trypsinized and single cells were resuspended in 7.5 µL of Resuspension Buffer R per electroporation condition, then 7.5 µL of RNP Complex Mix was added (Table EV3). The mixture was electroplated as shown in Table EV3. Immediately after electroporation, organoids were seeded onto a 24-well prewarmed plate. The complete medium was changed every 2 days, and genome editing efficiency was assessed using RT-qPCR and Western blot analysis.

## Mass spectrometry

DLD1 cells were seeded with a density around 70–80% in six-well plates. On the second day, cells were washed with PBS and replaced with 2 ml of FBS-free media (RPMI + 1% penicillin/ streptomycin +1% Glut.). After 48 h (day 4), the supernatants

from the cell culture were collected, centrifuged (300 g/5 min) to remove debris, followed by adding cold acetone at a ratio of 1:3. The mix was shaken thoroughly and stored at $-20\,°C$ overnight. Protein pellets were collected after centrifugation at $10,000 \times g$ for 15 min). The pellets were then stored at $-80\,°C$. Each protein pellet was resuspended in 20 μl of 8 M urea, followed by adding NuPAGE™ LDS Sample Buffer (4×) (Thermo Fisher). The mixture was heated to $90\,°C$ for 5 min and loaded into a 10% Bis-Tris gel, resolved for about 1 cm (80 volts; 63 mA; 8 watts) before being stained with Imperial protein stain (Thermo Fisher). After de-staining to remove the background, the whole section was excised and followed by an in-gel trypsin digestion overnight at $37\,°C$. In all, 500 μg of TMTpro reagents (Thermo Fisher) were added to the peptides (50 μg) along with acetonitrile and then incubated at room temperature for 1 h. After the labeling efficiency was checked, the reaction was quenched with hydroxylamine to a final concentration of 0.3% (v/v) for 15 min and all individual tags were combined as one. The sample was vacuum centrifuged to near dryness and subjected to C18 solid-phase extraction (SPE, Sep-Pak) for a clean-up.

MS data were collected using Orbitrap Fusion Lumos mass spectrometers. Orbitrap Fusion Lumos mass spectrometer was equipped with an Ultimate 3000 RSLC nano pump. Raw mass spectrometry data were processed into a peak list file with Proteome Discoverer (ThermoScientific v2.5). Processed data were then searched using Sequest search engine embedded in Proteome Discoverer v2.5 against the reviewed Swissprot Homo Sapiens database downloaded from Uniprot (http://www.uniprot.org/uniprot/). The mass spectrometry proteomics data have been deposited to the ProteomeXchange Consortium via the PRIDE partner repository with the dataset identifier PXD051666.

## Proteome profiler antibody array

Human (R&D Systems, ARY005B) cytokine arrays were used. Cells were seeded at $4 \times 10^5$/well in a six-well plate. The next day, the cells were replaced with flesh media with or without indicated drug. Tumor-conditioned medium (TCM) was collected after 48 h of treatment. In all, 0.5 ml of TCM was added to membrane, and soluble Proteome was analyzed following the manufacturer's instructions.

## Enzyme-linked immunosorbent assay (ELISA)

ELISAs for GDF15 (DGD150), IL-1β (DY201-05), and IFNγ (DY285B-05) were purchased from Biotechne and carried out as per the manufacturer's instructions. Plates were read on a CLARIOstar instrument at 450 nm, being corrected against 570 nm, and analyzed using MARS software and excel. The concentration of each sample was calculated using a standard curve.

## Immunofluorescence staining

In total, $5 \times 10^3$ DLD1 cells were plated into 35-mm glass bottom dishes. After 24 h, cells were treated with 10 μM oxaliplatin. 48 h post treatment, cells were rinsed with PBS, fixed for 20 min with 4% PFA, rinsed with PBS, permeabilized 10 min with 0.1% Triton-X100, rinsed with TBS-T. Subsequent labeling, imaging, and image analysis steps were as previously described (Vicencio et al, 2022).

## Generation of CRISPRa constructs

### dCas9-VPR

The 10XUAS-dCas9-VPR constructs have been previously described (Lin et al, 2015). Instructions are available at Addgene (https://www.addgene.org/78897/).

## Transfection of stable dCas9-VPR expressing cell lines with synthetic guide RNAs

Cells were seeded in six-well plates and cultured >50% confluency. Culture media was replaced with 1.6 ml of fresh media before transfection. Transfection reagents were prepared in two separated tubes (A and B): Tube A (195 μl serum/antibiotic-free media and 5 μl 10 μM guide RNA mix) and Tube B (195 μl Serum/antibiotic-free media and 5 μl DharmaFECT reagent 1). Tubes A and B were mixed thoroughly and incubated at room temperature for 20 min before being added to the cells. Sequences of crRNA oligonucleotides are as follows: CRISPRmod CRISPRa (activation) Human *MLXIP* Synthetic crRNA (Horizon, P-008976-01-0005), CRISPRmod CRISPRa (activation) Human *TXNIP* Synthetic crRNA (Horizon, P-010814-01-0005), CRISPRmod CRISPRa (activation) Human *GDF15* Synthetic crRNA (Horizon, P-019875-01-0005), CRISPRmod CRISPRa synthetic crRNA non-targeting controls (Horizon, U-009500-10-05).

## Immune cell isolation

Leukocyte cones were ordered from the National Health Service Blood and Transplant Service (NHSBTS) (The NHSBTS obtains informed consent from the donors and has internal ethical approval under the terms of their own HTA licence). Cells were mixed 1:1 with phosphate-buffered saline (PBS) and layered on Ficoll–Paque (GE Healthcare; 1714402). Cells were spun at $800 \times g$ for 30 min, with the brake off, and the PBMCs were taken from the buffy layer above the Ficoll–Paque. Naive CD4 T cells were isolated from PBMCs using the MACS system as per the manufacturer's instructions (Miltenyi Biotech; 130-094-131. LS Columns; 130-042-401). Purity was checked using anti-CD4 and anti-CD45RA antibodies and seen to be > 95% (using the manufacturer's recommended concentrations; 5μl per test). If purity was below 95%, the cells were disposed of.

## Flow cytometry

In all, $1–2 \times 10^5$ cells were stained with a live/dead dye (Thermo Fisher; L23102) in PBS for 10 min on ice in the dark, before being washed twice in FACS buffer (0.5% bovine serum albumin (Sigma; 05482) in PBS + 2 mM EDTA). Cells were then Fc blocked with Trustain (Biolegend; 422302) in FACS buffer for 10 min on ice in the dark. Cells were washed and then stained using a variety of antibodies ± secondary reagents described below, using concentrations recommended by the manufacturer (5 μl per test), on ice for 30 min in the dark. Cells were washed and either read immediately or fixed using 1% PFA in FACS buffer and read within 3 days. Cells were read using a BD Accuri C6 Plus flow cytometer, with analysis carried out using BD Accuri C6 Plus software. All cells were gated as follows: (a) Forward scatter and side scatter (SSC) to exclude cellular debris (while also adjusting threshold), (b) live/dead (only live cells carried forward) and (c) SSC-A vs. SSC-H—only singlets

carried forward. All MFIs were corrected against an appropriate isotype control. Intracellular flow cytometry was carried out using the intracellular fixation and permeabilization kit (eBioscience; 88-8824-00) according to the manufacturer's instructions. Antibodies and reagents used for flow cytometry: LIVE/DEAD™ Fixable Red Dead Cell Stain Kit (Thermo, L23102), PE Mouse IgG1, κ Isotype Ctrl Antibody (Biolegend, 400112), FITC Mouse IgG1, κ Isotype Ctrl Antibody (Biolegend, 400110), PerCP Mouse IgG1, κ Isotype Ctrl Antibody (Biolegend, 400148), FITC anti-human CD48 Antibody (Biolegend, 336706), PerCP anti-human CD4 Antibody (Biolegend, 317432), FITC anti-human CD3 Antibody (Biolegend, 317306), PE anti-human CD8 Antibody (Biolegend, 344706), PE anti-human FOXP3 Antibody (Biolegend, 320108), PE anti-human CD45RA Antibody (Biolegend, 304108), FOXP3 Fix/Perm Buffer Set (Biolegend, 421403).

## Proliferation assays

In all, 96-well tissue culture stimulation plates were prepared the night before by adding 100 μl/well 1 μg/ml anti-CD3 (OKT3) in PBS. PBMCs were stained using an eFluor™ 670 dye (65-0840-85; eBioscience) according to the manufacturer's instructions and plated at $2 \times 10^5$ cells in 100 μl. Overall, 100 μl of supernatant or other factors were added and cells were cultured for 4 days.

## Functional Treg assay

Anti-CD3 (OKT3) was plated at 1 μg/ml in PBS and incubated overnight at 4 °C. Supernatant was removed and $2 \times 10^5$/cell isolated naive CD4 cells were added in the presence of 1 μg/ml anti-CD28 in the presence of NTC or GDF15 (a) supernatant +/− isotype control (10 μg/ml) or anti-CD48 (10 μg/ml). Cells were cultured at 37 °C for 4 days. On day 3, anti-CD3 was plated at 1 μg/ml in PBS and incubated overnight at 4 °C. Allogeneic PBMCs were isolated, stained with eFluor™ 670 proliferation dye, and plated at $1 \times 10^5$ cells/well. Overall, $1 \times 10^5$ Tregs were added at a 1:1 ratio and the co-culture was run for 4 days. Cells were then harvested and stained with anti-CD3, anti-CD8, and anti-CD4 antibodies. The proliferation dye MFI in the responder population was normalized against matched cells stimulated in media alone.

## Establishment of oxaliplatin-resistant (OXAR) cell lines

Oxaliplatin-resistant cells (OXAR) cells were established by treatment with constant oxaliplatin concentration in vitro. Different oxaliplatin concentrations (50 μM for DLD1 and 25 μM for HCT15) were added to RPMI complete media. DLD1 and HCT15 cells were sub-cultured every 2 weeks. Finally, cell lines that are capable of growing exponentially in RPMI with high concentrations of oxaliplatin were identified as drug-resistant cell lines. The final tolerated drug concentrations were 109.20 μM ( $= 6.2 \times IC_{50}$) for DLD1-OXAR and 36.45 μM ( $= 5.4 \times IC_{50}$) for HCT15-OXAR. Experiments on resistant cell lines were performed after culturing in the medium without oxaliplatin for at least 2–3 weeks.

## Cell viability assay

The Deep Blue Cell Viability™ Kit (BioLegend, 424701) was used to analyze cell chemotherapy-induced cytotoxicity. After cells were

### The paper explained

**Problem**

Some colorectal cancer patients are resistant to oxaliplatin treatment, the most commonly used first line of therapy, whilst others develop resistance. Predicting which patients will or will not respond would allow for more appropriate treatment administration.

**Results**

This study identifies a relationship between two proteins, TXNIP and GDF15, that gives a readout of epithelial responsiveness to oxaliplatin-induced reactive oxygen species. The ratio of these proteins differs in aggressive vs non-aggressive tumors and in resistant vs sensitive tumors. The treatment of patient-derived organoids with oxaliplatin and the assessment of the GDF15/TXNIP ratio may be able to predict responsiveness, allowing for individualized care.

**Impact**

Patient-derived organoids are already being used as sentinels of disease in some clinical settings. This work allows for their use as prognosticators of treatment response to oxaliplatin by measuring the relationship between two factors, while also highlighting a cohort of patients who may benefit from anti-GDF15 therapy.

seeded into 96-well plates (5000 cells/well), oxaliplatin (Ebewe Pharma, Austria) was added to the wells in several doses for 48–72 h. The plate was incubated at 37 °C for 3 h following the addition of 1:10 volume ratio of Deep Blue Cell Viability™ reagent to each well. A CLARIOstar Plate Reader (Excitation: 530–570 nm, Emission = 590–620 nm) was used to detect the reduction of resazurin into resorufin and the OD value was used to calculate cell viability.

## Sample size estimates and blinding

For the clinical samples, we used the maximum number of cases we had access to with no cases removed. For the healthy primary material, we used at least five independent donors (unless stated), with no donors removed—this number was chosen based on previous work. Although not blinded with respect to diseased vs non-diseased when carrying out the assays, the experimentalists were blinded to clinical information until after the experiments had been performed and data analyzed.

## Statistical analyses

All in vitro experiments were performed in at least three independent replicates for three times. All quantitative data are presented as mean ± standard error of the mean (SEM) and were analyzed using GraphPad Prism 9.0. Variance was assessed as part of the GraphPad analysis and appropriate tests were chosen depending on whether a normal distribution was apparent and whether the variances were comparable. The means of the two datasets were compared using paired or unpaired $t$ tests unless otherwise stated in the legends. One-way ANOVA was used to evaluate multiple independent groups unless otherwise stated in the legends. The chi-squared test was applied to compare categorical variables. Kaplan–Meier analyses were performed via the survival package. A $P$ value <0.05 was considered statistically significant.

## For more information

Please see this website for further information on organoids, including their history and potential: https://www.technologynetworks.com/cell-science/articles/an-introduction-to-organoids-organoid-creation-culture-and-applications-369090. Please see these genecard links for the two main factors highlighted within this manuscript: https://www.genecards.org/cgi-bin/carddisp.pl?gene=GDF15; https://www.genecards.org/cgi-bin/carddisp.pl?gene=TXNIP.

## Data availability

The datasets produced in this study are available in the following databases. The RNA sequencing data have been deposited in the Sequence Read Archive (SRA) under the identifier PRJNA1105029. The mass spectrometry proteomics data have been deposited to the ProteomeXchange Consortium via the PRIDE partner repository with the dataset identifier PXD051666.

The source data of this paper are collected in the following database record: biostudies:S-SCDT-10_1038-S44321-024-00105-2.

## Peer review information

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

## Acknowledgements

This work was supported by China Scholarship Council Awards (No. 201806010012 to JD, No. 202006940028 to TP, No. 202306010087 to DW); CRUK Early Detection and Diagnosis Committee (Project grant, C1519/A27375). RB is supported by MR/R000026/1 and UCLH/UCL BRC who received a proportion of funding from the Department of Health's NIHR Biomedical Research Centres funding scheme. GA and JV are supported by CRUK Early Detection and Diagnosis Committee Project grant (C7675/A29313). ZA was supported by the KCL Breast Cancer Now Research Unit (grant KCL-Q2-Y5). KN was supported by Cancer Research UK Clinical Training Fellowship (Award number 176885). CG is supported by CRUK City of London Centre (CTRQQR-2021\100004). The authors acknowledge the CEMS proteomics core facility at KCL for proteomics data generation and analysis.

## Author contributions

**Jinhai Deng**: Conceptualization; Data curation; Validation; Investigation; Visualization; Methodology; Writing—original draft. **Teng Pan**: Conceptualization; Data curation; Validation; Investigation; Visualization; Methodology; Writing—review and editing. **Dan Wang**: Investigation. **Yourae Hong**: Formal analysis; Visualization. **Zaoqu Liu**: Investigation. **Xingang Zhou**: Formal analysis. **Zhengwen An**: Investigation. **Lifeng Li**: Investigation. **Giovanna Alfano**: Investigation. **Gang Li**: Formal analysis. **Luigi Dolcetti**: Investigation. **Rachel Evans**: Methodology. **Jose M Vicencio**: Methodology. **Petra Vlckova**: Investigation. **Yue Chen**: Formal analysis. **James Monypenny**: Methodology. **Camila Araujo De Carvalho Gomes**: Investigation. **Gregory Weitsman**: Supervision. **Kenrick Ng**: Methodology. **Caitlin McCarthy**: Investigation. **Xiaoping Yang**: Investigation. **Zedong Hu**: Formal analysis. **Joanna C Porter**: Supervision. **Christopher J Tape**: Supervision. **Mingzhu Yin**: Supervision. **Fengxiang Wei**: Supervision. **Manuel Rodriguez-Justo**: Project administration. **Jin Zhang**: Supervision. **Sabine Tejpar**: Supervision. **Richard Beatson**: Conceptualization; Data curation; Formal analysis; Supervision; Validation; Investigation; Visualization; Methodology; Writing—original draft; Writing—review and editing. **Tony Ng**: Conceptualization; Resources; Supervision; Funding acquisition; Writing—original draft; Project administration; Writing—review and editing.

Source data underlying figure panels in this paper may have individual authorship assigned. Where available, figure panel/source data authorship is listed in the following database record: biostudies:S-SCDT-10_1038-S44321-024-00105-2.

## Disclosure and competing interests statement

The authors declare no competing interests.

# Expanded View Figures

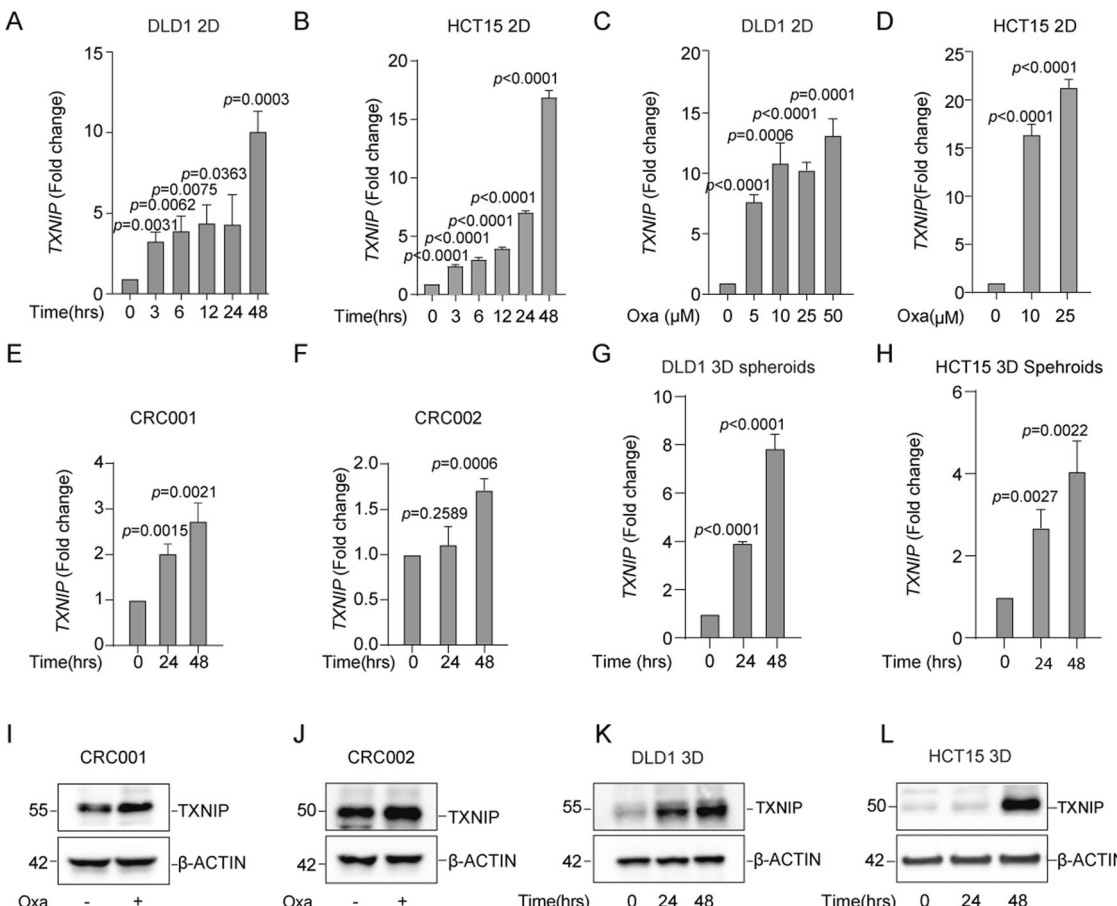

**Figure EV1. TXNIP expression is induced by oxaliplatin in different CRC models.**

(A, B) Assessment of *TXNIP* mRNA expression in DLD1 cells (A) or HCT15 cells (B) treated with oxaliplatin by q-RT-PCR analysis. Cells were treated with 10 μM oxaliplatin and harvested at indicated time points. (C, D) RT-qPCR analysis of *TXNIP* mRNA in DLD1 cells (C) or HCT15 cells (D) treated with oxaliplatin for 48 h at indicated concentrations. (E, F) RT-qPCR analysis of *TXNIP* mRNA in two different PDTOs treated with 10 μm oxaliplatin for indicated time periods. (G, H) RT-qPCR analysis of *TXNIP* mRNA in DLD1 (G) or HCT15 (H) spheroids treated with 10 μm oxaliplatin for indicated time periods. (I, J) Western blot analyzes of TXNIP post oxaliplatin treatment (10 μm) in two different PDTOs for 48 h. (K, L) Western blotting of TXNIP in DLD1 (K) or HCT15 (L) spheroids treated with 10 μm oxaliplatin for 48 h. Results shown are representative of three independent experiments. All values were expressed as mean ± SEM. *$P < 0.1$, **$P < 0.01$, ***$P < 0.001$, ****$P < 0.0001$, vs. Control. Source data are available online for this figure.

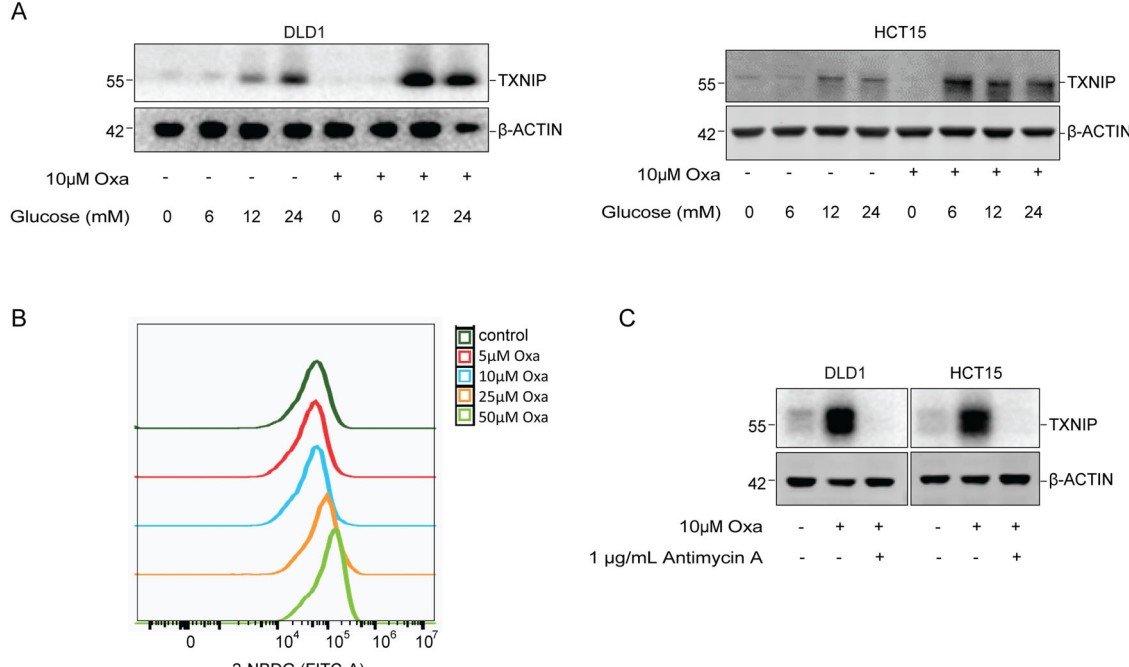

**Figure EV2. The induction of TXNIP by oxaliplatin is dependent on glucose availability, uptake and oxidative phosphorylation.**

(A) Western blotting analysis of TXNIP expression in DLD1 cells or HCT15 cells treated with different concentrations of glucose with or without 10 μM oxaliplatin for 48 h. β-actin was used as an internal reference. (B) DLD1 cells were treated with different concentration of oxaliplatin or vehicle (PBS) for 48 h. After treatment, 2-NBDG staining and flow cytometry were used to detect glucose uptake. (C) Immunoblot analysis of TXNIP in DLD1 cells or HCT15 cells treated with Antimycin A (1 μg/mL), an OXPHOS inhibitor or oxaliplatin (10 μm) or the combinational treatment for 48 h. Results shown are representative of two independent experiments.

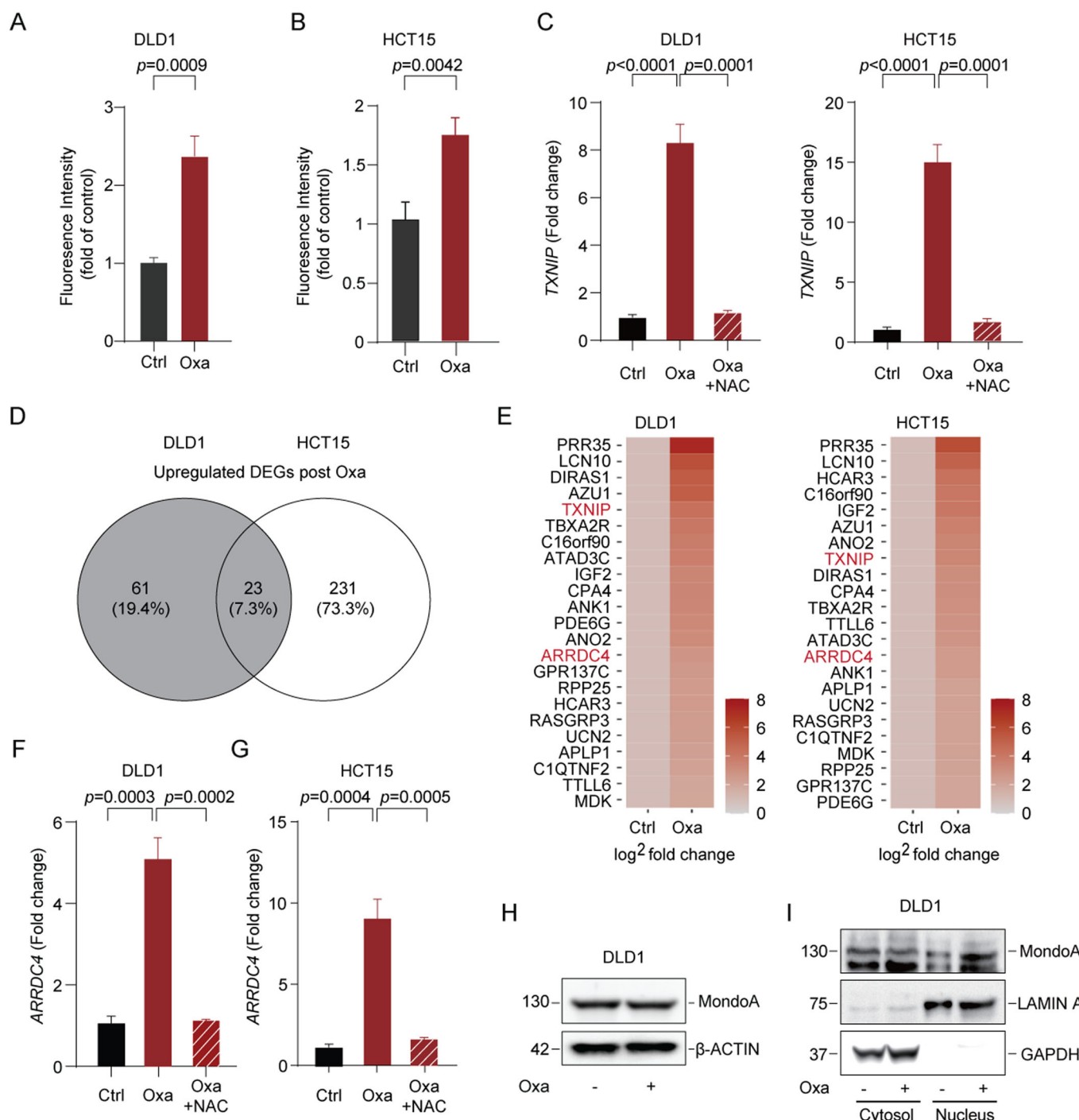

**Figure EV3. ROS drive the induction of TXNIP by inducing MondoA activity.**

(A, B) DLD1 cells (A) and HCT15 cells (B) were treated with 10 μm oxaliplatin with ROS measured at 48 h. (C) RT-qPCR analysis of *TXNIP* mRNA in DLD1 cells (left panel) or HCT15 cells (right panel) treated with N-acetyl-L-cysteine (NAC) (1.25 mM) or oxaliplatin (10 μm), or combinational treatment, for 48 h. (D) Overlapping DEGs ( > 4-fold change; Padj<0.05) from live DLD1 and HCT15 cells, after 48 h of 10 μm oxaliplatin treatment, as determined by RNA sequencing. (E) Heatmap showing 23 overlapping transcripts from D, in DLD1 cells (left panel) and HCT15 cells (right panel). (F, G) RT-qPCR analysis of *ARRDC4* mRNA in DLD1 cells (F) and HCT15 cells (G) treated with NAC (1.25 mM) or oxaliplatin (10 μm), or combinational treatment, for 48 h. (H) Immunoblot analysis of MondoA expression in DLD1 cells after 10 μm oxaliplatin treatment for 48 h. (I) Effects of oxaliplatin treatment (10 μm for 48 h) on subcellular localization of MondoA assessed by cell fractionation and immunoblotting, in DLD1 cells. LAMIN A - nuclear marker, GAPDH - cytoplasmic marker. Results shown (excluding D and E) are representative of three independent experiments. All values were expressed as mean ± SEM. **$P < 0.01$, ***$P < 0.001$, ****$P < 0.0001$, vs. Control. Source data are available online for this figure.

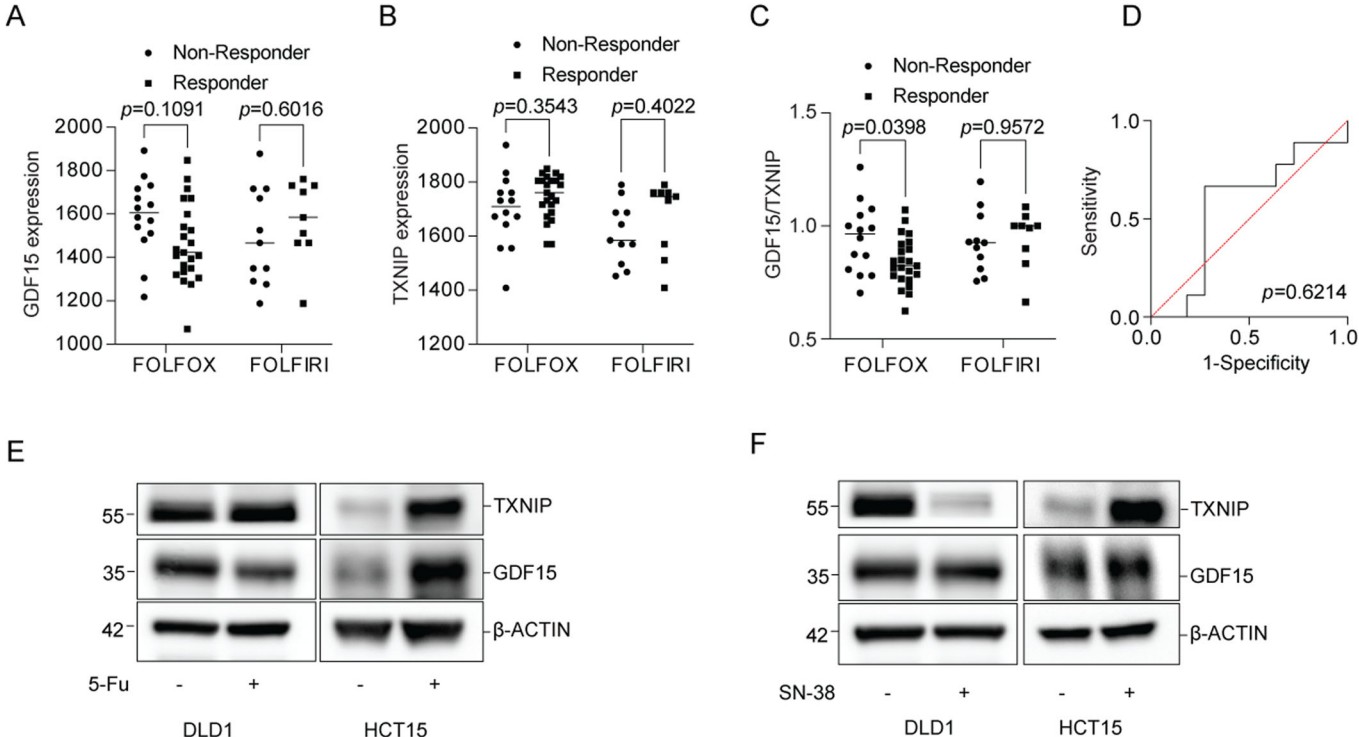

**Figure EV4. The described effects are specific for oxaliplatin.**

*GDF15* (**A**) or *TXNIP* (**B**) expression in responders or non-responders to FOLFOX or FOLFIRI treatment. (**C**) *GDF15/TXNIP* expression ratio in FOLFOX or FOLFIRI treated responders or non-responders. (**D**) Receiver operating characteristic (ROC) curve showing area under the curve and *P* value for the use of *GDF15/TXNIP* ratio in predicting therapeutic response to FOLFIRI (Responder [*n* = 9] nonresponder [*n* = 11]). (**E**) Immunoblot analysis of TXNIP and GDF15 after 48 h of 10 µm 5-Fu treatment in DLD1 and HCT15 cell lines. (**F**) Immunoblot analysis of TXNIP and GDF15 expression after 48 h of 10 µm SSN-38 treatment in DLD1 and HCT15 cell lines. (**E, F**) were repeated in three independent experiments. Two-way ANOVA, multiple comparisons test. *$P < 0.05$. Source data are available online for this figure.

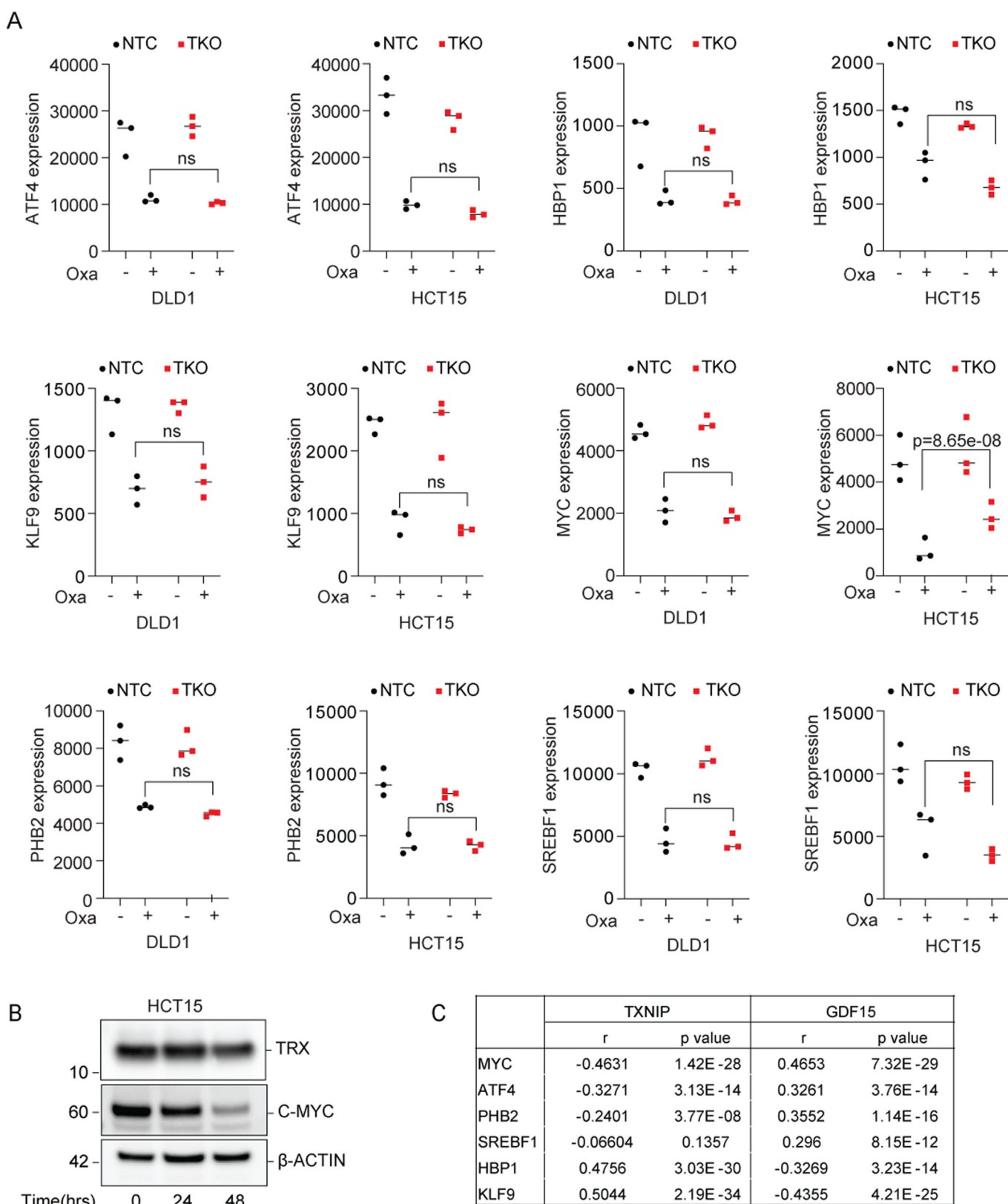

◀ **Figure EV5. C-MYC is downregulated by oxaliplatin in HCT15 cells in a TXNIP-dependent manner.**

(A) Transcript expression of GDF15 binding TFs (*ATF4, HBP1, KLF9, MYC, PHB2* and *SREBF1*) in control (NTC) and *TXNIP*-KO (TKO) cells (DLD1 or HCT15) with/ without 10 μm oxaliplatin treatment for 48 h. Each data point represents an biological replicate. (B) Western blotting analysis of TRX and C-MYC expression in HCT15 cells treated with 10 mM oxaliplatin at different time points. β-actin was used as an internal reference. (C) Correlations between the indicated TFs and *TXNIP* and *GDF15* in the TCGA COAD dataset. R and *P* values shown (Pearson's). (B) was repeated in 2 independent experiments. Source data are available online for this figure.

