## [Peer Review File · EMBO Molecular Medicine]

The MondoA-dependent TXNIP/GDF15 axis predicts oxaliplatin response in colorectal adenocarcinomas

Jinhai Deng, Teng Pan, Dan Wang, Yourae Hong, Zaoqu Liu, Xingang Zhou, Zhengwen An, Lifeng Li, Giovanna Alfano, Gang Li, Luigi Dolcetti, Rachel Evans, Jose Vicencio, Petra Vlckova, Yue Chen, James Monypenny, Camila Gomes, Gregory Weitsman, Kenrick Ng, Caitlin McCarthy, Xiaoping Yang, Zedong Hu, Joanna Porter, Chris Tape, Mingzhu Yin, Fengxiang Wei, Manuel Rodriguez-Justo, Jin Zhang, Sabine Tejpar, Richard Beatson, and Tony Ng

Corresponding authors: Richard Beatson (richard.1.beatson@kcl.ac.uk) , Tony Ng (tony.ng@kcl.ac.uk)

Review Timeline:

Transferred from Review Commons:	29th Jan 24
Editorial Decision:	19th Feb 24
Revision Received:	21st Jun 24
Accepted:	3rd Jul 24

Editor: Zeljko Durdevic

Transaction Report:

This manuscript was transferred to EMBO Molecular Medicine following peer review at Review Commons.

Review #1

1. Evidence, reproducibility and clarity:

Evidence, reproducibility and clarity (Required)

This is well done and interesting paper examining the connection between TXNIP and GDF15. The main thrust is that TXNIP upregulation chemotherapies, such as Oxa, results in an a down regulation of GDF15 early in tumorigenesis. Later in tumorigenesis, TXNIP upregulation is less pronounced, elevating GFP15 resulting in a blockage of tumor suppressive immune responses. Generally the work is convincing. For example, it's clear that TXNIP is up regulated by Oxa in an ROS and MondoA-dependent manner. Likewise its quite clear TXNIP loss reads to an upregulation of GDF15. However, it's also quite clear that Oxa suppresses GDF15 in a manner that appears to be completely independent of TXNIP. The writing in the paper implies strongly that there is a mechanistic connection between TXNIP and GDF15, but no experiments investigate this possibility. It seems equally likely that TXNIP and GDF15 represent independent parallel pathways. Even if TXNIP is a direct regulator of GDF15, it's also clear that other "factors" up or down-regulated by Oxa also contribute to the regulation of GDF15. These are not explored and even though TXNIP is highly regulated genes shown Figure 2 that are not identified or discussed that may also be contributing to GDF15 regulation. Further, the experiments treating PBMCs with conditioned media contain other cytokines/factors, in addition to GDF15, that likely also contribute the observed effects on the different immune cells understudy. The conditioned media from GDF15 knock out cells are a good experiment, but the media is not rigorously tested to see what other cytokines/factors might have also been depleted. Perhaps a GDF15 complementation experiment would help here. Finally, even if completely independent, a TXNIP/GDF15 ratio does seem to have utility in determining chemo-therapeutic response.

****Other major points:****

1. Please label the other highly regulated genes shown in Fig 2A and B. Might they also explain some of the underlying biology. This could be on the current figures or in a supplement, though the former is preferred.
2. Please address why the TXNIP induction is so much less in patient-derived organoids vs. cell line spheroids (Fig S2). By the western blots, TXNIP inductions in the organoids looks quite modest. Further, the text is quite cryptic and implies that the "upregulation" is similar in both organoids and spheroids.
3. What was the rationale of performing the MS experiment on control and TXNIP KO DLD1 cells in the absence of oxaliplatin? The other experiments in Fig 3 clearly show that Oxa can repress GDF15 even in the absence of TXNIP, which implicates other pathways. ARRDC4? Or something else? This needs to be addressed.
4. The data in 3J with the MondoA knockdown is not convincing. The knockdown is weak and TXNIP goes down a smidge. Agree that GDF15 goes up

****Minor points****

1. Line 79. The "loss" of TXNIP/GDF15 axis is confusing. It's really loss of TXNIP and upregulation of GDF15, right?
2. Please provide an explanation for the different stages in tables 1 and 2. This will likely not be clear to non-clinicians.
3. Line 231 should probably read ...cysteine (NAC), a reactive oxygen species inhibitor,
4. Line 247, should be RT-qPCR I think.
5. Lines 343-345. I don't quite understand the wording. Does this mean to say that 675 soluble proteins were not changed between the condition media from both cell populations?
6. The data in FigS1 B and C don't seem to reach the standard p value of > 0.05

****Referee Cross-Commenting****

cross comment regarding referees 2 and 3 above.

I'm am convinced that TXNIP is at least contemporaneously upregulated with GDF15 downregulation. However, the strong implication from the writing is that TXNIP regulates GDF15 directly. I agree with the comment above that exploring mechanisms may be open-ended especially as TXNIP has been implicated in gene regulation by several different mechanism. I'd be satisfied with a more open-minded discussion of potential mechanisms by which TXNIP may repress GDF15 and the possibility of other parallel pathways that likely contribute to GDF15 repression.

2. Significance:

Significance (Required)

This is an interesting contribution but the mechanistic connection between GDF15 and TXNIP is relatively weak. That said, even as independent variables they do seem to have utility in predicting therapeutic response.

3. How much time do you estimate the authors will need to complete the suggested revisions:

Estimated time to Complete Revisions (Required)

(Decision Recommendation)

Between 3 and 6 months

4. Review Commons values the work of reviewers and encourages them to get credit for their work. Select 'Yes' below to register your reviewing activity at Web of Science

Reviewer Recognition Service (formerly Publons); note that the content of your review will not be visible on Web of Science.

Yes

Review #2

1. Evidence, reproducibility and clarity:

Evidence, reproducibility and clarity (Required)

The manuscript by Deng et al. investigates a mechanistic link between TXNIP and GDF15 expression and oxaliplatin treatment and acquired resistance. They observe an upregulation in TXNIP expression in the tumors of patients who have previously received chemotherapy. They demonstrate oxaliplatin-driven MondoA transcriptional activity is what underlies the induction of TXNIP. They further demonstrate that TXNIP is a negative regulator of GDF15 expression. Together, oxaliplatin induces MondoA activity and TXNIP expression, resulting in a downregulation of GDF15 expression and consequently decreased Treg differentiation.

****Major Comments****

1. The authors suggest that TXNIP induction and GDF15 downregulation are a common effect of chemotherapies; however, the mechanistic studies were limited to oxaliplatin. The authors should clarify this point through further investigation using other commonly used CRC chemotherapies (5-FU, irinotecan, etc.), or through textual changes. To be clear, I think that the oxaliplatin results could potentially stand on their own but would require additional clarification. For example, regarding the patient samples analyzed in 1D and 4F, which patients received oxaliplatin? Could the analysis of publicly available molecular data be drilled down to just the patients who received oxaliplatin?

2. The data demonstrating the induction of MondoA transcriptional activity and TXNIP expression in response to oxaliplatin treatment is quite convincing. The data regarding ROS induction of TXNIP is interesting, especially in light of other studies arguing that ROS limits MondoA activity (PMID: 25332233). Given this apparent disparity, I think that this study could really be strengthened by also investigating other potential mechanisms of oxaliplatin induction of MondoA. In particular, given many studies arguing for direct nutrient-regulation of MondoA, the authors should address the potential for oxaliplatin regulation of glucose availability and a potential glucose dependence of oxaliplatin-induced TXNIP.

3. In line with the previous point, since MondoA activity and TXNIP expression are sensitive to glucose levels, the authors should investigate oxaliplatin-regulation of TXNIP under

physiological glucose levels. No need to replicate everything, just key experiments.

4. The authors did a good job of linking TXNIP and GDF15 in untreated conditions; however, the data arguing for oxaliplatin regulation of GDF15 through TXNIP is less clear. For example, in 3B-H, oxaliplatin treatment reduces GDF15 approximately to the same extent in the NTC and TKO cells, potentially in line with a mechanism of downregulation that doesn't involve TXNIP.

****Minor Comments****

1. The presentation of data in Figure 5 is confusing. A-B include raw cell numbers, whereas C-F show "normalized proliferation." What does this mean? And how was the normalization done?

****Referee Cross-Commenting****

cross-comment regarding reviewer #1

I agree with the referee that the link between TXNIP and GDF15 is weak, though as I mentioned before, this is particularly true in the context of oxaliplatin-regulation of TXNIP. I agree that given all the presented data, it is likely that oxaliplatin-regulation of TXNIP and GDF15 are independent. In my opinion, the referee brought up all valid concerns, but this is by far the biggest concern that I share.

cross-comment regarding reviewer #3

The major concern that this referee addresses is whether another transcription factor supersedes the proposed MondoA/TXNIP induction in regulating GDF15 expression in later stage CRC. In my opinion, this and other concerns of the referee are all valid, but still I remain unconvinced that TXNIP induction underlies the oxaliplatin-regulation of GDF15. I think fleshing out that aspect of the study would potentially help the authors tease apart how this potential MondoA-TXNIP-GDF15 axis is dysregulated later in CRC progression.

2. Significance:

Significance (Required)

Generally speaking the experiments are well controlled and the findings are significant and novel. Though the link between MondoA activity and ROS could be strengthened, and the data could be validated under more physiological settings. Further, the authors should clarify their interpretations so as to not overstate the findings.

3. How much time do you estimate the authors will need to complete the suggested revisions:

Estimated time to Complete Revisions (Required)

(Decision Recommendation)

Between 1 and 3 months

Yes

Review #3

1. Evidence, reproducibility and clarity:

Evidence, reproducibility and clarity (Required)

In this well-written manuscript, the authors show that chemotherapy increases a MondoA-dependent oxidative stress-associated protein, TXNIP, in chemotherapy-responsive colorectal cancer cells. They show that TXNIP negatively regulates GDF-15 expression. GDF-15, in turn, correlates with the presence of T cells (Treg), and inhibits CD4 and CD8 T cell stimulation. In advanced disease and chemo-resistant cancers, upregulation of TXNIP and downregulation of GDF-15 appear to get lost. Based on a somewhat smallish data set, the authors suggest that the pre-treatment GDF-15/TXNIP ratio can predict responses to oxaliplatin treatment.

This is a very interesting, novel finding. In general, the quality of the experiments and the data are high and the conclusions appear sound. Still, there are a number of aspects that should still be improved:

The observed loss of the ROS - MondoA - TXNIP - GDF15 axis in chemoresistant and/or metastatic tumors implies that another transcription factor or pathway becomes dominant upon tumor progression. As this switch would be key to better understanding the mechanism underlying the prognostic role of the TXNIP/GDF15 ratio, the authors should at least do data mining followed by ChEA or Encode (or other) analysis to identify transcription factors or pathways that become activated in late-stage/metastatic CRC cells. There is a high likelihood that a transcription factor or pathway involved in GDF-15 upregulation in cancer (e.g. p53, HIF1alpha, Nrf2, NF-kB, MITF, C/EBPβ, BRAF, PI3K/AKT, MAPK p38, EGR1) supersedes

the inhibitory effect of the MondoA-TXNIP axis. As it stands, the proposed loss of function of the ROS - MondoA - TXNIP - GDF-15 axis is far less convincing than almost all other aspects of the study.

My further criticisms are mostly more technical:

Figure 2 I-L: What was the extent of MondoA downregulation achieved by siRNA treatment? Could the effects also be seen with the small molecule mondoA inhibitor SBI-477 (or a related substance)?

How do you explain the different GDF-15 levels between untreated non-target control cells (NTC) and TXNIP knock-down cells (TKO) in Figures 3C-F?

In figures 3 E-G the dots for the individual measurements should be indicated. This would be more informative than just the bar graphs.

Figure 4C,D and Table 3: Data on the role of GDF-15 in CRC are largely valedictory of previous work (e.g. Brown et al. Clin Cancer Res 2003, 9(7):2642-2650, Wallin et al., Br J Cancer. 2011 May, 10;104(10):1619-27). Therefore, the previous studies should be cited.

Figure 5C-F: Please indicate in the figure legend how proliferation was assessed.

Figure S8E-G: Please indicate the analysed parameters in the graphs. In Figure S8G, the legend just indicates that "aggression of tumour" is dichotomized and plotted. This clearly requires a better definition.

The authors propose a novel ROS - MondoA - TXNIP - GDF15 - Treg axis, where MondoA activation, TXNIP up- and GDF-15 downregulation enhance tumor immunogenicity. While this axis has been analyzed in some detail, GDF-15 is not only linked to induction of regulatory T cells. There has been a report showing that GDF-15/MIC-1 expression in colorectal cancer correlates with the absence of immune cell infiltration (Brown et al. Clin Cancer Res 2003, 9(7):2642-2650). The link between GDF-15 and immune cell exclusion has also been confirmed in other conditions, including different cancers (Kempf et al. Nat Med 2011, 17(5):581-588, Roth P et al. Clin Cancer Res 2010, 16(15):3851-3859, Haake et al. Nat Commun 2023, 14(1):4253). A key mechanism is the GDF-15 mediated inhibition of LFA-1 activation on immune cells. As the authors argue that the described pathways turns cold tumors hot in response to oxaliplatin-based chemotherapy, this GDF-15 dependent immune cell exclusion mechanism might be at least as relevant than induction of Treg. Likewise, inhibition of dendritic cell maturation by GDF-15 (Zhou et al. PLoS One 2013, 8(11):e78618) could explain why GDF-15high tumors are immunologically cold.

The authors propose that the pathways discovered by them contributed to the "heating up" of the tumor microenvironment after oxaliplatin-based chemotherapy. The authors should thus look in their data sets for the presence of cytotoxic T cells and their possible correlation with TXNIP and GDF-15 levels.

The paragraph on GDF-15 receptors needs to be corrected: The purported role of a type 2 transforming growth factor (TGF)-beta receptor in GDF-15 signalling had been due to a frequent contamination of recombinant GDF-15 with TGF-beta (Olsen et al. PLoS One 2017, 12(11):e0187349). There have been a number of screenings for GDF-15 receptors that have all failed to show an interaction between GDF-15 and TGF-beta receptors. Instead, only GFRAL was found in these large-scale screenings (Emmerson et al. Nat Med 2017, 23(10):1215-1219, Hsu et al. Nature 2017, 550(7675):255-259, Mullican et al. Nat Med 2017, 23(10):1150-1157, Yang et al. Nat Med 2017, 23(10):1158-1166). The one subsequent report that shows a link between GDF-15, engagement of CD48 on T cells and induction of a regulatory phenotype (Wang et al. J Immunother Cancer 2021, 9(9)) still awaits independent validation. Considering that CD48 lacks an intracellular signaling domain that would be critical for a classical receptor function, I recommend to be more cautious regarding the role of CD48 as GDF-15 receptor. Given the mechanism outlined by Wang et al. the word interaction partner might be more apt. Moreover, an anti-GDF-15 antibody would be a good control for the experiments involving an anti-CD48 antibody in Figure 5.

Cell surface externalization of annexin A1 has been described as a failsafe mechanism to prevent inflammatory responses during secondary necrosis (PMID: 20007579). Thus, I am surprised that the authors list annexin A1 among the immune-stimulatory molecules exposed or released in response to chemotherapy-induced cell death (line 103). Please clarify!

****Referee Cross-Commenting****

Regarding the cross-comment by referee 2:

In my opinion, the data shown in Figure 3C-H clearly demonstrates that TXNIP can repress GDF-15 expression. I agree that there will likely be further regulators. The GDF-15 promoter is constantly regulated by a multitude of factors (which mostly induce transcription). As downregulation of GDF-15 in response to oxaliplatin is the opposite of the frequently described induction of GDF-15 upon chemotherapy, net effects may always be "smudged" by contributions from different pathways (e.g. by cell stress due to siRNA transfection). Therefore, I believe that the data are as good as it will get. Accordingly, I would not force the authors to further amplify the observed effect.

cross comment regarding referee #1:

I share the general assessment of the referee and recognize the very detailed mechanistic analysis. To further support the moderate effects of the MondoA knockdown, a small molecule inhibitor like SBI-477 might be useful. (I had already suggested using this inhibitor to support these data.)

Still, my view on the potential relevance of oxaliplatin-induced, TXNIP-independent downregulation of GDF-15 differs from that of referee 1. In the clinics, platinum-based chemotherapy is one of the strongest inducers of GDF-15 (compare Breen et al. GDF-15 Neutralization Alleviates Platinum-Based Chemotherapy-Induced Emesis, Anorexia, and Weight Loss in Mice and Nonhuman Primates. Cell Metabolism 32(6), P938-950, 2020.DOI:<https://doi.org/10.1016/j.cmet.2020.10.023>). I was thus surprised that the authors found a pathway, which leads to an outcome that an exactly opposite effect. Thus far, the only obvious reason for reduced GDF-15 secretion upon treatment with cytotoxic drugs was a

reduction in tumor cell number due to cytotoxicity. Still, the authors managed to convince me that the described pathway (ROS - MondoA - TXNIP - GDF-15) exists. (Here, I still largely concur with referee 1.) Moreover, as we have identified some factors required for GDF-15 biosynthesis that could easily interact with TXNIP, I find the proposed mechanism plausible. Nevertheless, as a downregulation of GDF-15 in response to chemotherapy is hardly ever observed in late-stage cancers, I believe that the observed switch in pathway activation between early- and late-stage cancers might be highly relevant - in particular, as there is so much evidence for platinum-based induction of GDF-15 in late-stage cancer patients. Emphasizing the divergent clinical observations (e.g. by Breen et al.) could thus help to put the finding into perspective. Analysing TXNIP-independent mechanisms involved in the oxaliplatin-dependent repression of GDF-15, as suggested by referee #1, will require enormous efforts and resources, and may still turn out to be fruitless. Personally, I would thus be content if the authors just mentioned possible contributions from other pathways upon cancer progression. To me, the described pathway seems to be limited to early-stage cancers, and the actual finding that GDF-15 is downregulated is an interesting observation, irrespective of further involved pathways.

cross comment regarding referee #2:

I fully agree with the referee that activation of the pathway by further chemotherapeutic drugs could be a valuable addition. As Guido Kroemer's lab has described oxaliplatin to induce a more immunogenic cell death compared to other platinum-based chemotherapies, even a rather limited comparison between oxaliplatin and cisplatin could be very interesting.

2. Significance:

Significance (Required)

In general, this is a very interesting manuscript describing a cascade of events that may contribute to successful chemotherapy (which likely requires induction of an immune response against dying tumor cells.) The observation that this pathway is only active in early/non-metastatic cancer cells is striking. Unfortunately, the authors cannot explain inactivation of this pathway in later stage/ metastatic/ highly aggressive cancers. Understanding this switch could easily be the most important finding triggered by this report. Therefore, I highly recommend to make some effort in this direction. Strikingly, the authors find that disruption of TXNIP-mediated GDF-15 downregulation is strongly associated with worse prognosis. They also suggest that this ratio could indicate whether a patient will respond to oxaliplatin-based chemotherapy.

Altogether, the findings described in manuscript are very novel and may have prognostic (or, in case of the presumed loss of the MondoA - TXNIP - GDF-15 pathway) therapeutic implications. Thus, the manuscript certainly fills various gaps and should be of major interest for cell biologists working on immunogenic cell death, or colorectal cancer, or MondoA, TXNIP or GDF-15. Still, due to its translational implications, it would also be worthwhile reading for a large number of researchers in the oncology field.

3. How much time do you estimate the authors will need to complete the suggested revisions:

Estimated time to Complete Revisions (Required)

(Decision Recommendation)

Between 1 and 3 months

Yes

29 January 2024

Dear Dr Monaco and Reviewers

Firstly can we convey our genuine thanks to the reviewers for their extremely helpful and extremely well-informed comments – their time spent critically and carefully appraising this manuscript has improved it markedly, and we are very grateful.

We will address the comments in order, with the original reviewer's comments included in red. However, in summary we have provided additional experimental data and new analyses in seven new figure panels, four additional Supplementary figures, as well as one figure for reviewers only.

Reviewer #1 (Evidence, reproducibility and clarity (Required)):

This well done and interesting paper examining the connection between TXNIP and GDF15. The main thrust is that TXNIP upregulation chemotherapies, such as Oxa, results in an a down regulation of GDF15 early in tumorigenesis. Later in tumorigenesis, TXNIP upregulation is less pronounced, elevating GFP15 resulting in a blockage of tumor suppressive immune responses. Generally the work is convincing. For example, it's clear that TXNIP is up regulated by Oxa in an ROS and MondoA-dependent manner. Likewise its quite clear TXNIP loss reads to an upregulation of GDF15. However, it's also quite clear that Oxa suppresses GDF15 in a manner that appears to be completely independent of TXNIP. The writing in the paper implies strongly that there is a mechanistic connection between TXNIP and GDF15, but no experiments investigate this possibility.

We feel this is very fair and is reflective of a) perhaps an overemphasis of the TXNIP knockout observation and supportive tissue data, which suggests a relationship but not a mechanistic understanding b) an underemphasis of the data in Figure 3 that shows a decrease in GDF15 after oxaliplatin treatment in TXNIP knockout lines.

We have addressed these concerns in several ways:

1. We have carried out knockdown experiments using siRNA for *ARRDC4*, which we felt, given its regulation by MondoA and ROS, and homology to TXNIP, may also regulate GDF15. This was found to be the case and may explain the data in Figure 3. At the very least it shows that other factors involved in oxidative stress management may have similar impacts – a form of functional redundancy.

Lines 553-559 “Finally, given our previous data (Figure S4) we looked to assess the role of *ARRDC4* on GDF15 expression. In the absence of oxaliplatin, knocking down *ARRDC4* in DLD1 and HCT15 cells drove an increase in GDF15. When challenged with oxaliplatin, both *ARRDC4* and TXNIP expression increased and GDF15 decreased. When the *ARRDC4*

knockdown was challenged TXNIP increased further and GDF15 decreased further (Figure S6G-J). Given the common regulatory pathways and homology between TXNIP and ARRD4, and their similar functional roles, we suggest these data are evidence of redundancy within this system. “

Figure S6. Oxaliplatin treatment, TXNIP and ARRD4 suppress GDF15 expression. (A-B) Immunoblotting of GDF15 in DLD1 cells after treatment with 10 μ M oxaliplatin at indicated time points. (A); after treatment of different dosages of oxaliplatin for 48 hours (B). (C-D) Immunoblotting of TXNIP and GDF15 in control (NTC) and TXNIP-overexpressing (TXNIPa) DLD1 cells with or without 10 μ M oxaliplatin treatment for 48h (C); pooled densitometric data from C (D). Standard error bars are shown n=3. (E) Quantitation of immunofluorescence from Figure 3I (GDF15 levels relative to cell area) from 3 independent experiments. (F) Immunoblotting of TXNIP and GDF15 in TXNIPa or NTC cells treated with oxaliplatin (10 μ M) or combined treatment with oxaliplatin and NAC (1.25 mM) for 48h. (G-J) DLD1 cells (G-H) or HCT15 cells (I-J) were treated with siARRDC4 or NTC +/- 10 mM oxaliplatin for 48h. TXNIP and GDF15 protein expression was measured by Western, with B-Actin as a loading control (G, I). ARRD4 transcript expression was measured using q-RT-PCR. Expression was normalised to GAPDH and fold change compared to NTC treated cells shown (H, J). Results shown are representative of three independent experiments. All values were expressed as mean \pm SEM. * p <0.1, ** p <0.01, **** p < 0.0001, vs. Control.

We have included some context in the discussion:

Lines 930-933: “Further support for both TXNIP and ARRDC4’s role in regulating GDF15 after the induction of ROS comes from a pan cancer meta-analysis assessing the impact of metformin (which has been reported to inhibit ROS) on gene expression. Here the top two downregulated genes were *TXNIP* and *ARRDC4* and the top four upregulated genes were *DDIT4*, *CHD2*, *ERN1* and *GDF15*⁷²

2. We have tempered the text:

Lines 522-524 “It is important to note however that here we saw clear evidence that TXNIP was not solely responsible for the downregulation of GDF15 post oxaliplatin treatment, with decreased levels seen in knockout lines (Figure 3C-G, S5E).”

Lines 926-929 “It must be stressed that these data do not place TXNIP as the sole regulator of GDF15, for example ARRDC4 can also be seen to regulate GDF15. We envisage TXNIP as one of a number of ROS-dependent GDF15 regulators, with this redundancy potential evidence of the importance of this regulatory framework.”

3. We have carried out additional analysis detailed in the discussion and in Figure S12 which suggests TXNIP impacts MYC function, as reported elsewhere (detailed below). For ease, the key paper can be accessed through this link <https://journals.plos.org/plosbiology/article?id=10.1371/journal.pbio.3001778>

Lines 934-956: “The main shortcoming of this paper is the lack of mechanistic understanding linking TXNIP to GDF15. There are 650 transcription factors that have been shown, or are predicted, to bind to GDF15 promoter and/or enhancer regions. By assessing our list of differentially expressed genes (Suppl. Table 1-2) for the presence of these factors we identified 6 GDF15 binding TFs that show significantly decreased expression after oxaliplatin treatment in both cell lines (*ATF4*, *MYC*, *SREBF1*, *PHB2*, *HBP1*, *KLF9*). There was only one, *MYC*, that was downregulated by oxaliplatin treatment (validated; Figure S12A), and with this downregulation partially being rescued in a matched *TXNIP* knockout line (Figure S12B). We then observed that c-myc has been shown or is predicted to bind to promoter/enhancer regions of the top five transcriptomic and proteomic differentials in *TXNIP* knockout lines, including TXNIP itself (apart from C16orf90). Even with c-myc’s promiscuity (binds to 10-20% of all promoters/enhancers) this may be suggestive of a specific relationship. Finally, when looking at the correlations between these 6 TFs and *TXNIP* and *GDF15* in the TCGA COAD dataset, *MYC* has the greatest and most significant negative correlation to *TXNIP* ($r=-0.4631$ $p=1.42e-28$) and the greatest and most significant positive correlation to *GDF15* ($r=0.4653$ $p=7.32e-29$). *ATF4* and *PHB2* are the other TFs in the list, that show the same significant trends (Figure S12C), and therefore may play a role in the TXNIP-independent oxaliplatin-dependent regulation of GDF15. Further exploration of these additional TFs is outside the scope of the current manuscript.

MYC’s role in bridging from TXNIP to GDF15 is further supported by a recent paper which shows that TXNIP is “a broad repressor of MYC genomic binding” and that “*TXNIP* loss mimics MYC overexpression”⁷³. Furthermore, the inter-dependent regulatory relationship between MondoA, TXNIP, and MYC has been seen in a variety of models⁷⁴, whilst the impact of NAC on MYC-dependent pathways has been seen in lymphoma⁷⁵. These studies lend credence to the idea that MYC is the most likely TXNIP-regulated TF that regulates GDF15 in our systems.”

A

B

C

	TXNIP		GDF15	
	r	p value	r	p value
MYC	-0.4631	1.42E-28	0.4653	7.32E-29
ATF4	-0.3271	3.13E-14	0.3261	3.76E-14
PHB2	-0.2401	3.77E-08	0.3552	1.14E-16
SREBF1	-0.06604	0.1357	0.296	8.15E-12
HBP1	0.4756	3.03E-30	-0.3269	3.23E-14
KLF9	0.5044	2.19E-34	-0.4355	4.21E-25

Figure S12. C-MYC is downregulated by oxaliplatin in HCT15 cells in a TXNIP dependent manner. (A) Transcript expression of GDF15 binding TFs (*ATF4*, *HBP1*, *KLF9*, *MYC*, *PHB2* and *SREBF1*) in control (NTC) and *TXNIP*-KO (TKO) cells (DLD1 or HCT15) with/ without 10µM oxaliplatin treatment for 48h. Each data point represents an biological replicate. (B) Western blotting analysis of TRX and C-MYC expression in HCT15 cells treated with 10µM oxaliplatin at different time points. β-actin was used as an internal reference. (C) Correlations between the indicated TFs and *TXNIP* and *GDF15* in the TCGA COAD dataset. R and p values shown (Pearson's).

It seems equally likely that TXNIP and GDF15 represent independent parallel pathways. Even if TXNIP is a direct regulator of GDF15, it's also clear that other "factors" up or down-regulated by Oxa also contribute to the regulation of GDF15. These are not explored and even though TXNIP is highly regulated genes shown Figure 2 that are not identified or discussed that may also be contributing to GDF15 regulation.

As mentioned above, the new data suggests that at least one other factor, ARRDC4, can regulate GDF15 (changes upon oxaliplatin treatment) and that MYC is a potential mechanistic bridge between TXNIP and GDF15. Whilst assessing for the transcription factor that may link TXNIP and GDF15 we found an additional 5 TXNIP-independent factors (ATF4, PHB2, SREBF1, HBP1, KLF9) that bind to GDF15 promoter/enhancer regions and are downregulated post-oxaliplatin treatment. When looking at correlations between these factors and GDF15 in the TCGA COAD dataset, ATF4 and PHB2 correlate most closely with GDF15 (when removing MYC) and so we would cautiously suggest that these may be the most pertinent. This data is now included.

Further, the experiments treating PBMCs with conditioned media contain other cytokines/factors, in addition to GDF15, that likely also contribute the observed effects on the different immune cells understudy. The conditioned media from GDF15 knock out cells are a good experiment, but the media is not rigorously tested to see what other cytokines/factors might have also been depleted.

The TXNIP knockout media is the same as that analysed by mass spec and the protein array, however as the reviewer states there is no analysis (excluding assessing for the presence or absence of GDF15) on the double knockout supernatant or over-expression supernatant. The text has been corrected as follows:

Lines 675-679. "In light of other secreted factors being seen to be regulated by TXNIP (Figure 3A-B), we included double knockouts (*TXNIP* and *GDF15* knockout; GTKO) as well as an overexpression system (*GDF15a*) to test for GDF15 specific effects. However, we do not know the impact of knocking out or overexpressing *GDF15* on the broader secretome."

Perhaps a GDF15 complementation experiment would help here.

We felt that the association between GDF15 and Treg induction is reasonably well established in the literature, and so once we saw that the supernatant from our GDF15 overexpression system (+/- CD48 blockade) complemented what has already been demonstrated, we were encouraged. However we needed more – hence the TCGA data and IHC staining.

Finally, even if completely independent, a TXNIP/GDF15 ratio does seem to have utility in determining chemo-therapeutic response.

We agree – we feel that conceptually this may be the most interesting part of the project and is an example of what can be done with these tools.

Other major points:
1. Please label the other highly regulated genes shown in Fig 2A and B. Might they also explain some of the underlying biology. This could be on the current figures or in a supplement, though the former is preferred.

Many thanks – we have done this.

2. Please address why the TXNIP induction is so much less in patient-derived organoids vs. cell line spheroids (Fig S2). By the western blots, TXNIP inductions in the organoids looks quite modest. Further, the text is quite cryptic and implies that the "upregulation" is similar in both organoids and spheroids.

You are absolutely correct. Many apologies, the wording has changed:

Lines 320-323 “In both models we observed the upregulation of *TXNIP* mRNA (Figure S2E-H) and TXNIP protein (Figure S2I-L) after oxaliplatin treatment, with spheroids showing greater responsiveness. This difference is most likely due to culturing conditions or differences in the number and location of cycling cells.”

We have two possible explanations. Firstly the media in which the organoids are cultured contains a lower glucose concentration than that used for the spheroids. As per some of our new data (Figure S3 – later in the rebuttal), the upregulation of TXNIP after oxaliplatin is glucose dependant, with lower concentrations resulting in less of a differential. Secondly, while restricted to the periphery, the Ki67 signal in DLD1 spheroids is quite pronounced indicating that, within the outer zone, many cells (probably the majority) are in the S/G1/G2 phase of the cell cycle at any given point in time (figure below this text).

This is not the case for the organoids, where the Ki67 (and pCDK1) signal is quite weak, and only sporadic in the outer layer. So we believe that there are many more rapidly cycling cells in the most drug-exposed layer of spheroids when compared to the comparable region in organoids. As the spheroid cells are likely cycling more rapidly, they would also be expected to be more adversely affected by the drug within the finite drug treatment window. Indeed, these spheroids grow large, and quite quickly. If the organoid cells are cycling more slowly and if, within the cell layer most exposed to drug, these cycling cells are less abundant, then the TXNIP response may well be subdued in organoids when compared with spheroids.

Analysis of Ki67 expression in CRC spheroids

Analysis of Ki67 and pCDK1 expression in CRC organoids

We have decided to not include the above (full) explanation and figure within the new draft, as we feel it may distract from the central message. However do let ourselves and the editor know if you disagree.

3. What was the rationale of performing the MS experiment on control and TXNIP KO DLD1 cells in the absence of oxaliplatin? The other experiments in Fig 3 clearly show that Oxa can repress GDF15 even in the absence of TXNIP, which implicates other pathways. ARRDC4? Or something else? This needs to be addressed.

We adopted this approach because of the order in which the assays occurred and technical issues surrounding the use of post-oxaliplatin treated supernatant. By the time we moved to the proteomics we had already identified, and validated, GDF15 as our number one

candidate (initially from the protein array), in terms of response to oxaliplatin and dependence on TXNIP. This led us to the next stage of the project – to assess the environmental impacts of this factor *in vitro* before validation *in situ*. To do this, aware of the issue of contaminated recombinant GDF15, we decided early on to use cell line supernatant. We carried out some pilot studies on immune cells using supernatant from oxaliplatin treated cell lines and we had several technical issues (difficulty in determining the correct controls, immune cell death...). This changed the emphasis to using supernatant from knockout models rather than knockout and treated models. Before we began these assays in earnest we wanted to assess exactly what was enriched in TXNIP knockout supernatant and so we turned to proteomics. When this further validated GDF15, we then generated GDF15 and TXNIP/GDF15 knockouts to further elucidate GDF15's role specifically.

With regards the other pathways, as you correctly predicted, ARRDC4 also appears to regulate GDF15 – many thanks for helping with this line of enquiry. Please see earlier in the rebuttal for more details and the data.

4. The data in 3J with the MondoA knockdown is not convincing. The knockdown is weak and TXNIP goes down a smidge. Agree that GDF15 goes up

We agree. We have re-run this and pooled the densitometry data – see new figure below (Panel 3J).

Minor points

1. Line 79. The "loss" of TXNIP/GDF15 axis is confusing. It's really loss of TXNIP and upregulation of GDF15, right?

Absolutely - corrected to responsiveness.

Lines 144-147: "Intriguingly, multiple models including patient-derived tumor organoids demonstrate that the loss of TXNIP and GDF15 responsiveness to oxaliplatin is associated with advanced disease or chemotherapeutic resistance, with transcriptomic or proteomic GDF15/TXNIP ratios showing potential as a prognostic biomarker."

2. Please provide an explanation for the different stages in tables 1 and 2. This will likely not be clear to non-clinicians.

Many thanks. The following has been added at the bottom of the second table.

Lines 304-309: "The TNM staging system stands for Tumor, Node, Metastasis. T describes the size of the primary tumor (T1-2; <5cm. T3-4; >5cm). N describes the presence of tumor cells in the lymph nodes (N0; no lymph nodes. N1-3 >0). M describes whether there are any observable metastases (M0; no metastases. M1; metastases). The clinical stage system is as follows: I/II; the tumor has remained stable or grown, but hasn't spread. III/IV; the tumor has spread, either locally (III) or systemically (IV)."

3. Line 231 should probably read ...cysteine (NAC), a reactive oxygen species inhibitor,

Many thanks - corrected

4. Line 247, should be RT-qPCR I think.

Many thanks - corrected

5. Lines 343-345. I don't quite understand the wording. Does this mean to say that 675 soluble proteins were not changed between the condition media from both cell populations?

Yes, exactly this. We have removed as this is superfluous and confusing.

6. The data in FigS1 B and C don't seem to reach the standard p value of > 0.05

Very true – we have rewritten the text to make sure the reader knows there is no significance.

Lines 269-271. "High levels of both the protein (significantly) and the transcript (not significantly) were seen to be associated with favourable prognosis (Figure 1G,H and S1B,C)."

**Referee

Cross-Commenting**

cross comment regarding referees 2 and 3 above. I'm am convinced that TXNIP is at least contemporaneously upregulated with GDF15 downregulation. However, the strong implication from the writing is that TXNIP regulates GDF15 directly. I agree with the comment above that exploring mechanisms may be open-ended especially as TXNIP has been implicated in gene regulation by several different mechanism. I'd be satisfied with a more open-minded discussion of potential mechanisms by which TXNIP may repress GDF15 and the possibility of other parallel pathways that likely contribute to GDF15 repression.

Many thanks, this is a generous and understanding approach. As described above we have carried out extra analysis and have found 6 differentially regulated transcription factors which have been shown to bind GDF15 promoter or enhancer regions with 1 of these, MYC, being significantly affected in the TXNIP knockout cell lines, which in combination with supportive literature suggests a degree of TXNIP dependence. We have also identified ARRDC4 as an additional regulator of GDF15 – again please see above.

Reviewer #1 (Significance (Required)):

This is an interesting contribution but the mechanistic connection between GDF15 and TXNIP is relatively weak. That said, even as independent variables they do seem to have utility in predicting therapeutic response.

Many thanks for the comment – we concur. We have reanalysed our data looking for relevant transcription factors (those that bind GDF15 promoter / enhancer regions) finding MYC as the most likely bridge. Please see above.

Reviewer #2 (Evidence, reproducibility and clarity (Required)):

The manuscript by Deng et al. investigates a mechanistic link between TXNIP and GDF15 expression and oxaliplatin treatment and acquired resistance. They observe an upregulation in TXNIP expression in the tumors of patients who have previously received chemotherapy. They demonstrate oxaliplatin-driven MondoA transcriptional activity is what underlies the induction of TXNIP. They further demonstrate that TXNIP is a negative regulator of GDF15 expression. Together, oxaliplatin induces MondoA activity and TXNIP expression, resulting in a downregulation of GDF15 expression and consequently decreased Treg differentiation.

Major Comments

1. The authors suggest that TXNIP induction and GDF15 downregulation are a common effect of chemotherapies; however, the mechanistic studies were limited to oxaliplatin. The authors should clarify this point through further investigation using other commonly used CRC chemotherapies (5-FU, irinotecan, etc.), or through textual changes. To be clear, I think that the oxaliplatin results could potentially stand on their own but would require additional clarification. For example, regarding the patient samples analyzed in 1D and 4F, which

patients received oxaliplatin? Could the analysis of publicly available molecular data be drilled down to just the patients who received oxaliplatin?

Many thanks – this is an excellent point. Firstly, all the patients in 1D and 4F received oxaliplatin. Secondly, we have included new data looking at the impact of other chemotherapies (FOLRIRI, FU-5 and SN-38) on aspects of the study, ultimately finding that these processes (especially an anti-correlation between GDF15 and TXNIP changes upon chemo treatment) appear to be specific to oxaliplatin. These data have been added (Figure S11) and throughout the emphasis has been switched from chemotherapeutic treatment to oxaliplatin treatment.

Figure S11. The described effects are specific for oxaliplatin. GDF15 (A) or TXNIP (B) expression in responders or non-responders to FOLFOX or FOLFIRI treatment. (C) GDF15/TXNIP expression ratio in FOLFOX or FOLFIRI treated responders or non-responders. (D) Receiver operating characteristic (ROC) curve showing area under the curve and p value for the use of GDF15/TXNIP ratio in predicting therapeutic response to FOLFIRI (Responder [n=9] non-responder [n=11]). E) Immunoblot analysis of TXNIP and GDF15 after 48h of 10 μ m 5-Fu treatment in DLD1 and HCT15 cell lines. (F) Immunoblot analysis of TXNIP and GDF15 expression after 48h of 10 μ m SN-38 treatment in DLD1 and HCT15 cell lines. Two-way ANOVA, multiple comparisons test. * $p<0.05$

Lines 796-799: “To check if the pre-treatment GDF15/TXNIP ratio could be used for patients treated with FOLFIRI we performed the same analyses finding no significance (S11A-D). This oxaliplatin specificity was then confirmed by western blot analysis in DLD1 and HCT15 cells treated with 5-FU or SN38 (Figure S11E-F).

Example of change of emphasis from ‘chemotherapy’ to ‘oxaliplatin’ – lines 139-142: “Here, in colorectal adenocarcinoma (CRC) we identify oxaliplatin-induced Thioredoxin Interacting Protein (TXNIP), a MondoA-dependent tumor suppressor gene, as a negative regulator of Growth/Differentiation Factor 15 (GDF15).”

2. The data demonstrating the induction of MondoA transcriptional activity and TXNIP expression in response to oxaliplatin treatment is quite convincing. The data regarding ROS induction of TXNIP is interesting, especially in light of other studies arguing that ROS limits MondoA activity (PMID: 25332233). Given this apparent disparity, I think that this study could really be strengthened by also investigating other potential mechanisms of oxaliplatin induction of MondoA. In particular, given many studies arguing for direct nutrient-regulation of MondoA, the authors should address the potential for oxaliplatin regulation of glucose availability and a potential glucose dependence of oxaliplatin-induced TXNIP.²

3. In line with the previous point, since MondoA activity and TXNIP expression are sensitive to glucose levels, the authors should investigate oxaliplatin-regulation of TXNIP under physiological glucose levels. No need to replicate everything, just key experiments.

We feel these are excellent point and really help the piece – many thanks. We have carried out assays around these points suggested and have included the findings in the new draft – see below.

A

B

C

Figure S3. The induction of TXNIP by oxaliplatin is dependent on glucose availability, uptake and oxidative phosphorylation. (A) Western blotting analysis of TXNIP expression in DLD1 cells or HCT15 cells treated with different concentrations of glucose with or without 10 μM oxaliplatin for 48h. β-actin was used as an internal reference. (B) DLD1 cells were treated with different concentration of oxaliplatin or vehicle (PBS) for 48h. After treatment, 2-NBDG staining and flow cytometry were used to detect glucose uptake. (C) Immunoblot analysis of TXNIP in DLD1 cells or HCT15 cells treated with Antimycin A (1 μg/mL), an OXPHOS inhibitor or oxaliplatin (10μm) or the combinational treatment for 48h.

Lines 332-339: "As such, we went back to first principles and assessed the impact of different concentrations of glucose on TXNIP induction +/- oxaliplatin treatment, finding a concentration dependent effect (Figure S3A). Intriguingly, high glucose alone was able to induce increased TXNIP expression. We then assessed if oxaliplatin treatment drove an increase in glucose uptake, with this seen at concentrations >10mM (Figure S3B). Next, to investigate the impact of glucose metabolism, and consequent ROS generation, on TXNIP induction we treated cells with Antimycin A, an inhibitor of oxidative phosphorylation, finding a complete block in oxaliplatin-induced TXNIP (Figure S3C)."

4. The authors did a good job of linking TXNIP and GDF15 in untreated conditions; however, the data arguing for oxaliplatin regulation of GDF15 through TXNIP is less clear. For example, in 3B-H, oxaliplatin treatment reduces GDF15 approximately to the same extent in the NTC and TKO cells, potentially in line with a mechanism of downregulation that doesn't involve TXNIP.

A very salient point and completely in line with the other reviewers. We have carried out a few additional analyses mentioned previously in this letter. The most pertinent for this specific point are the experiments around ARRDC4, where we found evidence to suggest that, like TXNIP, it regulates GDF15.

Minor Comments

1. The presentation of data in Figure 5 is confusing. A-B include raw cell numbers, whereas C-F show "normalized proliferation." What does this mean? And how was the normalization done?

Apologies for this. Legend text has been corrected to "Normalised proliferation (normalised to MFI from control: i.e. cells treated with supernatant from NTC cells) on gated CD3⁺CD8⁺ or CD3⁺CD4⁺ cells is shown. n=6. (G-H) Normalised IFN γ concentrations (normalised to MFI from control: i.e. cells treated with supernatant from NTC cells) in the supernatant of cells from C-F." (lines 727-729).

Referee Cross-Commenting

cross-comment regarding reviewer #1

I agree with the referee that the link between TXNIP and GDF15 is weak, though as I mentioned before, this is particularly true in the context of oxaliplatin-regulation of TXNIP. I agree that given all the presented data, it is likely that oxaliplatin-regulation of TXNIP and GDF15 are independent. In my opinion, the referee brought up all valid concerns, but this is by far the biggest concern that I share.

We agree that this is the weakest aspect of the paper, however our new analyses plus supportive literature, suggests that the relationship between TXNIP and GDF15 may be mediated by MYC (please see above)

cross-comment regarding reviewer #3

The major concern that this referee addresses is whether another transcription factor supersedes the proposed MondoA/TXNIP induction in regulating GDF15 expression in later stage CRC. In my opinion, this another other concerns of the referee are all valid, but still I remain unconvinced that TXNIP induction underlies the oxaliplatin-regulation of GDF15. I think fleshing out that aspect of the study would potentially help the authors tease apart how this potential MondoA-TXNIP-GDF15 axis is dysregulated later in CRC progression.

This is a great discussion. Interestingly enough, c-myc is seen at higher levels in late stage CRC (Hu X, Fatima S, Chen M, Huang T, Chen YW, Gong R, Wong HLX, Yu R, Song L, Kwan HY, Bian Z. Dihydroartemisinin is potential therapeutics for treating late-stage CRC by targeting the elevated c-Myc level. Cell Death Dis. 2021 Nov 5;12(11):1053. Doi: 10.1038/s41419-021-04247-w. PMID: 34741022; PMCID: PMC8571272.), is seen as an important factor in resistance, and as this review argues, is driven by stress (Saeed H, Leibowitz BJ, Zhang L, Yu J. Targeting Myc-driven stress addiction in colorectal cancer. Drug Resist Updat. 2023 Jul;69:100963. Doi: 10.1016/j.drup.2023.100963. Epub 2023 Apr 20. PMID: 37119690; PMCID: PMC10330748.). So it is very plausible that the partial TXNIP-mediated regulation of myc in early / sensitive CRCs that we may be observing, and has been reported recently ([TXNIP loss expands Myc-dependent transcriptional programs by increasing Myc genomic binding](https://doi.org/10.1371/journal.pbio.3001778) Lim TY, Wilde BR, Thomas ML, Murphy KE, Vahrenkamp JM, et al. (2023) TXNIP loss expands Myc-dependent transcriptional programs by increasing Myc genomic binding. PLOS Biology 21(3): e3001778. <https://doi.org/10.1371/journal.pbio.3001778>) is lost in late stage / resistant CRCs. If this is the case, in effect what we would have observed is the loss of a stress-associated method (TXNIP) of controlling c-myc activity. What makes our collective lives difficult is that, as reported “this expansion of Myc-dependent transcription following TXNIP loss occurs without an apparent increase in Myc’s intrinsic capacity to activate transcription and without increasing Myc levels.” ([TXNIP loss expands Myc-dependent transcriptional programs by increasing Myc genomic binding](https://doi.org/10.1371/journal.pbio.3001778) Lim TY, Wilde BR, Thomas ML, Murphy KE, Vahrenkamp JM, et al. (2023) TXNIP loss expands Myc-dependent transcriptional programs by increasing Myc genomic binding. PLOS Biology 21(3): e3001778. <https://doi.org/10.1371/journal.pbio.3001778>)

Reviewer #2 (Significance (Required)):

Generally speaking the experiments are well controlled and the findings are significant and novel. Though the link between MondoA activity and ROS could be strengthened, and the data could be validated under more physiological settings. Further, the authors should clarify their interpretations so as to not overstate the findings.

Many thanks for the comments. We have taken onboard the need for more physiological settings and have included varying levels of glucose to reflect concentrations in different environments. We have repeated the siMondoA work in 3J strengthening the conclusions wrt its impact on TXNIP and GDF15 expression (see above).

Reviewer #3 (Evidence, reproducibility and clarity (Required)):

In this well-written manuscript, the authors show that chemotherapy increases a MondoA-

dependent oxidative stress-associated protein, TXNIP, in chemotherapy-responsive colorectal cancer cells. They show that TXNIP negatively regulates GDF-15 expression. GDF-15, in turn, correlates with the presence of T cells (Treg), and inhibits CD4 and CD8 T cell stimulation. In advanced disease and chemo-resistant cancers, upregulation of TXNIP and downregulation of GDF-15 appear to get lost. Based on a somewhat smallish data set, the authors suggest that the pre-treatment GDF-15/TXNIP ratio can predict responses to oxaliplatin treatment. This is a very interesting, novel finding. In general, the quality of the experiments and the data are high and the conclusions appear sound. Still, there are a number of aspects that should still be improved:

The observed loss of the ROS - MondoA - TXNIP - GDF15 axis in chemoresistant and/or metastatic tumors implies that another transcription factor or pathway becomes dominant upon tumor progression. As this switch would be key to better understanding the mechanism underlying the prognostic role of the TXNIP/GDF15 ratio, the authors should at least do data mining followed by ChEA or Encode (or other) analysis to identify transcription factors or pathways that become activated in late-stage/metastatic CRC cells. There is a high likelihood that a transcription factor or pathway involved in GDF-15 upregulation in cancer (e.g. p53, HIF1alpha, Nrf2, NF-kB, MTF, C/EBP β , BRAF, PI3K/AKT, MAPK p38, EGR1) supersedes the inhibitory effect of the MondoA-TXNIP axis. As it stands, the proposed loss of function of the ROS - MondoA - TXNIP - GDF-15 axis is far less convincing than almost all other aspects of the study.

An extremely fair point. We adopted a similar approach to that suggested – as mentioned above, we looked at TFs that bind to GDF15 promoter/enhancer regions and then looked at the presence of these in our transcriptomic data – specifically any evidence of change post oxaliplatin treatment. We found 6 such TFs that were decreased post-oxaliplatin treatment. We then looked for any evidence of TXNIP dependence in these TFs by comparing post-oxaliplatin treatment across NTC and TXNIP knockout lines, when we did this we found only one GDF15 promoter/enhancer binding TF was significantly changed: MYC. We then looked at the relationship between MYC, TXNIP, and GDF15 against the other 5 ‘control’ TFs in the TCGA COAD dataset, we found that MYC showed the strongest correlations, in the ‘correct’ directions. This finding was further backed up in the literature where a TXNIP knockout in a breast cancer model drove c-myc-dependent transcription, whilst c-myc has been observed to increase in later stage CRC patients, is associated with cellular stress and resistance. The collective evidence therefore suggests that MYC is the factor that is initially at least partially regulated by TXNIP, before this regulation is lost in advanced / resistant disease. Continuing on this line, it is likely that the predictive GDF15/TXNIP ratio is at least in part, a measure of c-myc responsiveness to oxaliplatin. All the while we must bear in mind TXNIP-independent oxaliplatin-dependent regulation of GDF15, most likely ARRDC4, as described earlier in this document.

Using pathway analysis software to compare our transcriptomic data from cell lines treated with/without oxaliplatin, the most likely pathways upstream of MYC/c-myc that are negatively affected by chemotherapy are BAG2, Endothelin-1, telomerase, ErbB2-ErbB3 and Wnt/B-catenin. When looking at the comparison of UTC and resistant lines’ transcripts there is only one key component of these pathways which is upregulated in both lines - ERBB3 –

which has already been shown to be important in CRC metastasis and resistance (Desai O, Wang R. HER3- A key survival pathway and an emerging therapeutic target in metastatic colorectal cancer and pancreatic ductal adenocarcinoma. *Oncotarget*. 2023 May 10;14:439-443. doi: 10.18632/oncotarget.28421. PMID: 37163206; PMCID: PMC10171365.). It is highly speculative, but our data suggests the most likely pathway to supersede TXNIP in its (partial) regulation of MYC is the ErbB2-ErbB3 pathway.

My further criticisms are mostly more technical:

Figure 2 I-L: What was the extent of MondoA downregulation achieved by siRNA treatment? Could the effects also be seen with the small molecule mondoA inhibitor SBI-477 (or a related substance)?

This experiment has been repeated. The pooled densitometric data is also now given (please see above).

How do you explain the different GDF-15 levels between untreated non-target control cells (NTC) and TXNIP knock-down cells (TKO) in Figures 3C-F?

The only way to interpret this is that there is a TXNIP-independent pathway regulating GDF15 expression after oxaliplatin treatment, as described this is most likely to be ARRDC4 - the text has been updated to:

Lines 522-524: "It is important to note, however, that we saw clear evidence that *TXNIP* was not solely responsible for the downregulation of GDF15 post oxaliplatin treatment (Figure 3C-G, S6E)."

In figures 3 E-G the dots for the individual measurements should be indicated. This would be more informative than just the bar graphs.

Completed.

Figure 4C,D and Table 3: Data on the role of GDF-15 in CRC are largely valedictory of previous work (e.g. Brown et al. *Clin Cancer Res* 2003, 9(7):2642-2650, Wallin et al., *Br J Cancer*. 2011 May, 10;104(10):1619-27). Therefore, the previous studies should be cited.

Apologies for the oversight and many thanks – this is an excellent addition.

Figure 5C-F: Please indicate in the figure legend how proliferation was assessed.

Many thanks. This was noticed by another reviewer also. We have changed the text to include how the data was normalised: "(C-F) Labelled PBMCs were stimulated with anti-CD3 and anti-CD28 for 4 days in the presence of fresh supernatant from indicated cell lines, before being stained with anti-CD3 and anti-CD8 (C-D) or anti-CD4 (E-F) antibodies and measured by flow cytometry. Normalised proliferation (normalised to MFI from control: i.e. cells treated with supernatant from NTC cells) on gated CD3⁺CD8⁺ or CD3⁺CD4⁺ cells is shown. n=6. (G-H) Normalised IFN γ concentrations (normalised to MFI from control: i.e. cells

treated with supernatant from NTC cells) in the supernatant of cells from C-F." (lines 724-730)

Figure S8E-G: Please indicate the analysed parameters in the graphs. In Figure S8G, the legend just indicates that "aggression of tumour" is dichotomized and plotted. This clearly requires a better definition.

Many thanks, this has been changed as per the below.

Lines 862-868: "(E-G) Receiver operating characteristic (ROC) curves showing area under the curve and p values for the use of GDF15/TXNIP ratio in predicting origin of cell line (E; primary; DLD1, HCT15, HT29, SW48 [n=4] or secondary; DiFi, LIM1215 [n=2]), sensitivity to oxaliplatin (F; parental DLD1 (plus biological repeat), HCT15 [n=3] or resistant DLD1 (plus biological repeat), HCT15 [n=3]), aggression of tumor (G; non-aggressive; <T4M1. Patients 1, 4, 5, 7, 8, 9, 11 [n=7] or aggressive; ≥ T4M1. Patients 2, 3, 6 and 10 [n=4]) Please see supplementary table 8 for detailed clinical information for G."

The authors propose a novel ROS - MondoA - TXNIP - GDF15 - Treg axis, where MondoA activation, TXNIP up- and GDF-15 downregulation enhance tumor immunogenicity. While this axis has been analyzed in some detail, GDF-15 is not only linked to induction of regulatory T cells. There has been a report showing that GDF-15/MIC-1 expression in colorectal cancer correlates with the absence of immune cell infiltration (Brown et al. Clin Cancer Res 2003, 9(7):2642-2650). The link between GDF-15 and immune cell exclusion has also been confirmed in other conditions, including different cancers (Kempf et al. Nat Med 2011, 17(5):581-588, Roth P et al. Clin Cancer Res 2010, 16(15):3851-3859, Haake et al. Nat Commun 2023, 14(1):4253). A key mechanism is the GDF-15 mediated inhibition of LFA-1 activation on immune cells. As the authors argue that the described pathways turns cold tumors hot in response to oxaliplatin-based chemotherapy, this GDF-15 dependent immune cell exclusion mechanism might be at least as relevant than induction of Treg. Likewise, inhibition of dendritic cell maturation by GDF-15 (Zhou et al. PLoS One 2013, 8(11):e78618) could explain why GDF-15high tumors are immunologically cold. Reviewed in ³

The authors propose that the pathways discovered by them contributed to the "heating up" of the tumor microenvironment after oxaliplatin-based chemotherapy. The authors should thus look in their data sets for the presence of cytotoxic T cells and their possible correlation with TXNIP and GDF-15 levels.

This is a wonderful explanation – many thanks. We have taken the opportunity to assess the impact of GDF15 expression on a variety of T cell markers (Figure S9). In this data a negative association between GDF15 and CD8 CTLs can clearly be seen, as predicted by the reviewer.

Figure S9. Low GDF15 expressing tumors express more activated cytotoxic CD8 T cell transcripts. Analysis of TCGA COAD database. Comparative analysis of CD8A (A), CD4 (B), CD69 (C), IL2RA (D), CD28 (E), IFNG (F), PRF1 (G), GZMA (H), GZMK (I), FOXP3 (J), TBX21 (K), EOMES (L), IRF4 (M), RORC (N) and GATA3 (O) transcript expression between high GDF15 tumors and low GDF15 tumors. Normalised expression = transcript expression normalised to the mean of (CD3D+ CD3E)/2. GDF15 high and low groups were defined as the top and bottom quartiles when cases were ranked by GDF15 expression. Two-tailed Student's t test; *p<0.05, **p<0.01, ***p < 0.001, ****p < 0.0001.

Lines 712-717: "To assess if the *GDF15*-dependent presence of Tregs may be associated with a decrease in activated cytotoxic CD8 T cells, we interrogated the TCGA COAD dataset. We found that low *GDF15* tumors carried significantly higher levels of *CD8*, *CD69*, *IL2RA*, *CD28*, *PRF1*, *GZMA*, *GZMK*, *TBX21*, *EOMES* and *IRF4* (Figure S9); transcripts indicative of activated cytotoxic CD8 T cells. High *GDF15* tumors were enrichment for *FOXP3* and, interestingly, *RORC* (Figure S9). These data support the hypothesis that *GDF15* induces *Foxp3*^{+ve} Tregs which inhibit CD8 T cell proliferation and activation in the TME."

The paragraph on GDF-15 receptors needs to be corrected: The purported role of a type 2 transforming growth factor (TGF)-beta receptor in GDF-15 signalling had been due to a frequent contamination of recombinant GDF-15 with TGF-beta (Olsen et al. PLoS One 2017,

12(11):e0187349). There have been a number of screenings for GDF-15 receptors that have all failed to show an interaction between GDF-15 and TGF-beta receptors. Instead, only GFRAL was found in these large-scale screenings (Emmerson et al. Nat Med 2017, 23(10):1215-1219, Hsu et al. Nature 2017, 550(7675):255-259, Mullican et al. Nat Med 2017, 23(10):1150-1157, Yang et al. Nat Med 2017, 23(10):1158-1166). The one subsequent report that shows a link between GDF-15, engagement of CD48 on T cells and induction of a regulatory phenotype (Wang et al. J Immunother Cancer 2021, 9(9)) still awaits independent validation. Considering that CD48 lacks an intracellular signaling domain that would be critical for a classical receptor function, I recommend to be more cautious regarding the role of CD48 as GDF-15 receptor. Given the mechanism outlined by Wang et al. the word interaction partner might be more apt. Moreover, an anti-GDF-15 antibody would be a good control for the experiments involving an anti-CD48 antibody in Figure 5.

Thank you so much for this concise and highly informative paragraph. We have changed the text to read:

202-204: "As a soluble protein, GDF15 exerts its effects by binding to its cognate receptor, GDNF-family receptor a-like (GFRAL)^{44,45,46,47} or interaction partner, CD48 receptor (SLAMF2)⁴³, with the latter still requiring additional verification."

We would have ideally included an anti-GDF15 antibody in the CD48 assay at the time but didn't have the foresight. We have included the additional text to temper any conclusions.

Lines 701-711: "Furthermore, when stimulating naïve CD4 T cells in the presence of GDF15 enriched supernatant we were able to both differentiate these cells into functional Tregs and also block the generation of this functionality using an anti-CD48 antibody (Figure 5M-N). However, it must be stressed that the binding and functional impacts of GDF15's interaction with CD48 still require further verification."

Cell surface externalization of annexin A1 has been described as a failsafe mechanism to prevent inflammatory responses during secondary necrosis (PMID: 20007579). Thus, I am surprised that the authors list annexin A1 among the immune-stimulatory molecules exposed or released in response to chemotherapy-induced cell death (line 103). Please clarify!

We agree – it shouldn't be there!! Removed. Many thanks.

**Referee

Cross-Commenting**

Regarding the cross-comment by referee 2: In my opinion, the data shown in Figure 3C-H clearly demonstrates that TXNIP can repress GDF-15 expression. I agree that there will likely be further regulators. The GDF-15 promoter is constantly regulated by a multitude of factors (which mostly induce transcription). As downregulation of GDF-15 in response to oxaliplatin is the opposite of the frequently described induction of GDF-15 upon chemotherapy, net effects may always be "smudged" by contributions from different pathways (e.g. by cell stress due to siRNA transfection).

Therefore, I believe that the data are as good as it will get. Accordingly, I would not force the authors to further amplify the observed effect.

Many thanks for your understanding – yes, GDF15 has >650 TFs that bind its promoter/enhancer regions – a number we found rather daunting. Happily your comments and those of the other reviewers inspired us to dig and we now have data that is supportive of MYC's and ARRDC4's involvement – detailed throughout this reply.

cross comment regarding referee #1:
I share the general assessment of the referee and recognize the very detailed mechanistic analysis. To further support the moderate effects of the MondoA knockdown, a small molecule inhibitor like SBI-477 might be useful. (I had already suggested using this inhibitor to support these data.)

Many thanks for the suggestion. We opted to increase the number of siRNA repeats instead – with the data included in Figure 3J (above).

Still, my view on the potential relevance of oxaliplatin-induced, TXNIP-independent downregulation of GDF-15 differs from that of referee 1. In the clinics, platinum-based chemotherapy is one of the strongest inducers of GDF-15 (compare Breen et al. GDF-15 Neutralization Alleviates Platinum-Based Chemotherapy-Induced Emesis, Anorexia, and Weight Loss in Mice and Nonhuman Primates. Cell Metabolism 32(6), P938-950, 2020.DOI:<https://doi.org/10.1016/j.cmet.2020.10.023>). I was thus surprised that the authors found a pathway, which leads to an outcome that an exactly opposite effect.

This is fascinating that oxaliplatin drives this increase in GDF15 – we were unaware of this paper. Looking at figure 2(H-K), GDF15 is being produced from multiple non-diseased tissues after systemic chemotherapy – even at day 19 post-treatment – this suggests that wrt this study, systemic GDF15 could not be used as a readout of success or otherwise – which is extremely helpful! Thank you.

Thus far, the only obvious reason for reduced GDF-15 secretion upon treatment with cytotoxic drugs was a reduction in tumor cell number due to cytotoxicity.

Please do not discount this. This study was focused on the cells which survived oxaliplatin treatment – the cells which did not were discarded. Our view, given your input, would be a complex picture where in early stages systemic GDF15 goes up, due to off-target effects, but locally levels drop owing to cell death and this, and other, stress-related pathways in the remaining tumor cells.

Still, the authors managed to convince me that the described pathway (ROS - MondoA - TXNIP - GDF-15) exists. (Here, I still largely concur with referee 1.) Moreover, as we have identified some factors required for GDF-15 biosynthesis that could easily interact with TXNIP, I find the proposed mechanism plausible.

Extremely encouraging for us to hear!

Nevertheless, as a downregulation of GDF-15 in response to chemotherapy is hardly ever observed in late-stage cancers, I believe that the observed switch in pathway activation between early- and late-stage cancers might be highly relevant - in particular, as there is so much evidence for platinum-based induction of GDF-15 in late-stage cancer patients. Emphasizing the divergent clinical observations (e.g. by Breen et al.) could thus help to put the finding into perspective.

Very much agree. We did see this phenomenon in LIM1215 cells (Figure 6B) and the resistant lines we generated continually produced higher levels.

Analysing TXNIP-independent mechanisms involved in the oxaliplatin-dependent repression of GDF-15, as suggested by referee #1, will require enormous efforts and resources, and may still turn out to be fruitless. Personally, I would thus be content if the authors just mentioned possible contributions from other pathways upon cancer progression. To me, the described pathway seems to be limited to early-stage cancers, and the actual finding that GDF-15 is downregulated is an interesting observation, irrespective of further involved pathways.

Many thanks – this is extremely fair. Happily we have managed to make some tentative steps forward in highlighting the potential role of MYC, and the suggestion of redundancy wrt ARRDC4, but as you say, much more work needs to be done to fully understand these processes.

cross comment regarding referee #2:
I fully agree with the referee that activation of the pathway by further chemotherapeutic drugs could be a valuable addition. As Guido Kroemer's lab has described oxaliplatin to induce a more immunogenic cell death compared to other platinum-based chemotherapies, even a rather limited comparison between oxaliplatin and cisplatin could be very interesting.

Absolutely agree – extra data on this has been included in Figure S11, which is included earlier in this letter. We also uncovered a meta-analysis using metformin, which has been seen to inhibit ROS, where TXNIP and ARRDC4 are the top two downregulated transcripts whilst GDF15 appears in the top four upregulated. This may suggest that chemotherapeutic immunogenicity, at least through the presence or absence of GDF15, may in part be driven by ROS.

Lines 930-933: "Further support for both TXNIP and ARRDC4's role in regulating GDF15 after the induction of ROS comes from a pan cancer meta-analysis assessing the impact of metformin (which has been reported to inhibit ROS) on gene expression. Here the top two downregulated genes were *TXNIP* and *ARRDC4* and the top four upregulated genes were *DDIT4*, *CHD2*, *ERN1* and *GDF15*⁷² "

Reviewer #3 (Significance (Required)):

In general, this is a very interesting manuscript describing a cascade of events that may contribute to successful chemotherapy (which likely requires induction of an immune response against dying tumor cells.) The observation that this pathway is only active in

early/non-metastatic cancer cells is striking. Unfortunately, the authors cannot explain inactivation of this pathway in later stage/ metastatic/ highly aggressive cancers. Understanding this switch could easily be the most important finding triggered by this report. Therefore, I highly recommend to make some effort in this direction. Strikingly, the authors find that disruption of TXNIP-mediated GDF-15 downregulation is strongly associated with worse prognosis. They also suggest that this ratio could indicate whether a patient will respond to oxaliplatin-based chemotherapy.

This is again very fair – we have posited a potential mechanism for the loss of this switch elsewhere in this reply– one which involves a change in TXNIP-mediated MYC regulation and/or increased HER2-HER3 signalling – but although reasonable for a rebuttal (and publication in that context) we do not feel we have the evidence to include this within the full manuscript.

Altogether, the findings described in manuscript are very novel and may have prognostic (or, in case of the presumed loss of the MondoA - TXNIP - GDF-15 pathway) therapeutic implications. Thus, the manuscript certainly fills various gaps and should be of major interest for cell biologists working on immunogenic cell death, or colorectal cancer, or MondoA, TXNIP or GDF-15. Still, due to its translational implications, it would also be worthwhile reading for a large number of researchers in the oncology field.

We are very grateful for your kind comments.

1 Sinclair, L. V., Barthelemy, C. & Cantrell, D. A. Single Cell Glucose Uptake Assays: A Cautionary Tale. *Immunometabolism* **2**, e200029, doi:10.20900/immunometab20200029 (2020).

2 Yu, F. X., Chai, T. F., He, H., Hagen, T. & Luo, Y. Thioredoxin-interacting protein (Txnip) gene expression: sensing oxidative phosphorylation status and glycolytic rate. *J Biol Chem* **285**, 25822-25830, doi:10.1074/jbc.M110.108290 (2010).

3 Wischhusen, J., Melero, I. & Fridman, W. H. Growth/Differentiation Factor-15 (GDF-15): From Biomarker to Novel Targetable Immune Checkpoint. *Front Immunol* **11**, 951, doi:10.3389/fimmu.2020.00951 (2020).

With best regards,

Tony and Richard

19th Feb 2024

Dear Dr. Beatson,

Thank you for the submission of your revised manuscript to EMBO Molecular Medicine. I am pleased to inform you that we will be able to accept your manuscript pending the following final amendments:

- 1) Please address referees' minor concerns.
- 2) Title: Please revise the title. Phrase it as a concluding sentence using active voice. Title should emphasize the main message of the manuscript considering the scope of the journal and should contain all relevant information e.g., cancer type. For example, "Ratio between TXNIP and GDF15 predicts oxaliplatin response of colorectal adenocarcinoma".
- 3) Figures:
 - Please remove all figures from the manuscript and upload individual, high-resolution file for each main and EV figures. 5 Suppl. Figures should be made Figure EV 1-5. Main figure legends and EV figure legends should be placed at the end of the manuscript file. Rest of Suppl. Figures should be moved to "Appendix" - combined in a single PDF file, legend removed from manuscript text and added underneath the figure, and the figure renamed "Appendix Figure S1". Please also update all figure callouts in the main text. For more information on figure presentation please check "Author Guidelines". <https://www.embopress.org/page/journal/17574684/authorguide#datapresentationformat>
 - Figures: We note that some images/panels are reused. Figure 1A (H&E) is reused in Figure 4A and Figure 1E (TXNIP) in Figure 4E. Please cite in the respective figure legend every reused image/panel.
- 4) Author checklist: Please submit a complete checklist. <https://www.embopress.org/pb-assets/embo-site/EMBO%20Press%20Author%20Checklist-1642513524327.xlsx>
- 5) In the main manuscript file, please do the following:
 - Please address all comments suggested by our data editors listed below:
 - o Figure legends:
 1. Please note that a separate 'Data Information' section is required in the legends of figures 1b-c, e, g-h; 2a-b, i-j, m; 3a, e-g, 4b, g; 5a, c-j, m-o; 6c, h, k, m-o.
 2. Please indicate the statistical test used for data analysis in the legends of figures 4b-d, g-i; 5a-j, m-o; 6l.
 3. Please note that the box plot needs to be defined in terms of minima, maxima, centre, bounds of box and whiskers, and percentile in the legend of figure 1c.
 4. Please note that information related to n is missing in the legends of figures 1e; 4g; 5g-h; 6c, h, k, m-o.
 5. Although 'n' is provided, please describe the nature of entity for 'n' in the legends of figures 5a-f, i-j, m-o.
 6. Please note that the error bars are not defined in the legends of figures 6c, h, m-n.
 - Limit keywords to max 5.
 - Remove "Statement of Significance" and "Conflicts of interests" from the title page.
 - Add callouts for Fig 5O.
 - Provide the antibody dilutions that were used for each antibody.
 - Rename "Competing interests" to "Disclosure and competing interests statement". We updated our journal's competing interests policy in January 2022 and request authors to consider both actual and perceived competing interests. Please review the policy <https://www.embopress.org/competing-interests> and update your competing interests if necessary.
 - Author contributions: Please remove it from the manuscript and specify author contributions in our submission system. CRediT has replaced the traditional author contributions section because it offers a systematic machine-readable author contributions format that allows for more effective research assessment. You are encouraged to use the free text boxes beneath each contributing author's name to add specific details on the author's contribution. More information is available in our guide to authors: <https://www.embopress.org/page/journal/17574684/authorguide#authorshipguidelines>
 - Please move "Ethics approval and consent to participate" and "Consent for publication" to "Human samples" paragraph.
 - Also, please confirm that in addition to the WMA Declaration of Helsinki the experiments also conformed to the principles set out in the Department of Health and Human Services Belmont Report.
 - Rename "Acknowledgements and Funding" to "Acknowledgements".
 - Remove DOI from references.
 - Please rename "Availability of data and material" to "Data Availability". Remove the sentence "Other datasets generated during and/or analysed during the current study are available from the corresponding author on reasonable request." Please be aware that all deposited data should be made freely available upon publication. Use the following format to report the accession number of your RNA Seq data:

The datasets produced in this study are available in the following databases:
[data type]: [full name of the resource] [accession number/identifier] ([doi or URL or identifiers.org/DATABASE:ACCESSION])

Please check "Author Guidelines" for more information.

<https://www.embopress.org/page/journal/17574684/authorguide#availabilityofpublishedmaterial>

6) Tables: Please move all tables to the "Appendix" and rename them to "Appendix Table S1" etc. If a table is too large, upload it

as Dataset EV1 etc. Please also update all table callouts in the main text. Callouts should be in sequential order.

7) Appendix: Please upload it as a single PDF file and add table of content with page numbers on the title page.

8) Funding: Please make sure that information about all sources of funding are complete in both our submission system and in the manuscript. Currently, project grant, C1519/A27375), R/R000026/1 and UCLH/UCL BRC, EU IMI2 IMMUCAN (Grant agreement number 821558) are missing in our submission system.

9) The Paper Explained: Please provide "The Paper Explained" and add it to the main manuscript text. Please check "Author Guidelines" for more information. <https://www.embopress.org/page/journal/17574684/authorguide#researcharticleguide>

10) Synopsis: Every published paper now includes a 'Synopsis' to further enhance discoverability. Synopses are displayed on the journal webpage and are freely accessible to all readers. They include separate synopsis image and synopsis text.

- Synopsis image: Please provide a striking image or visual abstract as a high-resolution jpeg file 550 px-wide x (250-400)-px high to illustrate your article.

- Synopsis text: Please provide a short standfirst (maximum of 300 characters, including space) as well as 2-5 one sentence bullet points that summarise the paper as a .doc file. Please write the bullet points to summarise the key NEW findings. They should be designed to be complementary to the abstract - i.e. not repeat the same text. We encourage inclusion of key acronyms and quantitative information (maximum of 30 words / bullet point). Please use the passive voice.

11) For more information: This space should be used to list relevant web links for further consultation by our readers. Could you identify some relevant ones and provide such information as well? Some examples are patient associations, relevant databases, OMIM/proteins/genes links, author's websites, etc...

12) As part of the EMBO Publications transparent editorial process initiative (see our Editorial at <http://embomolmed.embopress.org/content/2/9/329>), EMBO Molecular Medicine will publish online a Review Process File (RPF) to accompany accepted manuscripts. This file will be published in conjunction with your paper and will include the anonymous referee reports, your point-by-point response and all pertinent correspondence relating to the manuscript. Let us know whether you agree with the publication of the RPF and as here, if you want to remove or not any figures from it prior to publication. Please note that the Authors checklist will be published at the end of the RPF.

13) Please provide a point-by-point letter INCLUDING my comments as well as the reviewer's reports and your detailed responses (as Word file).

I look forward to reading a new revised version of your manuscript as soon as possible.

Yours sincerely,

Zeljko Durdevic

*** Instructions to submit your revised manuscript ***

1) a .docx formatted version of the manuscript text (including Figure legends and tables)

2) Separate figure files*

3) supplemental information as Expanded View and/or Appendix. Please carefully check the authors guidelines for formatting Expanded view and Appendix figures and tables at <https://www.embopress.org/page/journal/17574684/authorguide#expandedview>

4) a letter INCLUDING the reviewer's reports and your detailed responses to their comments (as Word file).

5) The paper explained: EMBO Molecular Medicine articles are accompanied by a summary of the articles to emphasize the major findings in the paper and their medical implications for the non-specialist reader. Please provide a draft summary of your article highlighting

6) For more information: There is space at the end of each article to list relevant web links for further consultation by our readers. Could you identify some relevant ones and provide such information as well? Some examples are patient associations, relevant databases, OMIM/proteins/genes links, author's websites, etc...

7) Author contributions: the contribution of every author must be detailed in a separate section.

8) EMBO Molecular Medicine now requires a complete author checklist (<https://www.embopress.org/page/journal/17574684/authorguide>) to be submitted with all revised manuscripts. Please use the checklist as guideline for the sort of information we need WITHIN the manuscript. The checklist should only be filled with page numbers where the information can be found. This is particularly important for animal reporting, antibody dilutions (missing) and exact values and n that should be indicated instead of a range.

9) Every published paper now includes a 'Synopsis' to further enhance discoverability. Synopses are displayed on the journal webpage and are freely accessible to all readers. They include a short stand first (maximum of 300 characters, including space) as well as 2-5 one sentence bullet points that summarise the paper. Please write the bullet points to summarise the key NEW findings. They should be designed to be complementary to the abstract - i.e. not repeat the same text. We encourage inclusion of key acronyms and quantitative information (maximum of 30 words / bullet point). Please use the passive voice. Please attach these in a separate file or send them by email, we will incorporate them accordingly.

You are also welcome to suggest a striking image or visual abstract to illustrate your article. If you do please provide a jpeg file 550 px-wide x 300-800px high.

10) A Conflict of Interest statement should be provided in the main text

11) Please note that we now mandate that all corresponding authors list an ORCID digital identifier. This takes <90 seconds to complete. We encourage all authors to supply an ORCID identifier, which will be linked to their name for unambiguous name identification.

Currently, our records indicate that the ORCID for your account is 0000-0002-0411-932X.

Link Not Available

- Graphs 800-1,200 DPI
- Photos 400-800 DPI
- Colour (only CMYK) 300-400 DPI"

*Additional important information regarding figures and illustrations can be found at <https://bit.ly/EMBOPressFigurePreparationGuideline>. See also figure legend preparation guidelines:

**** Reviewer's comments ****

Referee #1 (Comments on Novelty/Model System for Author):

this is a comprehensive reply to my concerns. They have been adequately addressed with no additional issues remaining to be addressed.

Referee #1 (Remarks for Author):

this is good to go. I appreciate the thoughtful and in-depth rebuttal.

Referee #2 (Remarks for Author):

The authors have thoroughly addressed the issues raised by the referees. The identification of c-myc as a regulator of TXNIP and thus, indirectly, of GDF-15 is an interesting additional finding. Remarkably, both myc-driven and GDF-15 overexpressing tumors are poorly infiltrated by T cells, which supports the newly found link. This has interesting implications as GDF-15 is a druggable target for cancer immunotherapy (<https://dailyreporter.esmo.org/esmo-immuno-oncology-congress-2023/esmo-immuno-oncology-2023/blocking-gdf-15-may-be-promising-to-overcome-resistance-to-immunotherapy>), whereas a direct targeting of Myc remains challenging.

The second remarkable new finding is the different behaviour of oxaliplatin compared to other chemotherapeutics. It would be interesting to know whether this is linked to the purported immunogenicity of oxaliplatin-induced cell death.

Nevertheless, it is impossible to address all questions in one single publication. Still, these implications may warrant future follow-up studies.

Referee #3 (Remarks for Author):

The authors largely satisfied my concerns; however, I think that the paper could benefit from more discussion of the points they made with the additional supplemental figure 3. The additional data could suggest a potential role for changes in glucose as a mechanism underlying oxa-induced MondoA activity and TXNIP expression. No need to perform additional experiments to address this model, but an acknowledgement of potential mechanisms outside of ROS driving MondoA activity would be appropriate.

1. Their data indicates that oxa treatment alone increases glucose uptake, and as they and others have shown, TXNIP expression is extremely responsive to glucose levels in the cell.
2. The authors use Antimycin A treatment as a way of dysregulating ROS; however, as an ETC inhibitor, the reduction in ETC-derived mitochondrial ATP will also reduce MondoA activity and TXNIP expression (PMID:30717828).
3. Though a bit perplexing is the lack of TXNIP induction by oxa at physiological glucose levels in DLD1 cells.

Rev_Com_number: RC-2023-02124

New_manu_number: EMM-2024-19383

Corr_author: Beatson

Title: Chemotherapeutic regulation of the ROS/MondoA-dependent TXNIP/GDF15 axis; and derivation of a new organoid metric as a predictive biomarker

Dear Dr Durdevic,

Many thanks for all your help in facilitating this final version. Please find below the point-by-point letter including your comments as well as the reviewers as requested. The comments are highlighted in bold.

1) Please address referees' minor concerns.

Referee #1 (Comments on Novelty/Model System for Author):

this is a comprehensive reply to my concerns. They have been adequately addressed with no additional issues remaining to be addressed.

Referee #1 (Remarks for Author):

this is good to go. I appreciate the thoughtful and in-depth rebuttal.

Many thanks – we really do appreciate your input – the paper is greatly improved. Wishing you all the best for the future.

Referee #2 (Remarks for Author):

The authors have thoroughly addressed the issues raised by the referees. The identification of c-myc as a regulator of TXNIP and thus, indirectly, of GDF-15 is an interesting additional finding. Remarkably, both myc-driven and GDF-15 overexpressing tumors are poorly infiltrated by T cells, which supports the newly found link. This has interesting implications as GDF-15 is a druggable target for cancer immunotherapy (<https://dailyreporter.esmo.org/esmo-immuno-oncology-congress-2023/esmo-immuno-oncology-2023/blocking-gdf-15-may-be-promising-to-overcome-resistance-to-immunotherapy>), whereas a direct targeting of Myc remains challenging.

Many thanks for pointing us in this direction – it really does seem as if there is something here with TXNIP, Myc and GDF15 – hopefully this will be followed up on. As you say, there is a fair bit of interest in targeting GDF15 – hopefully this paper can aid in stratifying (at an early stage) in the future.

The second remarkable new finding is the different behaviour of oxaliplatin compared to other chemotherapeutics. It would be interesting to know whether this is linked to the purported immunogenicity of oxaliplatin-induced cell death. Nevertheless, it is impossible to address all questions in one single publication. Still, these implications may warrant future follow-up studies.

Completely agree – it does add further weight to the idea that choice of combinational chemotherapy is critical in getting the most out of any particular immunotherapy. Thank you so much for all your input and discussion – we have thoroughly enjoyed the process. All the best with your work.

Referee #3 (Remarks for Author):

The authors largely satisfied my concerns; however, I think that the paper could benefit from more discussion of the points they made with the additional supplemental figure 3. The additional data could suggest a potential role for changes in glucose as a mechanism underlying oxa-induced MondoA activity and TXNIP expression. No need to perform additional experiments to address this model, but an acknowledgement of potential mechanisms outside of ROS driving MondoA activity would be appropriate.

1. Their data indicates that oxa treatment alone increases glucose uptake, and as they and others have shown, TXNIP expression is extremely responsive to glucose levels in the cell.
2. The authors use Antimycin A treatment as a way of dysregulating ROS; however, as an ETC inhibitor, the reduction in ETC-derived mitochondrial ATP will also reduce MondoA activity and TXNIP expression (PMID:30717828).
3. Though a bit perplexing is the lack of TXNIP induction by oxa at physiological glucose levels in DLD1 cells.

Many thanks for your comments.

1. Yes, we were surprised by these assays – the sensitivity of TXNIP to glucose in these lines is really quite remarkable.

2. Thank you for your input ~~wrt~~with respect to Antimycin A – we have added the following text in the results section when describing these data:

“However we cannot state whether this finding is driven by a decrease in ROS or by the inhibitor’s effect on ATP-mediated activation of MondoA (<https://pubmed.ncbi.nlm.nih.gov/30717828/>).”

3. Yes, the DLD1 results looks a little odd – we wonder about ARRDC4 here as it looks to me as if TXNIP levels go from zero to maximum quickly. If there is redundancy here, as it seems, perhaps ARRDC4 is triggered first (it has a reported role in regulating glucose uptake - <https://pubmed.ncbi.nlm.nih.gov/19605364/>), followed by TXNIP if required? Highly speculative though!

We agree completely with your comments ~~wrt~~with respect to discussion points around these findings and so have taken the opportunity to include the following in the discussion:

“It is important to remember at this point that one of TXNIP’s key roles is to prevent ROS accumulation by preventing uptake of glucose by inhibiting GLUT receptors directly. This is driven by MondoA’s activation in response to glycolytic products, including ATP. This ‘cutting off at source’ cellular approach should not be overlooked ~~wrt~~with respect to translation, as manipulating the glucose environment, and therefore these processes, could potentially affect the potency of oxaliplatin in CRC.”

3rd Jul 2024

Dear Dr. Beatson,

We are pleased to inform you that your manuscript is accepted for publication and is now being sent to our publisher to be included in the next available issue of EMBO Molecular Medicine.
